# REINFORCEMENT LEARNING IN PRESENCE OF DISCRETE MARKOVIAN CONTEXT EVOLUTION

**Hang Ren**[*]
Huawei UK R&D

**Aivar Sootla**[*]
Huawei UK R&D

**Taher Jafferjee**
Huawei UK R&D

**Junxiao Shen**
Huawei UK R&D
University of Cambridge

**Jun Wang**
University College London
jun.wang@cs.ucl.ac.uk

**Haitham Bou-Ammar**
Huawei UK R&D and Honorary Lecturer at UCL
haitham.ammar@huawei.com

## ABSTRACT

We consider a context-dependent Reinforcement Learning (RL) setting, which is characterized by: a) an unknown finite number of not directly observable contexts; b) abrupt (discontinuous) context changes occurring during an episode; and c) Markovian context evolution. We argue that this challenging case is often met in applications and we tackle it using a Bayesian model-based approach and variational inference. We adapt a sticky Hierarchical Dirichlet Process (HDP) prior for model learning, which is arguably best-suited for infinite Markov chain modeling. We then derive a context distillation procedure, which identifies and removes spurious contexts in an unsupervised fashion. We argue that the combination of these two components allows inferring the number of contexts from data thus dealing with the context cardinality assumption. We then find the representation of the optimal policy enabling efficient policy learning using off-the-shelf RL algorithms. Finally, we demonstrate empirically (using gym environments cart-pole swing-up, drone, intersection) that our approach succeeds where state-of-the-art methods of other frameworks fail and elaborate on the reasons for such failures.

## 1 INTRODUCTION

Our world becomes more automated every day with the development of self-driving cars, robotics, and unmanned factories. Many of these automation processes rely on solutions to sequential decision-making problems. Reinforcement Learning (RL) has recently been shown to be an effective tool for solving such problems achieving notable successes, e.g., solving Atari games (Mnih et al., 2013), defeating the (arguably all-time) best human players in the game of GO (Silver et al., 2016), accelerating robot skill acquisition (Kober et al., 2013). Most of the successful RL algorithms rely on abstracting the sequential nature of the decision-making as Markov Decision Processes (MDPs), which typically assume both *stationary* transition dynamics and reward functions.

As classic RL departs a well-behaved laboratory setting, stationarity assumptions can quickly become prohibitive, sometimes leading to catastrophic consequences. As an illustration, imagine an autonomous agent driving a vehicle with changing weather conditions impacting visibility and tyre grip. The agent must identify and quickly adapt to these weather conditions changes in order to avoid losing control of the vehicle. Similarly, an unmanned aerial vehicle hovering around a fixed set of coordinates needs to deal with sudden atmospheric condition changes (e.g., wind, humidity etc). Another similar and realistic example is an actuator failure, which changes how the action affects the MDP. Following Menke & Maybeck (1995) we distinguish "soft" (a percentage drop in action efficiency) and "hard" (action is not affecting the MDP) failures. The failures can also be dynamic as a "soft" failure in one actuator can overload other actuators introducing a chain of failures. An environment with a fixed weather condition or an actuator failure can be modeled as an MDP, however, with the changing weather or arising failures the environment becomes non-stationary.

---

[*]equal contribution

We can model this type of environments by making MDP state transitions dependant on the *context* variable, which encapsulates the non-stationary and/or other dependencies. This kind of contextual Markov Decision Processes (C-MDPs) incorporate a number of different RL settings and RL frameworks (Khetarpal et al., 2020): non-stationary RL, where the context changes over time and the agent needs to adapt to the context (e.g., the weather conditions are slowly changing over time); continual and/or meta RL, where the context is sampled from a distribution before the start of the episode (e.g, the weather changes abruptly between the instances the vehicle has been deployed).

Although a significant progress in solving specific instances of C-MDPs has been made, the setting with a countable number of contexts with Markovian transitions between the contexts has not received sufficient attention in the literature — a gap we are aiming to fill. The closest related works consider only special cases of our setting assuming: no context transitions (Xu et al., 2020); Markovian context transitions with *a priori* known context transition times (Xie et al., 2020); finite state-action spaces (Choi et al., 2000). To enable sample efficient context adaptation Xu et al. (2020) and Xie et al. (2020) developed model-based reinforcement learning algorithms. Specifically, Xie et al. (2020) learned a latent space variational auto-encoder model with Markovian evolution in continuous context-space, while Xu et al. (2020) adopted a Gaussian Process model for MDPs and a Dirichlet process (DP) prior to detect context changes without modeling the context evolution. Note the DP prior, which is a conjugate prior for a categorical distribution, is not sufficient to model context transitions. We also propose a model-based RL algorithm, however, we model the context and state transitions using the Hierarchical Dirichlet Process (HDP) (Teh et al., 2006; Fox et al., 2008a) prior and a neural network with outputs parametrizing a Guassian distribution (i.e., its mean and variance), respectively, and refer to the model as HDP-C-MDP. We chose the HDP prior since it only requires the knowledge of an upper bound on context cardinality for tractable inference, and it is better suited for Markov chain modeling than other priors (Teh et al., 2006). Inspired by Blei et al. (2006) we derive a model learning algorithm using variational inference, which is amenable to RL applications using off-the-shelf algorithms.

Our algorithm relies on two theoretical results: a) we propose a context distillation procedure (i.e., removing spurious contexts); b) we show that the optimal policy depends on the context belief (context posterior probability given past observations). We derive another theoretical result, which shows performance improvement bounds for the fully observable context case. Equipped with these results, we experimentally demonstrate that we can infer the true context cardinality from data. Further, the context distillation procedure can be used during training as a regularizer. Interestingly, it can also be used to merge similar contexts, where the measure of similarity is only implicitly defined through the learning loss. Thus context merging is completely unsupervised. We then show that our model learning algorithm appears to provide an optimization profile with fewer local optima than the maximum likelihood approach, which we attribute to the Bayesian nature of our algorithm. Finally, we illustrate RL applications on an autonomous car left turn and an autonomous drone take-off tasks. We also demonstrate that state-of-the-art algorithms of different frameworks (such as continual RL and Partially-Observable Markov Decision Processes (POMDPs)) fail to solve C-MDPs in our setting, and we elaborate on potential reasons why this is the case.

## 2 Problem Formulation and Related Work

We define a contextual Markov Decision Process (C-MDP) as a tuple $\mathcal{M}_c = \langle \mathcal{C}, \mathcal{S}, \mathcal{A}, \mathcal{P}_\mathcal{C}, \mathcal{P}_\mathcal{S}, \mathcal{R}, \gamma_d \rangle$, where $\mathcal{S}$ is the continuous state space; $\mathcal{A}$ is the action space; $\gamma_d \in [0, 1]$ is the discount factor; and $\mathcal{C}$ denotes the context set with cardinality $|\mathcal{C}|$. In our setting, the state transition and reward function depend on the context, i.e., $\mathcal{P}_\mathcal{S} : \mathcal{C} \times \mathcal{S} \times \mathcal{A} \times \mathcal{S} \to [0, 1]$, $\mathcal{R} : \mathcal{C} \times \mathcal{S} \times \mathcal{A} \to \mathbb{R}$. Finally, the context distribution probability $\mathcal{P}_\mathcal{C} : \mathcal{T}_t \times \mathcal{C} \to [0, 1]$ is conditioned on $\mathcal{T}_t$ - the past states, actions and contexts $\{\boldsymbol{s}_0, \boldsymbol{a}_0, \boldsymbol{c}_0, \dots, \boldsymbol{a}_{t-1}, \boldsymbol{c}_{t-1}, \boldsymbol{s}_t\}$. Our definition is a generalization of the C-MDP definition by Hallak et al. (2015), where the contexts are stationary, i.e., $\mathcal{P}_\mathcal{C} : \mathcal{C} \to [0, 1]$. We adapt our definition in order to encompass all the settings presented by Khetarpal et al. (2020), where such C-MDPs were used but not formally defined.

Throughout the paper, we will restrict the class of C-MDPs by making the following assumptions: (a) **Contexts are unknown and not directly observed** (b) **Context set cardinality is finite and we know its upper bound** $K$; (c) **Contexts switches can occur during an episode and they are Markovian**. In particular, we consider the contexts $\boldsymbol{c}_k$ representing the parameters of the state

transition function $\boldsymbol{\theta}_k$, and the context set $\mathcal{C}$ to be a subset of the parameter space $\Theta$. To deal with uncertainty, we consider a set $\widetilde{\mathcal{C}}$ such that: a) $|\widetilde{\mathcal{C}}| = K > |\mathcal{C}|$; b) all its elements $\boldsymbol{\theta}_k \in \widetilde{\mathcal{C}}$ are sampled from a distribution $H(\lambda)$, where $\lambda$ is a hyper-parameter. Let $z_t \in [1, \ldots, K)$ be the index variable pointing toward a particular parameter vector $\boldsymbol{\theta}_{z_t}$, which leads to:

$$
\begin{aligned}
z_1 \mid \boldsymbol{\rho}_0 &\sim \mathbf{Cat}(\boldsymbol{\rho}_0), \qquad z_t \mid z_{t-1}, \{\boldsymbol{\rho}_j\}_{j=1}^{|\widetilde{\mathcal{C}}|} \sim \mathbf{Cat}(\boldsymbol{\rho}_{z_{t-1}}), \\
\boldsymbol{s}_t \mid \boldsymbol{s}_{t-1}, \boldsymbol{a}_{t-1}, z_t, \{\boldsymbol{\theta}_k\}_{k=1}^{|\widetilde{\mathcal{C}}|} &\sim p(\boldsymbol{s}_t \mid \boldsymbol{s}_{t-1}, \boldsymbol{a}_{t-1}, \boldsymbol{\theta}_{z_t}), \quad \boldsymbol{\theta}_k \mid \lambda \sim H(\lambda), t \geq 1,
\end{aligned}
\tag{1}
$$

where $\boldsymbol{\rho}_0$ is the initial context distribution, while $\boldsymbol{R} = [\boldsymbol{\rho}_1, ..., \boldsymbol{\rho}_{|\widetilde{\mathcal{C}}|}]$ represents the context transition operator.

As the reader may notice our model is tailored to the case, where the model parameters change abruptly due to external factors such as weather conditions, cascading actuator failures etc. The change is formalized by a Markov variable $z_t$, which changes the MDP parameters $\boldsymbol{\theta}_{z_t}$. Our approach can also be related to switching systems modeling (cf. Fox et al. (2008a); Becker-Ehmck et al. (2019); Dong et al. (2020)) and in this case the context is representing the system's mode. While we can draw parallels with these works, we improve the model by using nonlinear dynamics (in comparison to Becker-Ehmck et al. (2019); Fox et al. (2008a)), by using the HDP prior (Becker-Ehmck et al. (2019); Dong et al. (2020) use maximum likelihood estimators), and finally, by proposing the distillation procedure and using deep learning (in comparison to Fox et al. (2008a)). Also note that typically a switching system aims to represent a complex nonlinear (Markov) model using a collection of simpler (e.g., linear) models, which is different from our case. Other restrictions on the space of C-MDPs lead to different problems and solutions (Khetarpal et al., 2020). We briefly mention a few notable cases, while relegating a detailed discussion to Appendix A. Assuming that the context is deterministic with $z_t = t$ puts us in the non-stationary RL setting (cf. Chandak et al. (2020)), where it is common to assume a slowly or smoothly changing non-stationarity as opposed to our case of possibly abrupt changes. Restricting the context to a stationary distribution sampled in specific time points (e.g., at the start of the episode) can be tackled from the continual RL (cf. Nagabandi et al. (2018)) and meta-RL perspectives (cf. Finn et al. (2017)), but both are not designed to handle the Markovian context case. Finally, our C-MDP can be seen as a POMDP. Recall that a POMDP is defined by a tuple $\mathcal{M}_{po} = \{\mathcal{X}, \mathcal{A}, \mathcal{O}, \mathcal{P}_{\mathcal{X}}, \mathcal{P}_{\mathcal{O}}, \mathcal{R}, \gamma_d, p(\boldsymbol{x}_0)\}$, where $\mathcal{X}, \mathcal{A}, \mathcal{O}$ are the state, action, observation spaces, respectively; $\mathcal{P}_{\mathcal{X}} : \mathcal{X} \times \mathcal{A} \times \mathcal{X} \to [0, 1]$ is the state transition probability; $\mathcal{P}_{\mathcal{O}} : \mathcal{X} \times \mathcal{A} \times \mathcal{O} \to [0, 1]$ is the conditional observation probability; and $p(\boldsymbol{x}_0)$ is the initial state distribution. In our case, $\boldsymbol{x} = (\boldsymbol{s}, z)$ and $\boldsymbol{o} = \boldsymbol{s}$.

## 3 REINFORCEMENT LEARNING FOR MARKOV PROCESSES WITH MARKOVIAN CONTEXT EVOLUTION

There are three main components in our algorithm: the HDP-C-MDP derivation, the model learning algorithm using probabilistic inference and the control algorithms. We firstly briefly comment on each on these components to give an overview of the results and then explain our main contributions to each. The detailed description of all parts of our approach can be found in Appendix.

In order to learn the model of the context transitions, we choose the Bayesian approach and we employ Hierarchical Dirichlet Processes (HDP) as priors for context transitions, inspired by time-series modeling and analysis tools reported by Fox et al. (2008a;b) (see also Appendix C.1). We improve the model by proposing a context spuriosity measure allowing for reconstruction of ground truth contexts. We then derive a model learning algorithm using probabilistic inference. Having a model, we can take off-the-shelf algorithms such as a Model Predictive Control (MPC) approach using Cross-Entropy Minimization (CEM) (cf. Chua et al. (2018) and Appendix C.5), or a policy-gradient approach Soft-actor critic (SAC) (cf. Haarnoja et al. (2018) and Appendix C.6), which are both well-suited for model-based reinforcement learning. While MPC can be directly applied to our model, for policy-based control we first derive the representation of the optimal policy.

**Generative model: HDP-C-MDP.** Before presenting our probabilistic model, let us develop some necessary tools. *A Dirichlet process (DP),* denoted as $\mathbf{DP}(\gamma, H)$, is characterized by a concentration parameter $\gamma$ and a base distribution $H(\lambda)$ defined over the parameter space $\Theta$. A sample $G$ from $\mathbf{DP}(\gamma, H)$ is a probability distribution satisfying $(G(A_1), ..., G(A_r)) \sim \mathbf{Dir}(\gamma H(A_1), ..., \gamma H(A_r))$ for every finite measurable partition $A_1, ..., A_r$ of $\Theta$, where $\mathbf{Dir}$ denotes the Dirichlet distribution.

Sampling $G$ is often performed using the stick-breaking process (Sethuraman, 1994) and constructed by randomly mixing atoms independently and identically distributed samples $\boldsymbol{\theta}_k$ from $H$:

$$\nu_k \sim \mathbf{Beta}(1, \gamma), \quad \beta_k = \nu_k \prod_{i=1}^{k-1}(1 - \nu_i), \quad G = \sum_{k=1}^{\infty} \beta_k \delta_{\boldsymbol{\theta}_k}, \tag{2}$$

where $\delta_{\boldsymbol{\theta}_k}$ is the Dirac distribution at $\boldsymbol{\theta}_k$. We note that the stick-breaking procedure assigns progressively smaller values to $\beta_k$ for large $k$, thus encouraging a smaller number of meaningful atoms. *The Hierarchical Dirichlet Process (HDP)* is a group of DPs sharing a base distribution, which itself is a sample from a DP: $G \sim \mathbf{DP}(\gamma, H)$, $G_j \sim \mathbf{DP}(\alpha, G)$ for all $j = 0, 1, 2, \ldots$ (Teh et al., 2006). The distribution $G$ guarantees that all $G_j$ inherit the same set of atoms, i.e., atoms of $G$, while keeping the benefits of DPs in the distributions $G_j$. It can be shown that $G_j = \sum_{k=0}^{\infty} \rho_{jk}\delta_{\boldsymbol{\theta}_k}$ for some $\rho_{jk}$ whose sampling can be performed using another stick-breaking process (Teh et al., 2006). We consider its modified version introduced by Fox et al. (2011):

$$\mu_{jk} \mid \alpha, \kappa, \boldsymbol{\beta} \sim \mathbf{Beta}\left(\alpha\beta_k + \kappa\tilde{\delta}_{jk}, \ \alpha + \kappa - \left(\sum_{i=1}^{k}\alpha\beta_i + \kappa\tilde{\delta}_{ji}\right)\right), \quad \rho_{jk} = \mu_{jk}\prod_{i=1}^{k-1}(1 - \mu_{ji}), \tag{3}$$

where $k \geq 1$, $j \geq 0$, $\tilde{\delta}_{jk}$ is the Kronecker delta, the parameter $\kappa \geq 0$, called the sticky factor, modifies the transition matrix priors encouraging self-transitions. The sticky factor serves as another measure of regularization reducing the average number of transitions.

In our case, the atoms $\{\boldsymbol{\theta}_k\}$ forming the context set $\widetilde{\mathcal{C}}$ are sampled from $H(\lambda)$, while $\rho_{jk}$ are the parameters of the Hidden Markov Model: $\boldsymbol{\rho}_0$ is the initial context distribution and $\boldsymbol{\rho}_j$ are the rows in the transition matrix $\boldsymbol{R}$. Our probabilistic model is constructed in Equations 1,2,3 and illustrated in Figure 1 as a graphical model. We stress that the HDP in its stick-breaking construction assumes that $|\widetilde{\mathcal{C}}|$ is infinite and countable. In practice, however, we incorporate the upper cardinality bound $K$ by using a truncated variational distribution, as explained later.

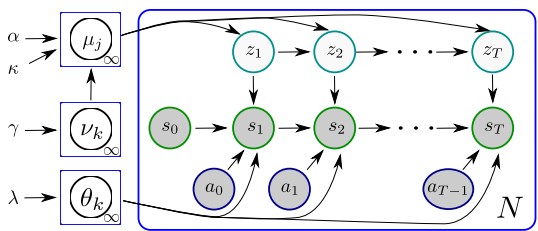

Figure 1: HDP-C-MDP

*Context Distillation.* HDP-C-MDP promotes a small number of meaningful contexts and some contexts will almost surely be spurious, i.e., we will transition to these contexts with a very small probability. While this probability is small we may still need to explicitly remove these spurious contexts. Here we propose a measure of context spuriosity and derive a distillation procedure removing these spurious contexts. As a spuriosity measure we will use the stationary distribution of the chain $\boldsymbol{p}^{\infty}$, which is computed by solving $\boldsymbol{p}^{\infty} = \boldsymbol{p}^{\infty}\boldsymbol{R}$. The distillation is then performed as follows: if in stationarity the probability mass of a context is smaller than a threshold $\varepsilon_{\text{distil}}$ then transitioning to this context is unlikely and it can be removed. We develop the corresponding distilled Markov chain in the following result, which we prove in Appendix B.1, while the distillation algorithm can be found in Appendix C.4.

**Theorem 1** *Consider a Markov chain $\boldsymbol{p}^t = \boldsymbol{p}^{t-1}\boldsymbol{R}$ with a stationary distribution $\boldsymbol{p}^{\infty}$ and distilled $\mathcal{I}_1 = \{i | \boldsymbol{p}_i^{\infty} \geq \varepsilon_{\text{distil}}\}$ and spurious $\mathcal{I}_2 = \{i | \boldsymbol{p}_i^{\infty} < \varepsilon_{\text{distil}}\}$ state indexes, respectively. Then a) the matrix $\widehat{\boldsymbol{R}} = \boldsymbol{R}_{\mathcal{I}_1, \mathcal{I}_1} + \boldsymbol{R}_{\mathcal{I}_1, \mathcal{I}_2}(\boldsymbol{I} - \boldsymbol{R}_{\mathcal{I}_2, \mathcal{I}_2})^{-1}\boldsymbol{R}_{\mathcal{I}_2, \mathcal{I}_1}$ is a valid probability transition matrix; b) the Markov chain $\widehat{\boldsymbol{p}}^t = \widehat{\boldsymbol{p}}^{t-1}\widehat{\boldsymbol{R}}$ is such that its stationary distribution $\widehat{\boldsymbol{p}}^{\infty} \propto \boldsymbol{p}_{\mathcal{I}_1}^{\infty}$.*

**Model Learning using Probabilistic Inference.** We aim to find a variational distribution $q(\boldsymbol{\nu}, \boldsymbol{\mu}, \boldsymbol{\theta})$ to approximate the true posterior $p(\boldsymbol{\nu}, \boldsymbol{\mu}, \boldsymbol{\theta}|\mathcal{D})$, for a dataset $\mathcal{D} = \{(\boldsymbol{s}^i, \boldsymbol{a}^i)\}_{i=1}^N$, where $\boldsymbol{s}^i = \{\boldsymbol{s}_t^i\}_{t=-1}^T$ and $\boldsymbol{a}^i = \{\boldsymbol{a}_t^i\}_{t=-1}^T$ are the state and action sequences in the $i$-th trajectory. We minimize $\mathcal{KL}\left(q(\boldsymbol{\nu}, \boldsymbol{\mu}, \boldsymbol{\theta}) \| p(\boldsymbol{\nu}, \boldsymbol{\mu}, \boldsymbol{\theta}|\mathcal{D})\right)$, or equivalently, maximize the evidence lower bound (ELBO):

$$\text{ELBO} = \mathbb{E}_{q(\boldsymbol{\mu}, \boldsymbol{\theta})}\left[\sum_{i=1}^N \log p(\boldsymbol{s}^i|\boldsymbol{a}^i, \boldsymbol{\mu}, \boldsymbol{\theta})\right] - \mathcal{KL}\left(q(\boldsymbol{\nu}, \boldsymbol{\mu}, \boldsymbol{\theta}) \| p(\boldsymbol{\nu}, \boldsymbol{\mu}, \boldsymbol{\theta})\right). \tag{4}$$

The variational distribution above involves infinite-dimensional random variables $\boldsymbol{\nu}, \boldsymbol{\mu}, \boldsymbol{\theta}$. To reach a tractable solution, we assume $|\widetilde{\mathcal{C}}| = K$ and exploit the standard mean-field assumption (Blei et al., 2017) and the $K$-truncated variational distribution similarly to Blei et al. (2006); Hughes et al. (2015); Bryant & Sudderth (2012) as follows:

$$q(\boldsymbol{\nu}, \boldsymbol{\mu}, \boldsymbol{\theta}) = q(\boldsymbol{\nu})q(\boldsymbol{\mu})q(\boldsymbol{\theta}), \ q(\boldsymbol{\theta}|\hat{\boldsymbol{\theta}}) = \prod_{k=1}^{K} \delta(\boldsymbol{\theta}_k|\hat{\boldsymbol{\theta}}_k), \ q(\boldsymbol{\nu}|\hat{\boldsymbol{\nu}}) = \prod_{k=1}^{K-1} \delta(\nu_k|\hat{\nu}_k), \ q(\nu_K = 1) = 1,$$

$$q(\boldsymbol{\mu}|\hat{\boldsymbol{\mu}}) = \prod_{j=0}^{K} \prod_{k=1}^{K-1} \mathbf{Beta}\left(\mu_{jk}\Big|\hat{\mu}_{jk}, \hat{\mu}_j - \sum_{i=1}^{k} \hat{\mu}_{ji}\right), \quad q(\mu_{jK} = 1) = 1, \tag{5}$$

where hatted symbols represent free parameters. For $\boldsymbol{\theta}$ and $\boldsymbol{\nu}$, we seek a MAP point estimate instead of a full posterior (see Appendix C.3 for a discussion on our design choices). Random variables not shown in the truncated variational distribution are conditionally independent of data, and thus can be discarded from the problem. We maximize ELBO using stochastic gradient ascent, while the gradient computations are performed using the following two techniques: (a) we compute the exact context posterior using a forward-backward message passing algorithm, (b) we use implicit reparametrized gradients to differentiate with respect to parameters of variational distributions (Figurnov et al., 2018). We present the detailed derivations in Appendix C.2. We also can perform context distillation during training as discussed in Appendix C.4. While adding some computational complexity, this procedure acts as a regularization for model learning as we show in our experiments.

**Representation of the optimal policy.** First, we notice that the model in Equation 1 is a POMDP, which we get by setting $\boldsymbol{x}_t := (z_t, \boldsymbol{s}_t)$ and $\boldsymbol{o}_t := \boldsymbol{s}_t$. In the POMDP case, we cannot claim that $\boldsymbol{o}_{t+1}$ depends *only* on $\boldsymbol{o}_t$ and $\boldsymbol{a}_t$. Therefore the Bellman dynamic programming principle does not hold for these variables and solving the problem is more involved. In practice, one constructs *the belief state* $\boldsymbol{b}_t = p(\boldsymbol{x}_t|\boldsymbol{I}_t^C)$ (Astrom, 1965), where $\boldsymbol{I}_t^C = \{\boldsymbol{b}_0, \boldsymbol{o}_{\leq t}, \boldsymbol{a}_{<t}\}$ is called the information state and is used to compute the optimal policy. Since the belief state is a distribution, it is generally costly to estimate in continuous observation or state spaces. In our case, estimating the belief is tractable, since the belief of the state $\boldsymbol{s}_t$ is the state itself (as the state $\boldsymbol{s}_t$ is observable) and the belief of $z_t$, which we denote as $\boldsymbol{b}_t^z$, is a vector of a fixed length at every time step (as $z_t$ is discrete and $\boldsymbol{b}_t^z$ is its filtering distribution). We have the following result with the proof in Appendix B.2.

**Theorem 2** *a) The belief of $z$ can be computed as $p(z_{t+1}|\boldsymbol{I}_t^C) = \boldsymbol{b}_{t+1}^z$, where $(\boldsymbol{b}_{t+1}^z)_i \propto \boldsymbol{N}_i = \sum_j p(\boldsymbol{s}_{t+1}|\boldsymbol{s}_t, \boldsymbol{\theta}_i, \boldsymbol{a}_t)\rho_{ji}(\boldsymbol{b}_t^z)_j$, where $(\boldsymbol{b}_t^z)_i$ are the entries of $\boldsymbol{b}_t^z$; b) the optimal policy can be computed as $\pi(\boldsymbol{s}, \boldsymbol{b}^z) = \operatorname{argmax}_{\boldsymbol{a}} \boldsymbol{Q}(\boldsymbol{s}, \boldsymbol{b}^z, \boldsymbol{a})$, where the value function satisfies the dynamic programming principle $\boldsymbol{Q}(\boldsymbol{s}_t, \boldsymbol{b}_t^z, \boldsymbol{a}_t) = \boldsymbol{r}(\boldsymbol{s}_t, \boldsymbol{b}_t^z, \boldsymbol{a}_t) + \gamma \int \sum_i \boldsymbol{N}_i \max_{\boldsymbol{a}_{t+1}} \boldsymbol{Q}(\boldsymbol{s}_{t+1}, \boldsymbol{b}_{t+1}^z, \boldsymbol{a}_{t+1}) \, d\boldsymbol{s}_{t+1}$.*

**Computational framework.** Algorithm 1 summarizes our approach and is based on the standard model-based RL frameworks (e.g., Pineda et al. (2021)). Effectively, we alternate between model updates and policy updates. For the policy updates we relabel (recompute) the beliefs for the historical transition data. As MPC methods compute the sequence of actions based solely on the model such relabeling is not required.

**Performance gain for observable contexts.** It is not surprising that observing the ground truth of the contexts should improve the maximum expected return. In particular, even knowing the ground truth context model we can correctly estimate the context $z_{t+1}$ only *a posteriori*, i.e., after observing the next state $\boldsymbol{s}_{t+1}$. Therefore at every context switch we can mislabel it with a high probability. This leads to a performance loss, which the following result quantifies using the value functions. We have the following result with the proof in Appendix B.3.

**Theorem 3** *Assume we know the true transition model of the contexts and states and consider two settings: we observe the ground truth $z_t$ and we estimate it using $\boldsymbol{b}_t^z$. Assume we computed the optimal model-based policy $\pi(\cdot|\boldsymbol{s}_t, \boldsymbol{b}_t^z)$ with the return $\mathcal{R}$ and the optimal ground-truth policy $\pi_{\mathrm{gt}}(\cdot|\boldsymbol{s}_t, z_{t+1})$ with the corresponding optimal value functions $V_{\mathrm{gt}}(\boldsymbol{s}, z)$ and $Q_{\mathrm{gt}}(\boldsymbol{s}, z, \boldsymbol{a})$, then:*

$$\mathbb{E}_{z_1, \boldsymbol{s}_0} V_{\mathrm{gt}}(\boldsymbol{s}_0, z_1) - \mathcal{R} \geq \mathbb{E}_{\tau, \boldsymbol{a}_{t_m}^{\mathrm{gt}} \sim \pi_{\mathrm{gt}}, \boldsymbol{a}_{t_m} \sim \pi} \sum_{m=1}^{M} \gamma^{t_m}(Q(\boldsymbol{s}_{t_m}, z_{t_m+1}, \boldsymbol{a}_{t_m}^{\mathrm{gt}}) - Q(\boldsymbol{s}_{t_m}, z_{t_m+1}, \boldsymbol{a}_{t_m})),$$

*where $M$ is the number of misidentified context switches in a trajectory $\tau$.*

---

**Algorithm 1:** Learning to Control HDP-C-MDP

---

**Input:** $\varepsilon_{\text{distill}}$ - distillation threshold, $N_{\text{warm}}$ - number of trajectories for warm start, $N_{\text{traj}}$ - number of newly collected trajectories per epoch, $N_{\text{epochs}}$ - number of training epochs, AGENT - policy gradient or MPC agent

Initialize AGENT with RANDOM AGENT, $\mathcal{D} = \emptyset$;

**for** $i = 1, \dots, N_{\text{epochs}}$ **do**

    Sample $N_{\text{traj}}$ ($N_{\text{warm}}$ if $i = 1$) trajectories from the environment with AGENT;

    Set $\mathcal{D}_{\text{new}} = \{(\boldsymbol{s}^i, \boldsymbol{a}^i)\}_{i=1}^{N_{\text{traj}}}$, where $\boldsymbol{s}^i = \{\boldsymbol{s}_t^i\}_{t=-1}^T$ and $\boldsymbol{a}^i = \{\boldsymbol{a}_t^i\}_{t=-1}^T$ are the state and action sequences in the $i$-th trajectory. Set $\mathcal{D} = \mathcal{D} \cup \mathcal{D}_{\text{new}}$;

    Update generative model parameters by gradient ascent on ELBO in Equation 4;

    Perform context distillation with $\varepsilon_{\text{distill}}$;

    **if** AGENT *is* POLICY **then**

        Sample trajectories for policy update from $\mathcal{D}$;

        Recompute the beliefs using the model for these trajectories;

        Update policy parameters

    **end**

**end**

**return** AGENT

---

## 4 EXPERIMENTS

In this section, we demonstrate that the HDP offers an effective prior for model learning, while the distillation procedure refines the model and can regulate the context set complexity. We also explain why state-of-the-art methods from continual RL, meta-RL and POMDP literature can fail in our setting. We finally show that our algorithm can be adapted to high dimensional environments. We delegate several experiments to Appendix due to space limitations. We show that we can learn additional unseen contexts without relearning the whole model from scratch. We also illustrate how the context distillation during training can be used to merge contexts in an unsupervised manner thus reducing model complexity. We finally show that our model can generalize to non-Markovian and state dependent context transitions.

We choose **the switching process** to be a chain, however, we enforce a cool-off period, i.e, the chain cannot transition to a new state until the cool-off period has ended. This makes the context switching itself a non-stationary MDP. This is done to avoid switches at every time step, but also to show that our method is not limited to the stationary Markov context evolution.

**Control Baselines:** (1) SAC algorithm with access to the ground truth context information (one-hot-encoded variable $z_t$) denoted as *FI-SAC*; (2) SAC algorithm with no context information denoted as *NI-SAC* (3) a continual RL algorithm for contextual MDPs (Xu et al., 2020), where a Gaussian process is used to learn the dynamics while identifying and labeling the data with contexts, which is denoted as *GPMM*; (4) A POMDP approach, where the context set cardinality is known and the belief is estimated using an RNN, while PPO (Schulman et al., 2017; Kostrikov, 2018) is used to update the policy. We denote this approach as *RNN-PPO*.

**Modeling Prior Baselines:** (1) a model with sticky Dirichlet priors $\boldsymbol{\rho}_j \sim \mathbf{Dir}(\alpha_k = \alpha/K + \kappa\tilde{\delta}_{jk})$; (2) a model which removes all priors and conducts a maximum-likelihood (MLE) learning. All the other relevant experimental details (including hyper-parameters) are provided in Appendix D.

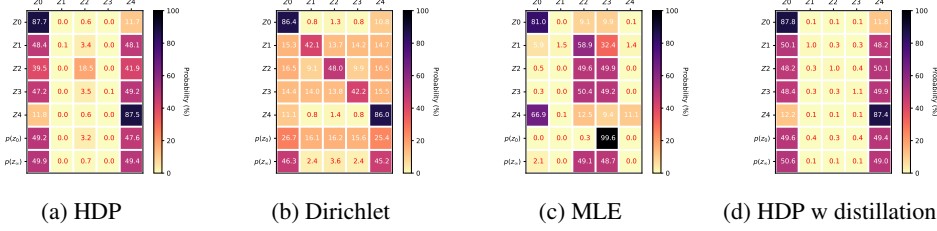

(a) HDP      (b) Dirichlet      (c) MLE      (d) HDP w distillation

Figure 2: Cart-Pole Swing-Up. Transition matrices, initial $p(z_0)$ and stationary $p(z_\infty)$ distributions of the learned context models for Result A. $Z0 - Z4$ stand for the learned contexts.

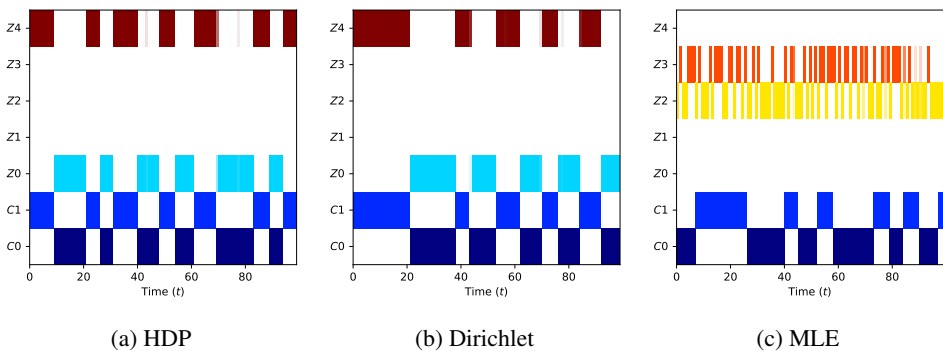

|                        |                        |                        |
|:----------------------:|:----------------------:|:----------------------:|
| (a) HDP                | (b) Dirichlet          | (c) MLE                |

Figure 3: Cart-Pole Swing-Up. Time courses the learned context models for Result A. $C0$ and $C1$ stand for the ground true contexts, while $Z0 - Z4$ are the learned contexts.

**Initial testing on Cart-Pole Swing-up Task (Lovatto, 2019).** We attempt to swing up and balance a pole attached to a cart. This environment has four states and one action. We introduce the contexts by multiplying the action with a constant $\chi$ thus modulating the actuation effect. We will allow negative $\chi$ modeling catastrophic (or hard) failures, and positive $\chi$ modeling soft actuation failures. **Result A: HDP is an effective prior for learning an accurate and interpretable model.** In Figure 2, we plot the expectation of $\boldsymbol{\rho}_0$ and $\boldsymbol{R}$ extracted from the variational distribution $q(\boldsymbol{\mu})$ for HDP, Dirichlet and MLE priors for the Cart-Pole Swing-up Environment with the context set $\mathcal{C} = \{1, -1\}$ and $|\widetilde{\mathcal{C}}| = K = 5$. The MLE learning appears to be trapped in a local optimum as the results in Figure 2(c) suggest. A similar phenomenon has been reported by Dong et al. (2020), where an MLE method was used and a heuristic entropy regularization and temperature annealing method is adopted to alleviate the issue. All in all, while MLE learning can appear to be competitive with a different random seed, this approach does not give consistent results. The use of Dirichlet priors appears to provide a better model. Furthermore, with an appropriate distillation threshold the distilled transition matrices with HDP and Dirichlet priors are very similar to each other. However, the threshold for Dirichlet prior distillation needs to be much higher as calculations of the stationary distributions suggest. This implies that spurious transitions are still quite likely. In contrast, the HDP prior helps to successfully identify two main contexts ($Z0$ and $Z2$) and accurately predict the context evolution (see Figure 3). Furthermore, the model is more interpretable and the meaningful contexts can often be identified with a naked eye.

**Result B: Distillation acts as a regularizer.** We noticed that the context $Z2$ has a low probability mass in stationarity, but a high probability of self-transition (Figure 2(a)). This suggest that spurious transitions can happen, while highly unlikely. We speculate that the learning algorithm tries to fit the uncertainty in the model (e.g., due to unseen data) to one context. This can lead to over-fitting and unwanted side-effects. Results in Figure 2(d) suggest that distillation during training can act as a regularizer when we used a high enough threshold $\varepsilon_{\text{distil}} = 0.1$. We proceed by varying the context set cardinality $|\widetilde{\mathcal{C}}|$ (taking values 4, 5, 6, 8, 10 and 20) and the distillation threshold $\varepsilon_{\text{distil}}$ (taking values 0, 0.01, and 0.1). Note that we distill during training and we refer to the transition matrix for the distilled Markov chain as the distilled transition matrix. As the ground truth context cardinality is equal to two, the probability of the third most likely context would signify the learning error. In Table 1, we present the stationary probability of the context with the third largest probability mass. In particular, for $|\widetilde{\mathcal{C}}| = 20$ the probability mass values for this context are larger than 0.01. This indicates a small but not insignificant possibility of a transition to this context, if the distillation does not remove this context. We present some additional details on this experiment in Appendix E.1. Overall, we can conclude that it is safe to overestimate the context cardinality.

**Result C: MDP, POMDP and continual RL methods can be ineffective.** In Figure 4(a), we plot the learning curves for our algorithms and compare them to each other for $\chi = -1$. CEM, which is known to perform well in low-dimensional environments, learns faster than SAC. Note that there is no significant performance loss of C-SAC in comparison with the full information case exhibiting the power of our modeling approach. We evaluated FI-SAC, C-SAC, C-CEM on three seeds.

We now compare the control algorithms for various values of $\chi$. We present the results of our experiments in Table 2 and we also discuss the comparison protocols in Appendix E.6. Here we

Table 1: Comparing the probability mass of the third most probable state in the stationary distribution. We vary the cardinality of the estimated context set $\widetilde{\mathcal{C}}$ and the distillation threshold $\varepsilon_{\text{distil}}$. Red indicates underestimation of distillation threshold.

| $\varepsilon_{\text{distil}} \downarrow$ $\quad$ $|\widetilde{\mathcal{C}}| \rightarrow$ | 4 | 5 | 6 | 8 | 10 | 20 |
|---|---|---|---|---|---|---|
| **0** | **8.58e-03** | **7.06e-03** | **3.71e-03** | **6.85e-03** | **2.20e-03** | **2.25e-02** |
| **0.01** | 1.06e-03 | 1.24e-03 | 1.37e-03 | 2.19e-03 | 2.56e-03 | **1.60e-02** |
| **0.1** | 1.21e-03 | 1.54e-03 | 1.70e-03 | 2.80e-03 | 3.54e-03 | 9.86e-03 |

(a) Learning curves    (b) GPMM $\chi = -1$    (c) GPMM $\chi = 0.5$    (d) RNN "Beliefs"

Figure 4: Cart-Pole Swing-Up. Learning curves for $\chi = -1$ (a), time courses the learned context models using GPMM (b)-(c) and the learned model belief by RNN-PPO (d).

focus on the reasons why both RNN-PPO and GPMM can fail in some experiments and seem to perform well in others. In GPMM, it is explicitly assumed that the context does not change during the episode, however, the algorithm can adapt to a new context. While *a posteriori* context estimation has a limited success for $\chi = 0.5$ (see Figure 4), the context adaptation is rather slow for our setting resulting in many context estimation errors, which reduces the performance. Furthermore, it appears that estimating hard failures is a challenge for GPMM. RNN-PPO appears to perform very well for $\chi > 0$ (see Table 2) and fail for $\chi = -1$, however, when we plot the output of the RNN, which is meant to predict the beliefs, we see that the average context prediction is quite similar across different experiments (see Figure 4). It is worth noting that the mean of the true belief variable is $0.5$ for all $\chi$, as both contexts are equally probable at every time step. Therefore, the RNN approach does not actually learn a belief model, but an "average" adjustment signal for the policy, and hence it will often fail to solve a C-MDP. Interestingly, with $\chi = 0.5$ our modeling algorithm learns only **one meaningful context** with high distillation threshold while still solving the task. This is because for both $\chi = 0.5$ and $\chi = 1$ the sign of optimal actions for swing up are the same and both have sufficient power to solve the task. We compare to further baselines in Appendix E.6.

**Our model is effective for control in twelve dimensional environments (Drone and Intersection)**. In the drone environment (Panerati et al., 2021), the agent aims at balancing roll and pitch angles of the drone, while accelerating vertically, i.e., the task is to maximize the upward velocity. This environment has twelve states (positions, velocities, Euler angles, angular velocities in three dimensions) and four actions (motor speeds in rotation per minute). In the highway intersection environment (Leurent, 2018), the agent aims at performing the unprotected left turn maneuver with an incoming vehicle turning in the same direction. The goal of the agent is to make the left turn and follow the social vehicle without colliding with it. The agent measures positions, velocities and headings in $x$, $y$ axes of the ego and social vehicles (twelve states in total), while controlling the steering angle and acceleration / deceleration. In both environments we introduce the contexts *by multiplying the maximum actuation effect by a constant $\chi$*, specifically, motor speeds in the drone environment and steering angle in the highway intersection environment. The results in Table 3 demonstrate that both MPC and policy learning approaches with the model are able to solve the task, while using no information (NI) about the contexts dramatically reduces the performance. Note that in the drone environment C-CEM algorithm exhibits slightly better performance than C-SAC, while in the intersection environment C-SAC controls the car much better. We can only hypothesize that the policy's feedback architecture (mapping states to actions) is better suited for complex tasks such as low-level vehicle control, where MPC approaches require a substantial tuning and computational effort to compete with a policy based approach.

| failure → 
 algo ↓ | hard | soft $\alpha = 0.1$ | soft $\alpha = 0.3$ | soft $\alpha = 0.5$ |
|---|---|---|---|---|
| FI-SAC | $84.50 \pm 1.79$ | $\mathbf{76.63 \pm 8.54}$ | $\mathbf{84.75 \pm 3.07}$ | $\mathbf{86.92 \pm 1.03}$ |
| C-SAC | $85.38 \pm 1.64$ | $\mathbf{76.80 \pm 8.91}$ | $\mathbf{86.76 \pm 2.88}$ | $\mathbf{88.35 \pm 1.30}$ |
| C-CEM | $\mathbf{87.63 \pm 0.14}$ | $60.15 \pm 25.91$ | $83.15 \pm 7.72$ | $\mathbf{89.08 \pm 1.90}$ |
| GPMM | $3.50 \pm 18.59$ | $3.55 \pm 7.83$ | $10.64 \pm 16.10$ | $49.61 \pm 19.13$ |
| RNN-PPO | $-0.17 \pm 18.06$ | $64.10 \pm 21.37$ | $74.58 \pm 20.66$ | $67.01 \pm 8.52$ |

Table 2: Mean $\pm$ standard deviation of expected return for: our algorithms (C-SAC, C-CEM), a continual RL algorithm (GPMM), a POMDP algo (RNN-PPO), and SAC with a known context (FI-SAC). For soft failure experiments, we have increased the maximum applicable force by the factor of two. Best performances are highlighted in bold.

| | FI-SAC | NI-SAC | C-SAC | C-CEM |
|---|---|---|---|---|
| Drone | $\mathbf{36.13 \pm 0.26}$ | $-0.80 \pm 3.08$ | $28.41 \pm 1.16$ | $32.30 \pm 2.78$ |
| Intersection | $\mathbf{572.09 \pm 20.25}$ | $499.62 \pm 19.98$ | $555.11 \pm 20.21$ | $529.75 \pm 78.12$ |

Table 3: Mean $\pm$ standard deviation of expected return for various algorithms and tasks over three seeds. Our contextual approaches, which marked by the letter C, are competitive with FI-SAC (SAC with full context information) and outperform NI-SAC (SAC with no context information).

## 5 CONCLUSION AND DISCUSSION

We studied a hybrid discrete-continuous variable process, where unobserved discrete variable represents the context and observed continuous variables represents the dynamics state. We proposed a variational inference algorithm for model learning using a sticky HDP prior. This prior allows for effective learning of an interpretable model and coupled with our context distillation procedure offers a powerful tool for learning C-MDPs. In particular, we showed that the combination of the HDP prior and the context distillation method allows learning the true context cardinality. We also showed that the model quality is not affected if the upper bound on context cardinality set is overestimated. Furthermore, we illustrated that the distillation threshold can be used as a regularization trade-off parameter and it can also be used to merge similar contexts in an unsupervised manner. Furthermore, we present additional experiments in Appendix suggesting that our model can potentially generalize to non-Markovian and state-dependent settings. While we presented several experiments in various environments, further experimental evaluation is required, e.g., using Benjamins et al. (2021).

We showed that continual and meta-RL approaches are likely to fail as their underlying assumptions on the environment do not fit our setting. The learned models do not appear to capture the complexity of Markovian context transitions. This, however, should not be surprising as these methods are tailored to a different problem: adapting existing policy / model to a new setting. If the context is very different and / or the contexts changing too fast then the continual and meta-RL algorithms would struggle by design. We derived our policy by exploiting the relation of our setting and POMDPs. We demonstrated the necessity of our model by observing that standard POMDP approaches (i.e., modeling the context dynamics using an RNN) fail to learn the model. We attribute this behavior to the lack of effective priors and model structure. While in some cases it can appear that the RNN policy is effective, disregarding the context altogether has a similar effect.

Our model-based algorithm can be further enhanced by using synthetic one-step transitions similarly to Janner et al. (2019), which would improve sample efficiency. We can also use an ensemble of models, which would allow to constantly improve the model using cross-validation over models in the ensemble. However, evaluation of the model quality is more involved since the context is unobservable. In future, we also plan to extend our model to account for a partially observable setting, i.e., where the state only indirectly measured similarly to POMDPs. This setting would allow for a rigorous treatment of controlling from pictures in the context-dependent setting. While we show that we can learn unseen contexts without re-learning the entire model, this procedure is not fully automated. Hence it can benefit from adaptation of continual learning methods in order to increase efficiency of the learning procedure.

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

# Appendices

## A  DETAILED LITERATURE REVIEW

For convenience, we reproduce the definitions and assumptions from the main text to make the appendix self-contained. We define a contextual Markov Decision Process (C-MDP) as a tuple $\mathcal{M}_c = \langle \mathcal{C}, \mathcal{S}, \mathcal{A}, \mathcal{P}_{\mathcal{C}}, \mathcal{P}_{\mathcal{S}}, \mathcal{R}, \gamma_d \rangle$, where $\mathcal{S}$ is the continuous state space; $\mathcal{A}$ is the action space; $\gamma_d \in [0, 1]$ is the discount factor; and $\mathcal{C}$ denotes the context set with cardinality $|\mathcal{C}|$. In our setting, the state transition and reward function depend on the context, i.e., $\mathcal{P}_{\mathcal{S}} : \mathcal{C} \times \mathcal{S} \times \mathcal{A} \times \mathcal{S} \to [0, 1]$, $\mathcal{R} : \mathcal{C} \times \mathcal{S} \times \mathcal{A} \to \mathbb{R}$. Finally, the context distribution probability $\mathcal{P}_{\mathcal{C}} : \mathcal{T}_t \times \mathcal{C} \to [0, 1]$ is conditioned on $\mathcal{T}_t$ - the past states, actions and contexts $\{s_0, a_0, c_0, \ldots, a_{t-1}, c_{t-1}, s_t\}$. Our definition is a generalization of the C-MDP definition by Hallak et al. (2015), where the contexts are stationary, i.e., $\mathcal{P}_{\mathcal{C}} : \mathcal{C} \to [0, 1]$. We adapt our definition in order to encompass all the settings presented by Khetarpal et al. (2020), where such C-MDPs were used but not formally defined.

Throughout the paper, we will restrict the class of C-MDPs by making the following assumptions: (a) **Contexts are unknown and not directly observed** (b) **Context cardinality is finite and we know its upper bound** $K$; (c) **Context distribution is Markovian**. In particular, we consider the contexts $c_k$ representing the parameters of the state transition function $\boldsymbol{\theta}_k$, and the context set $\mathcal{C}$ to be a subset of the parameter space $\Theta$. To deal with uncertainty, we consider a set $\widetilde{\mathcal{C}}$ such that: a) $|\widetilde{\mathcal{C}}| = K > |\mathcal{C}|$; b) all its elements $\boldsymbol{\theta}_k \in \widetilde{\mathcal{C}}$ are sampled from a distribution $H(\lambda)$, where $\lambda$ is a hyper-parameter. Let $z_t \in [0, \ldots, K)$ be the index variable pointing toward a particular parameter vector $\boldsymbol{\theta}_{z_t}$. We thus

write the environment model as:

$$z_0 \mid \boldsymbol{\rho}_0 \sim \mathbf{Cat}(\boldsymbol{\rho}_0), \qquad z_t \mid z_{t-1}, \{\boldsymbol{\rho}_j\}_{j=1}^{|\widetilde{\mathcal{C}}|} \sim \mathbf{Cat}(\boldsymbol{\rho}_{z_{t-1}}), \tag{A1a}$$

$$\boldsymbol{s}_t \mid \boldsymbol{s}_{t-1}, \boldsymbol{a}_{t-1}, z_t, \{\boldsymbol{\theta}_k\}_{k=1}^{|\widetilde{\mathcal{C}}|} \sim p(\boldsymbol{s}_t | \boldsymbol{s}_{t-1}, \boldsymbol{a}_{t-1}, \boldsymbol{\theta}_{z_t}), \quad \boldsymbol{\theta}_k \mid \lambda \sim H(\lambda), t \geq 1. \tag{A1b}$$

For convenience we also write $\boldsymbol{R} = [\boldsymbol{\rho}_1, ..., \boldsymbol{\rho}_{|\widetilde{\mathcal{C}}|}]$ representing the context transition operator, even if $|\widetilde{\mathcal{C}}|$ is countable and infinite.

Other restrictions on the space of C-MDPs lead to different problems and solutions. We present some settings graphically in Figure A1 adapted from the review by Khetarpal et al. (2020).

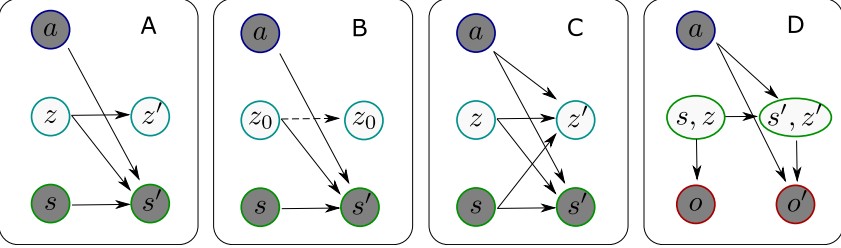

Figure A1: Graphical models for C-MDP modeling. In all panels $a$ stands for action, $s$ - state, $o$ - observation, $z$ - context, while the operator $'$ indicates the next time step. A: The context variable $z$ evolves according to a Markov process; B: The context variable $z$ is drawn once per episode; C: The context variable $z$ evolves according to a Markov process and depends on state and/or action variables; D: a POMDP.

**Setting A.** In this setting, a Markov assumption is made on the context evolution. Only a few works make such an assumption similarly to our work. For example, Choi et al. (2000) assume a discrete state, action and context spaces and fix the exact number of contexts thus significantly limiting applicability of their approach. We can also mention the work by Xu et al. (2020), where the individual points are classified to different context during the episode. However, the agent explicitly assumes that the context will not change during the execution of its plan. The Markovian context evolution can also be related to switching systems (cf. Becker-Ehmck et al. (2019); Fox et al. (2008a); Dong et al. (2020)), where the context is representing the system's mode. While we take inspiration from these works, we improve the model in comparison to Becker-Ehmck et al. (2019); Fox et al. (2008a) by using non-linear dynamics, in comparison to Fox et al. (2008a) by proposing the distillation procedure and using deep learning, in comparison to Becker-Ehmck et al. (2019); Dong et al. (2020) by using the hierarchical Dirichlet process prior.

**Setting B.** Here, an episodic non-stationarity constraint on the context probability distribution $\mathcal{P}_\mathcal{C}$ is introduced, that is, the context may change only (without loss of generality) at the start of the episode. Episodic non-stationarity constraint is widely adopted by meta-RL (Doshi-Velez & Konidaris, 2016; Finn et al., 2017; Al-Shedivat et al., 2018; Rothfuss et al., 2019; Clavera et al., 2018; Rakelly et al., 2019; Zintgraf et al., 2020; Xie et al., 2020) and continual RL (Chandak et al., 2020; Nagabandi et al., 2018; Khetarpal et al., 2020; Bou-Ammar et al., 2014; Riemer et al., 2019; Rolnick et al., 2019) communities. We can distinguish optimization-based and context-based Meta-RL families of methods. Optimization-based Meta-RL methods (Finn et al., 2017; Al-Shedivat et al., 2018; Rothfuss et al., 2019; Clavera et al., 2018) sample training contexts from a stationary distribution $\mathcal{P}_\mathcal{C}$ and optimize model or policy parameters, which can be quickly adapted to a test context. In contrast, context-based Meta-RL methods (Rakelly et al., 2019; Zintgraf et al., 2020; Xie et al., 2020; Duan et al., 2016) attempt to infer a deterministic representation or a probabilistic belief on the context from the episode history. We also remark recent work on domain shift while controlling from pixels Hansen & Wang (2021); Kostrikov et al. (2020). In these papers the authors assume that some pixels may change from an episode to an episode (e.g., red color becomes blue), but the underlying dynamics stay the same. Besides the episodic nature of the context change, this setting differs from ours on two fronts. First, controlling from pictures constitutes a POMDP problem, where the true states (positions, velocities, accelerations) are observed through a proxy (through pixels) and hence need to be inferred. Second, the underlying dynamics stay the same (Hansen & Wang, 2021; Kostrikov et al., 2020) and only observations change. Our case is the opposite: the underlying dynamics change, but the observation

function stays the same. In future work we aim to extend our methods to control from pixels.We also note the work by Ball et al. (2021), who proposed to learn a contextual policy by augmenting an offline world model. In the online setting the algorithm had to learn only the current context thus improving efficiency of the learning procedure while delivering good performance. Continual (lifelong) RL mainly adopts a context incremental setting, where the agent is exposed to a sequence of contexts (Delange et al., 2021). While the agent's goal is still to adapt efficiently to the unseen contexts, the emphasis is on overcoming catastrophic forgetting, i.e., maintaining a good performance on old contexts while improving performance on the current one (Rolnick et al., 2019; Riemer et al., 2019).

We further remark that the Markovian context model and episodic non-stationarity assumptions are often incompatible. Indeed, learning the transition model for $\mathcal{P}_\mathcal{C}$ in C-MDPs with an episodic non-stationarity constraint is somewhat redundant because $\mathcal{P}_\mathcal{C}$ can be assumed to be a stationary distribution rather than a Markov process. On the other hand, if the context changes according to a Markov process then the episodic non-stationarity assumption may not be enough to capture the rich context dynamics. This can lead not only to suboptimal policies, but also to policies not solving the task at all. Xie et al. (2020) tried to combine the two settings by taking a hierarchical view on non-stationary modeling. In particular, they assume that the context changes in a Markovian fashion, but only between the episodes and during the episode the context does not change. This setting allows to model a two time-scale process: the context transitions on a slow time-scale prescribed by the episodes, while the process transitions on a fast time-scale prescribed by the steps. While this approach has its merits, it also has the some limitations, e.g., the hierarchical model is artificially imposed on the learning process.

**Settings C and D.** Many frameworks can fit into the settings C and D, (c.f., Ong et al. (2010); Chandak et al. (2019); Zhang et al. (2020)), however, this makes it hard to compare the differences and similarities between them. We will refer the reader to the review by Khetarpal et al. (2020) for a further discussion. We make a few comments, however, on change point detection methods and on the relation to POMDP formulation (Astrom, 1965; Hauskrecht, 2000) in the case of unobservable contexts. Change-point detection methods can be readily used for the context estimation in an online fashion (da Silva et al., 2006; Hadoux et al., 2014; Banerjee et al., 2017; Padakandla et al., 2019; Li et al., 2019), however, these methods either track change-points in a heuristic way (da Silva et al., 2006; Hadoux et al., 2014) or assume strong prior knowledge on the non-stationarity (Banerjee et al., 2017; Padakandla et al., 2019). A POMDP is defined by tuple $\mathcal{M}_{po} = \{\mathcal{X}, \mathcal{A}, \mathcal{O}, \mathcal{P}_\mathcal{X}, \mathcal{P}_\mathcal{O}, \mathcal{R}, \gamma_d, p(\boldsymbol{x}_0)\}$, where $\mathcal{X}, \mathcal{A}, \mathcal{O}$ are the (unobservable) state, action, observation spaces, respectively; $\mathcal{P}_\mathcal{X} : \mathcal{X} \times \mathcal{A} \times \mathcal{X} \to [0,1]$ is the state transition probability; $\mathcal{P}_\mathcal{O} : \mathcal{X} \times \mathcal{A} \times \mathcal{O} \to [0,1]$ is the conditional observation probability; and $p(\boldsymbol{x}_0)$ is the initial state distribution. In our case, the state is $\boldsymbol{x}_t = (\boldsymbol{s}_t, z_t)$ and the observation is $\boldsymbol{o}_t = \boldsymbol{s}_t$. A key step in POMDP literature is estimating or approximating the belief, i.e., the probability distribution of the current state using history of observations and actions (Hausknecht & Stone, 2015; Zhu et al., 2017; Igl et al., 2018). Since the belief is generally infinite-dimensional, it is prudent to resort to sampling, point-estimates or other approximations (Hausknecht & Stone, 2015; Zhu et al., 2017; Igl et al., 2018). In our case, this is not necessary, however, as the context is a discrete variable. We further stress, that while general C-MDPs can be represented using POMDPs, this representation may not offer any benefits at all due to complexity of belief estimation.

Finally, we can classify the literature by cardinality of $\mathcal{C}$. Some Meta-RL algorithms mainly consider a continuous contextual set $\mathcal{C}$ by implicitly or explicitly parametrizing $\mathcal{C}$ with real-valued vectors (Doshi-Velez & Konidaris, 2016; Zintgraf et al., 2020; Xie et al., 2020; Rakelly et al., 2019). On the other hand, exploiting a discrete $\mathcal{C}$ can be motivated by controlling switching dynamic systems (Fox et al., 2008a) and may achieve better interpretability. Due to complexity, however, some works fix the contextual cardinality $|\mathcal{C}|$ (Choi et al., 2000; Banerjee et al., 2017; Padakandla et al., 2019; Lee et al., 2019), or infer $|\mathcal{C}|$ from data in an online fashion (Xu et al., 2020; da Silva et al., 2006; Hadoux et al., 2014). Besides, most works in continual RL adopt a setting, where there is no explicit specification on $\mathcal{C}$, but the agent can directly access the discrete contexts of all time steps/episodes (Rolnick et al., 2019; Riemer et al., 2019). In contrast, our approach learns the context evolution from data.

## B   PROOFS

### B.1   PROOF OF THEOREM 1

For completeness, we restate the theorem below.

**Theorem A1** *Consider a Markov chain $\boldsymbol{p}^t = \boldsymbol{p}^{t-1}\boldsymbol{R}$ with a stationary distribution $\boldsymbol{p}^\infty$ and distilled $\mathcal{I}_1 = \{i | \boldsymbol{p}_i^\infty \geq \varepsilon_{\text{distil}}\}$ and spurious $\mathcal{I}_2 = \{i | \boldsymbol{p}_i^\infty < \varepsilon_{\text{distil}}\}$ state indexes, respectively. Then a) the matrix $\widehat{\boldsymbol{R}} = \boldsymbol{R}_{\mathcal{I}_1,\mathcal{I}_1} + \boldsymbol{R}_{\mathcal{I}_1,\mathcal{I}_2}(\boldsymbol{I} - \boldsymbol{R}_{\mathcal{I}_2,\mathcal{I}_2})^{-1}\boldsymbol{R}_{\mathcal{I}_2,\mathcal{I}_1}$ is a valid probability transition matrix; b) the Markov chain $\widehat{\boldsymbol{p}}^t = \widehat{\boldsymbol{p}}^{t-1}\widehat{\boldsymbol{R}}$ is such that its stationary distribution $\widehat{\boldsymbol{p}}^\infty \propto \boldsymbol{p}_{\mathcal{I}_1}^\infty$.*

Let us first provide an insight into our technical result. In order to so consider the Markov chain evolution:

$$\boldsymbol{p}_{\mathcal{I}_1}^t = \boldsymbol{p}_{\mathcal{I}_1}^{t-1}\boldsymbol{R}_{\mathcal{I}_1,\mathcal{I}_1} + \boldsymbol{p}_{\mathcal{I}_2}^{t-1}\boldsymbol{R}_{\mathcal{I}_2,\mathcal{I}_1},$$
$$\boldsymbol{p}_{\mathcal{I}_2}^t = \boldsymbol{p}_{\mathcal{I}_1}^{t-1}\boldsymbol{R}_{\mathcal{I}_1,\mathcal{I}_2} + \boldsymbol{p}_{\mathcal{I}_2}^{t-1}\boldsymbol{R}_{\mathcal{I}_2,\mathcal{I}_2}.$$

Now assume that $\boldsymbol{p}_{\mathcal{I}_2}^t \approx \boldsymbol{p}_{\mathcal{I}_2}^\infty$ for all large enough $t$, which leads to

$$\boldsymbol{p}_{\mathcal{I}_1}^t = \boldsymbol{p}_{\mathcal{I}_1}^{t-1}\boldsymbol{R}_{\mathcal{I}_1,\mathcal{I}_1} + \boldsymbol{p}_{\mathcal{I}_2}^\infty\boldsymbol{R}_{\mathcal{I}_2,\mathcal{I}_1},$$
$$\boldsymbol{p}_{\mathcal{I}_2}^\infty \approx \boldsymbol{p}_{\mathcal{I}_1}^{t-1}\boldsymbol{R}_{\mathcal{I}_1,\mathcal{I}_2} + \boldsymbol{p}_{\mathcal{I}_2}^\infty\boldsymbol{R}_{\mathcal{I}_2,\mathcal{I}_2}.$$

If $\boldsymbol{p}_{\mathcal{I}_2}^\infty$ is small enough, then the Markov chain will rarely end up in the states with indexes $\mathcal{I}_2$. Meaning that we can remove these states, but we would need to take them into account while computing the new probability transitions. Furthermore, we need to do so while obtaining a new Markov chain in the process. The solution to this question is straightforward, i.e., we can solve for $\boldsymbol{p}_{\mathcal{I}_2}^\infty$ to obtain:

$$\boldsymbol{p}_{\mathcal{I}_1}^t = \boldsymbol{p}_{\mathcal{I}_1}^{t-1}\left(\boldsymbol{R}_{\mathcal{I}_1,\mathcal{I}_1} + \boldsymbol{R}_{\mathcal{I}_1,\mathcal{I}_2}(\boldsymbol{I} - \boldsymbol{R}_{\mathcal{I}_2,\mathcal{I}_2})^{-1}\boldsymbol{R}_{\mathcal{I}_2,\mathcal{I}_1}\right).$$

Remarkably, the resulting process is also a Markov chain! In order to verify this we need to prove the following:

a) the matrix $\boldsymbol{I} - \boldsymbol{R}_{\mathcal{I}_2,\mathcal{I}_2}$ is invertible ;

b) the matrix $(\boldsymbol{I} - \boldsymbol{R}_{\mathcal{I}_2,\mathcal{I}_2})^{-1}$ is nonnegative ;

c) the right eigenvector of the matrix $\boldsymbol{R}_{\mathcal{I}_1,\mathcal{I}_1} + \boldsymbol{R}_{\mathcal{I}_1,\mathcal{I}_2}(\boldsymbol{I} - \boldsymbol{R}_{\mathcal{I}_2,\mathcal{I}_2})^{-1}\boldsymbol{R}_{\mathcal{I}_2,\mathcal{I}_1}$ can be chosen to be the vector of ones $\boldsymbol{1}$.

We will need to develop some mathematical tools in order to prove these statements. For a matrix $\boldsymbol{A} \in \mathbb{R}^{n \times n}$ the spectral radius $r(\boldsymbol{A})$ is the maximum absolute value of the eigenvalues $\lambda_i$ of $\boldsymbol{A}$ and $r(\boldsymbol{A}) = \max_i |\lambda_i(\boldsymbol{A})|$. The matrix $\boldsymbol{A} \in \mathbb{R}^{n \times m}$ is called nonnegative if all its entries are nonnegative and denoted as $\boldsymbol{A} \geq 0$, if at least one element is positive we write $\boldsymbol{A} > 0$ and if all elements are positive we write $\boldsymbol{A} \gg 0$. The matrix $\boldsymbol{A} \in \mathbb{R}^{n \times n}$ is called reducible if there exists a permutation matrix $\boldsymbol{T}$ such that $\boldsymbol{T}\boldsymbol{A}\boldsymbol{T}^T = \begin{pmatrix} \boldsymbol{B} & \boldsymbol{C} \\ \boldsymbol{0} & \boldsymbol{D} \end{pmatrix}$ for some square matrices $\boldsymbol{B}$ and $\boldsymbol{D}$. The matrix is called irreducible if such a permutation matrix does not exist. The matrix $\boldsymbol{A} \in \mathbb{R}^{n \times n}$ is called M-matrix, if all its off-diagonal entries are nonpositive and it can be represented as $\boldsymbol{A} = s\boldsymbol{I} - \boldsymbol{B}$ with $s \geq r(\boldsymbol{B})$ (Berman & Plemmons, 1994). Interestingly the inverse of an M-matrix is always nonnegative (the opposite is generally false) (Berman & Plemmons, 1994). We will also use the following results:

**Proposition A1 (Perron-Frobenius Theorem)** *Let $\boldsymbol{A} \in \mathbb{R}^{n \times n}$ be a nonnegative matrix, then the spectral radius $r(\boldsymbol{A})$ is an eigenvalue of $\boldsymbol{A}$ and the corresponding left and right eigenvectors can be chosen to be nonnegative.*

*If $\boldsymbol{A}$ is additionally irreducible then $r(\boldsymbol{A})$ is a simple eigenvalue of $\boldsymbol{A}$ and the corresponding left and right eigenvectors can be chosen to be positive.*

**Proposition A2** *Consider $\boldsymbol{A}, \boldsymbol{B} \in \mathbb{R}^{n \times n}$ and let $\boldsymbol{A} > \boldsymbol{B} \geq 0$, then $r(\boldsymbol{A}) \geq r(\boldsymbol{B})$. If additionally $\boldsymbol{A}$ is irreducible then the inequality is strict $r(\boldsymbol{A}) > r(\boldsymbol{B})$.*

**Proof:** This result is well-known, but for completeness we show the proof here (we adapt a similar technique to Noutsos (2006)). Let $\boldsymbol{b}$ be the right nonnegative eigenvector of $\boldsymbol{B}$ corresponding to $r(\boldsymbol{B})$ and let $\boldsymbol{a}$ be the left nonnegative eigenvector of $\boldsymbol{A}$ corresponding to $r(\boldsymbol{A})$. Now we have

$$r(\boldsymbol{A})\boldsymbol{a}^T\boldsymbol{b} = \boldsymbol{a}^T\boldsymbol{A}\boldsymbol{b} \geq^{\text{since } \boldsymbol{A} > \boldsymbol{B}} \boldsymbol{a}^T\boldsymbol{B}\boldsymbol{b} = r(\boldsymbol{B})\boldsymbol{a}^T\boldsymbol{b}.$$

If $\boldsymbol{a}^T\boldsymbol{b}$ is positive, then the first part is shown. If $\boldsymbol{a}^T\boldsymbol{b} = 0$, then we can make a continuity argument by perturbing $\boldsymbol{A}$ and $\boldsymbol{B}$ to $\boldsymbol{A}'$ and $\boldsymbol{B}'$ in such a way that for their corresponding eigenvectors we have $(\boldsymbol{a}')^T\boldsymbol{b}' > 0$. Hence the first part of the proof is shown.

Now consider the case when $\boldsymbol{A}$ is irreducible. According to Proposition A1, since $\boldsymbol{A}$ is irreducible the spectral radius $r(\boldsymbol{A})$ is a simple eigenvalue, the corresponding eigenvector $\boldsymbol{a}$ can be chosen to be positive. Let us prove the second part by contradiction and assume that $r(\boldsymbol{A}) = r(\boldsymbol{B})$, which implies that $\boldsymbol{a}^T\boldsymbol{A}\boldsymbol{b} = \boldsymbol{a}^T\boldsymbol{B}\boldsymbol{b}$ and consequently $\boldsymbol{A}\boldsymbol{b} = \boldsymbol{B}\boldsymbol{b}$ since $\boldsymbol{A}\boldsymbol{b} \geq \boldsymbol{B}\boldsymbol{b}$ and $\boldsymbol{a} \gg 0$. Now since we also have $\boldsymbol{A} > \boldsymbol{B}$ this means that the vector $\boldsymbol{b}$ has at least one zero entry. Since we also have $\boldsymbol{A}\boldsymbol{b} = r(\boldsymbol{A})\boldsymbol{b}$, the vector $\boldsymbol{b}$ is an eigenvector of $\boldsymbol{A}$ corresponding to $r(\boldsymbol{A})$ and has zero entries by assumption above. This contradicts Perron-Frobenius theorem since the eigenspace corresponding to $r(\boldsymbol{A})$ is a ray (since $r(\boldsymbol{A})$ is a simple eigenvalue) and $\boldsymbol{b}$ does not lie in this eigenspace. Therefore, $r(\boldsymbol{A}) > r(\boldsymbol{B})$. $\square$

Now we proceed with the proof. We note that the stationary distribution (a positive normalized left eigenvector of the transition matrix) is unique if the Markov chain is irreducible (i.e., the transition matrix is irreducible) due to Proposition A1. It is also almost surely true if the transition model is learned. This is because the subset of reducible matrices is measure zero in the set of nonnegative matrices.

Recall that we have a Markov chain $\boldsymbol{p}^t = \boldsymbol{p}^{t-1}\boldsymbol{R}$ with a stationary distribution $\boldsymbol{p}^\infty$, that is $\boldsymbol{p}^\infty = \boldsymbol{p}^\infty\boldsymbol{R}$. Consider the index sets: the distilled context indexes $\mathcal{I}_1 = \{i|\boldsymbol{p}_i^\infty \geq \varepsilon_{\text{distil}}\}$ and the spurious context indexes $\mathcal{I}_2 = \{i|\boldsymbol{p}_i^\infty < \varepsilon_{\text{distil}}\}$.

First, we will show that the matrix $\widehat{\boldsymbol{R}} = \boldsymbol{R}_{\mathcal{I}_1,\mathcal{I}_1} + \boldsymbol{R}_{\mathcal{I}_1,\mathcal{I}_2}(\boldsymbol{I} - \boldsymbol{R}_{\mathcal{I}_2,\mathcal{I}_2})^{-1}\boldsymbol{R}_{\mathcal{I}_2,\mathcal{I}_1}$ is nonnegative. Since the matrix $\boldsymbol{R}_{\mathcal{I}_2,\mathcal{I}_2}$ is a nonnegative submatrix of $\boldsymbol{R}$, due to Proposition A2 we have that $r(\boldsymbol{R}_{\mathcal{I}_2,\mathcal{I}_2}) < r(\boldsymbol{R}) = 1$. This means that $\boldsymbol{I} - \boldsymbol{R}_{\mathcal{I}_2,\mathcal{I}_2}$ is an M-matrix, which implies that it is invertible and its inverse is a nonnegative matrix. Hence $\widehat{\boldsymbol{R}}$ is nonnegative (Berman & Plemmons, 1994).

Since $\boldsymbol{R}$ describes a Markov chain we have $\boldsymbol{R}\mathbf{1} = \mathbf{1}$ (where $\mathbf{1}$ is the vector of ones) and $\boldsymbol{p}^\infty\boldsymbol{R} = \boldsymbol{p}^\infty$

$$\mathbf{1}_{\mathcal{I}_1} = \boldsymbol{R}_{\mathcal{I}_1,\mathcal{I}_1}\mathbf{1}_{\mathcal{I}_1}^T + \boldsymbol{R}_{\mathcal{I}_1,\mathcal{I}_2}\mathbf{1}_{\mathcal{I}_2}^T, \tag{A2a}$$

$$\mathbf{1}_{\mathcal{I}_2} = \boldsymbol{R}_{\mathcal{I}_2,\mathcal{I}_1}\mathbf{1}_{\mathcal{I}_1}^T + \boldsymbol{R}_{\mathcal{I}_2,\mathcal{I}_2}\mathbf{1}_{\mathcal{I}_2}^T, \tag{A2b}$$

$$\boldsymbol{p}_{\mathcal{I}_1}^\infty = \boldsymbol{p}_{\mathcal{I}_1}^\infty\boldsymbol{R}_{\mathcal{I}_1,\mathcal{I}_1} + \boldsymbol{p}_{\mathcal{I}_2}^\infty\boldsymbol{R}_{\mathcal{I}_2,\mathcal{I}_1}, \tag{A2c}$$

$$\boldsymbol{p}_{\mathcal{I}_2}^\infty = \boldsymbol{p}_{\mathcal{I}_1}^\infty\boldsymbol{R}_{\mathcal{I}_1,\mathcal{I}_2} + \boldsymbol{p}_{\mathcal{I}_2}^\infty\boldsymbol{R}_{\mathcal{I}_2,\mathcal{I}_2}. \tag{A2d}$$

Now using elementary algebra we can establish that $\widehat{\boldsymbol{R}}\mathbf{1}_{\mathcal{I}_1} = \mathbf{1}_{\mathcal{I}_1}$:

$$\widehat{\boldsymbol{R}}\mathbf{1}_{\mathcal{I}_1} = \left(\boldsymbol{R}_{\mathcal{I}_1,\mathcal{I}_1} + \boldsymbol{R}_{\mathcal{I}_1,\mathcal{I}_2}(\boldsymbol{I} - \boldsymbol{R}_{\mathcal{I}_2,\mathcal{I}_2})^{-1}\boldsymbol{R}_{\mathcal{I}_2,\mathcal{I}_1}\right)\mathbf{1}_{\mathcal{I}_1} = $$
$$\boldsymbol{R}_{\mathcal{I}_1,\mathcal{I}_1}\mathbf{1}_{\mathcal{I}_1} + \boldsymbol{R}_{\mathcal{I}_1,\mathcal{I}_2}(\boldsymbol{I} - \boldsymbol{R}_{\mathcal{I}_2,\mathcal{I}_2})^{-1}\boldsymbol{R}_{\mathcal{I}_2,\mathcal{I}_1}\mathbf{1}_{\mathcal{I}_1} =^{\text{due to (A2b)}}$$
$$\boldsymbol{R}_{\mathcal{I}_1,\mathcal{I}_1}\mathbf{1}_{\mathcal{I}_1} + \boldsymbol{R}_{\mathcal{I}_1,\mathcal{I}_2}\mathbf{1}_{\mathcal{I}_2} =^{\text{due to (A2a)}} \mathbf{1}_{\mathcal{I}_1}.$$

Similarly for $\boldsymbol{p}_{\mathcal{I}_1}^\infty\widehat{\boldsymbol{R}} = \boldsymbol{p}_{\mathcal{I}_1}^\infty$ we have:

$$\boldsymbol{p}_{\mathcal{I}_1}^\infty\widehat{\boldsymbol{R}} = \boldsymbol{p}_{\mathcal{I}_1}^\infty\left(\boldsymbol{R}_{\mathcal{I}_1,\mathcal{I}_1} + \boldsymbol{R}_{\mathcal{I}_1,\mathcal{I}_2}(\boldsymbol{I} - \boldsymbol{R}_{\mathcal{I}_2,\mathcal{I}_2})^{-1}\boldsymbol{R}_{\mathcal{I}_2,\mathcal{I}_1}\right) = $$
$$\boldsymbol{p}_{\mathcal{I}_1}^\infty\boldsymbol{R}_{\mathcal{I}_1,\mathcal{I}_1} + \boldsymbol{p}_{\mathcal{I}_1}^\infty\boldsymbol{R}_{\mathcal{I}_1,\mathcal{I}_2}(\boldsymbol{I} - \boldsymbol{R}_{\mathcal{I}_2,\mathcal{I}_2})^{-1}\boldsymbol{R}_{\mathcal{I}_2,\mathcal{I}_1} =^{\text{due to (A2d)}}$$
$$\boldsymbol{p}_{\mathcal{I}_1}^\infty\boldsymbol{R}_{\mathcal{I}_1,\mathcal{I}_1} + \boldsymbol{p}_{\mathcal{I}_2}^\infty\boldsymbol{R}_{\mathcal{I}_1,\mathcal{I}_2} =^{\text{due to (A2c)}} \boldsymbol{p}_{\mathcal{I}_1}^\infty.$$

Since $\boldsymbol{p}_{\mathcal{I}_1}^\infty\widehat{\boldsymbol{R}} = \boldsymbol{p}_{\mathcal{I}_1}^\infty$, the normalized vector $\boldsymbol{p}_{\mathcal{I}_1}^\infty$ is the stationary distribution of the distilled Markov chain and hence $\widehat{\boldsymbol{p}}^\infty \propto \boldsymbol{p}_{\mathcal{I}_1}^\infty$. This completes the proof.

## B.2 DERIVATION OF DYNAMIC PROGRAMMING PRINCIPLE AND PROOF OF THEOREM 2

For completeness we reproduce the theorem formulation below

**Theorem A2** *a) The belief of $z$ can be computed as $p(z_{t+1}|I_t^C) = b_{t+1}^z$, where $(b_{t+1}^z)_i \propto N_i = \sum_j p(s_{t+1}|s_t, \theta_i, a_t)\rho_{ji}(b_t^z)_j$, where $(b_t^z)_i$ are the entries of $b_t^z$; b) the optimal policy can be computed as $\pi(s, b^z) = \arg\max_a Q(s, b^z, a)$, where the value function satisfies the dynamic programming principle $Q(s_t, b_t^z, a_t) = r(s_t, b_t^z, a_t) + \gamma \int \sum_i N_i \max_{a_{t+1}} Q(s_{t+1}, b_{t+1}^z, a_{t+1}) \, ds_{t+1}$.*

The following definitions and derivations are in line with previous work by Hauskrecht (2000); Porta et al. (2006). We introduce the complete information state at the time $t$ as follows:

$$I_t^C = \{b_0, o_{\leq t}, a_{<t}\},$$

where $b_0 = p(z_0)\delta_{o_0}(s_0)$. In order to tackle the intractability of the complete information state, one can use any information state that is sufficient in some sense:

**Definition A1** *Consider a partially observable Markov decision process $\{\mathcal{X}, \mathcal{A}, \mathcal{O}, \mathcal{P}_\mathcal{X}, \mathcal{P}_\mathcal{O}, \mathcal{R}, b_0\}$. Let $\mathcal{I}$ be an information state space and $\xi : \mathcal{I} \times \mathcal{A} \times \mathcal{O} \to \mathcal{I}$ be an update function defining an information process $I_t = \xi(I_{t-1}, a_{t-1}, o_t)$. We say that $I_t^S$ is a sufficient information process with regard to the optimal control if it is an information process and for any time step $t$, it satisfies*

$$p(x_t|I_t^S) = p(x_t|I_t^C),$$
$$p(o_t|I_{t-1}^S, a_{t-1}) = p(o_t|I_{t-1}^C, a_{t-1}).$$

As a sufficient information state of the process $I_t^S$, we will use the belief $b_t$ defined as follows:

$$b_t = p(x_t|I_t^C) = p(x_t|o_{\leq t}, a_{<t}) = p(s_t|o_{\leq t}, a_{<t})p(z_t|o_{<t}, a_{<t}).$$

Effectively, we introduce the belief $b_t^s = p(s_t|o_{\leq t}, a_{<t})$ of the state $s_t$ and the belief $b_t^z = p(z_t|o_{\leq t}, a_{<t})$ of the state $z_t$ given an observation $o_t$. However, since $o_t = s_t$ we can consider only the belief of $z_t$:

$$b_t^s = p(s_t|I_t^C) = p(s_t|o_{\leq t}, a_{<t}) = \delta_{s_t}(o_t),$$
$$b_t^z = p(z_t|I_t^C) = p(z_t|s_{\leq t}, a_{<t}).$$

One of the features of our model is that the updates of the beliefs can be analytically computed.

**Lemma A1** *The belief $b_t^z = p(z_t|I_t^C)$ is a sufficient state information with the updates:*

$$(b_{t+1}^z)_i \propto N_i = \sum_j p(s_{t+1}|s_t, z_{t+1} = i, a_t, I_t^C)\rho_{ji}(b_t^z)_j.$$

**Proof:** First, let us check that the belief $b_t$ satisfies both conditions of Definition A1. By definition of our belief we have $p(x_t|I_{t-1}^C) = b_t = p(x_t|I_{t-1}^S)$, which satisfies the first condition, as for the second condition we have:

$$p(o_t|I_{t-1}^C, a_{t-1}) = \int_{x_{t-1}\in\mathcal{X}} p(o_t|x_{t-1}, I_{t-1}^C, a_{t-1})p(x_{t-1}|I_{t-1}^C)dx_{t-1} =$$

$$\int_{x_{t-1}\in\mathcal{X}} p(o_t|x_{t-1}, a_{t-1})b_{t-1}dx_{t-1} = p(o_t|I_{t-1}^S, a_{t-1}).$$

Now straightforward derivations yield:

$$b_{t+1}^z = p(z_{t+1}|I_t^C, s_{t+1}, a_t) \propto p(z_{t+1}, s_{t+1}|I_t^C, a_t) =$$

$$p(s_{t+1}|z_{t+1}, I_t^C, a_t)\sum_{z_t} p(z_{t+1}|z_t)p(z_t|I_t^C) = p(s_{t+1}|s_t, z_{t+1}, a_t)\sum_{z_t} p(z_{t+1}|z_t)b_t^z,$$

completing the proof.  $\square$

As discussed by Porta et al. (2006), POMDP in continuous state, action and observation spaces also satisfy Bellman dynamic programming principle, albeit in a different space. Recall that the control problem is typically formulated as

$$J = \mathbb{E}_{\tilde{\tau}} \sum_{t=0}^{T} \gamma^t \boldsymbol{r}_t,$$

where $\tilde{\tau} = \{\boldsymbol{x}_0, \boldsymbol{a}_0, \dots, \boldsymbol{x}_{T-1}, \boldsymbol{a}_{T-1}, \boldsymbol{x}_T\}$ and the horizon $T$ can be infinite. As we do not have access to the state transitions we need to rewrite the problem in the observation or the belief spaces. We have

$$J = \mathbb{E}_{\boldsymbol{x}_0} \left\{ \sum_{t=0}^{T} \gamma^t \boldsymbol{r}_t(\boldsymbol{x}_t, \boldsymbol{a}_t) \big| \boldsymbol{x}_{t+1} \sim p(\boldsymbol{x}_{t+1} | \boldsymbol{x}_t, \boldsymbol{a}_t) \right\} =$$

$$\mathbb{E}_{\boldsymbol{x}_0} \left\{ \sum_{t=0}^{T} \gamma^t \tilde{\boldsymbol{r}}_t(\boldsymbol{b}_t, \boldsymbol{a}_t) \big| \boldsymbol{b}_t = \xi(\boldsymbol{b}_{t-1}, \boldsymbol{o}_t, \boldsymbol{a}_{t-1}) \right\} =$$

$$\mathbb{E}_{\tau} \left\{ \sum_{t=0}^{T} \gamma^t \tilde{\boldsymbol{r}}_t(\boldsymbol{b}_t, \boldsymbol{a}_t) \big| \boldsymbol{b}_t = \xi(\boldsymbol{b}_{t-1}, \boldsymbol{o}_t, \boldsymbol{a}_{t-1}) \right\},$$

where $\tau = \{\boldsymbol{b}_0, \boldsymbol{o}_0, \boldsymbol{a}_0, \dots, \boldsymbol{o}_{T-1}, \boldsymbol{a}_{T-1}, \boldsymbol{o}_T\}$ and:

$$\tilde{\boldsymbol{r}}_t(\boldsymbol{b}_t, \boldsymbol{a}_t) = \int \boldsymbol{r}_t(\boldsymbol{x}, \boldsymbol{a}_t) \boldsymbol{b}_t(\boldsymbol{x}) \, d\boldsymbol{x}.$$

Given this reparameterization we can introduce the value functions and derive the Bellman equation similarly to Porta et al. (2006), which can be written in terms of the Q-function as follows:

$$\boldsymbol{Q}(\boldsymbol{b}_t^s, \boldsymbol{b}_t^z, \boldsymbol{a}_t) = \int p_{\boldsymbol{r}}(\boldsymbol{r} | \boldsymbol{s}_t, \boldsymbol{a}_t) \boldsymbol{b}_t^s \, d\boldsymbol{s}_t +$$

$$\gamma \int \int p(\boldsymbol{o}_{t+1} | \boldsymbol{b}_t^s, \boldsymbol{b}_t^z, \boldsymbol{a}_t) \max_{\boldsymbol{a}_{t+1}} \boldsymbol{Q}(\boldsymbol{b}_{t+1}^s, \boldsymbol{b}_{t+1}^z, \boldsymbol{a}_{t+1}) d\boldsymbol{o}_{t+1}.$$

We, however, have an additional structure that allows for simplified value functions. First note that $\tilde{\boldsymbol{r}}_t(\boldsymbol{b}_t, \boldsymbol{a}_t) = \boldsymbol{r}_t(\boldsymbol{s}_t, \boldsymbol{b}_t^z, \boldsymbol{a}_t)$, i.e., our reward depends directly on the observation and the belief of $z$.

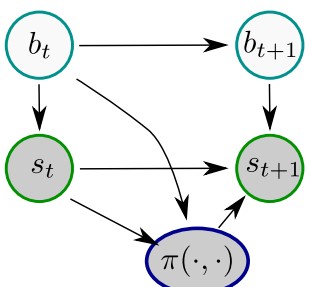

Now we need to estimate $\mathbb{E} \max_{\boldsymbol{a}_{t+1}} \boldsymbol{Q}(\boldsymbol{b}_{t+1}^s, \boldsymbol{b}_{t+1}^z, \boldsymbol{a}_{t+1})$, where the expectation is taken over new observation, the probability distribution of which can be computed as:

$$p(\boldsymbol{o}_{t+1} | \boldsymbol{b}_t^s, \boldsymbol{b}_t^z, \boldsymbol{a}_t) = p(\boldsymbol{s}_{t+1} | \boldsymbol{s}_t, \boldsymbol{b}_t^z, \boldsymbol{a}_t) =$$

$$\sum_j p(\boldsymbol{s}_{t+1} | \boldsymbol{s}_t, z_{t+1} = i, \boldsymbol{a}_t) \boldsymbol{\rho}_{ji}(\boldsymbol{b}_t^z)_j = \boldsymbol{N}_i.$$

This allows rewriting the second part of the Bellman equation as follows:

Figure A2: Graphical model for policy optimization

$$\int p(\boldsymbol{o}_{t+1} | \boldsymbol{b}_t^s, \boldsymbol{b}_t^z, \boldsymbol{a}_t) \max_{\boldsymbol{a}_{t+1}} \boldsymbol{Q}(\boldsymbol{b}_{t+1}^s, \boldsymbol{b}_{t+1}^z, \boldsymbol{a}_{t+1}) d\boldsymbol{o}_{t+1}$$

$$= \int \sum_i \boldsymbol{N}_i \max_{\boldsymbol{a}_{t+1}} \boldsymbol{Q}(\boldsymbol{s}_{t+1}, \boldsymbol{b}_{t+1}^z, \boldsymbol{a}_{t+1}) \, d\boldsymbol{s}_{t+1},$$

where $\boldsymbol{N}_i, \boldsymbol{b}_{t+1}^z$ depend on $\boldsymbol{s}_{t+1}$. Finally, we have:

$$\boldsymbol{Q}(\boldsymbol{s}_t, \boldsymbol{b}_t^z, \boldsymbol{a}_t) = \boldsymbol{r}(\boldsymbol{s}_t, \boldsymbol{b}_t^z, \boldsymbol{a}_t) + \gamma \int \sum_i \boldsymbol{N}_i \max_{\boldsymbol{a}_{t+1}} \boldsymbol{Q}(\boldsymbol{s}_{t+1}, \boldsymbol{b}_{t+1}^z, \boldsymbol{a}_{t+1}) \, d\boldsymbol{s}_{t+1},$$

which completes the proof.

### B.3 PERFORMANCE GAIN FOR OBSERVABLE CONTEXTS

It is not surprising that observing the ground truth of the contexts should improve the maximum expected return. In particular, even knowing the ground truth context model we can correctly estimate the context $z_{t+1}$ only *a posteriori*, i.e., after observing the next state $s_{t+1}$. This means that with every context switch we will mislabel a context with a high probability. This will lead to a sub-optimal action and performance loss, which the following result quantifies using the value functions.

**Theorem A3** *Assume we know the true transition model of the contexts and states and consider two settings: we observe the ground truth $z_t$ and we estimate it using $b_t^z$. Assume we computed the optimal model-based policy $\pi(\cdot|s_t, b_t^z)$ with the return $\mathcal{R}$ and the optimal ground-truth policy $\pi_{\mathrm{gt}}(\cdot|s_t, z_{t+1})$ with the corresponding optimal value functions $V_{\mathrm{gt}}(s, z)$ and $Q_{\mathrm{gt}}(s, z, a)$, then:*

$$\mathbb{E}_{z_1, s_0} V_{\mathrm{gt}}(s_0, z_1) - \mathcal{R} \geq \mathbb{E}_{\tau, a_{t_m}^{\mathrm{gt}} \sim \pi_{\mathrm{gt}}, a_{t_m} \sim \pi} \sum_{m=1}^{M} \gamma^{t_m} (Q(s_{t_m}, z_{t_m+1}, a_{t_m}^{\mathrm{gt}}) - Q(s_{t_m}, z_{t_m+1}, a_{t_m})),$$

*where $M$ is the number of misidentified context switches in a trajectory $\tau$.*

**Proof:**   Let us consider the best case scenario. As we assume that we have access to the ground truth model for computing the policy $\pi(\cdot|s_t, b_t^z)$, we can assume that the ground truth value of $z_t$ has the highest probability mass in the vector $b_t^z$. That is, we can assume that we can identify the correct context *a posteriori*. In effect, we can use *a priori* estimate of $z_{t+1}$ by transitioning to the next time step, i.e., using the vector $b_t^z R$. We can also assume that the action distributions of the policy $\pi$ and the ground truth policy $\pi_{\mathrm{gt}}$ are identical provided our *a priori* estimate of the context and the ground truth context are the same. This, however, is almost surely not true when the context switch occurs, as we need at least one sample from the transition model in the new context. Now if we can estimate the effect of this mismatch on the performance, this will provide us with a lower bound on the performance gain.

Let $V_{\mathrm{gt}}(s)$, $Q_{\mathrm{gt}}(s, a)$ be the optimal value functions for the ground truth policy $\pi_{\mathrm{gt}}$ satisfying the Bellman equation:

$$Q_{\mathrm{gt}}(s, a, z) = \mathbb{E}_{s' \sim p(\cdot|s, a, z), z' \sim \mathbf{Cat}(\rho_z)} \left( r(s, a, s') + V_{\mathrm{gt}}(s', z') \right),$$
$$V_{\mathrm{gt}}(s, z) = \mathbb{E}_{a \sim \pi_{\mathrm{gt}}(\cdot|s, z)} Q_{\mathrm{gt}}(s, z, a).$$

Consider a particular realization of the stochastic context variable $z_t$ (which is independent of $s_t$, $a_t$) and assume the context switched only once at $t_1$. Then we have

$$\mathcal{R}(s_0, z_1) = \mathbb{E}_{a_t \sim \pi(\cdot)} \sum_{t=0}^{T} \gamma^t r(s_t, a_t, s_{t+1}) = \mathbb{E}_{a_t, a_{t_1} \sim \pi} \left( \sum_{t=0}^{t_1-1} \gamma^t r(s_t, a_t, s_{t+1}) + \right.$$

$$+ \gamma^{t_1} r(s_{t_1}, a_{t_1}, s_{t_1+1}) + \gamma^{t_1+1} \sum_{t=t_1+1}^{T} \gamma^{t-t_1-1} r(s_t, a_t, s_{t+1}) \left. \right) =$$

$$V_{\mathrm{gt}}(s_0, z_1) - \gamma^{t_1} \mathbb{E}_{a \sim \pi_{\mathrm{gt}}, s_{t_1+1}, z_{t_1+2}} \left( r(s_{t_1}, a, s_{t_1+1}) + \gamma V_{\mathrm{gt}}(s_{t_1+1}, z_{t_1+2}) \right) +$$

$$+ \gamma^{t_1} \mathbb{E}_{a \sim \pi, s_{t_1+1}, z_{t_1+2}} \left( r(s_{t_1}, a, s_{t_1+1}) + \gamma V_{\mathrm{gt}}(s_{t_1+1}, z_{t_1+2}) \right) =$$

$$V_{\mathrm{gt}}(s_0, z_1) - \gamma^{t_1} \mathbb{E}_{a^{\mathrm{gt}} \sim \pi_{\mathrm{gt}}, a \sim \pi} (Q_{\mathrm{gt}}(s_{t_1}, z_{t_1+1}, a^{\mathrm{gt}}) - Q_{\mathrm{gt}}(s_{t_1}, z_{t_1+1}, a)).$$

In effect, we are using the $Q_{\mathrm{gt}}$ function to estimate the performance loss of one mistake. The same procedure can be repeated for context realizations with $M$ misidentified switches, where the number $M$ depends on the realization of the context variable

$$\mathcal{R}(s_0, z_1) =$$

$$V(s_0, z_1) - \sum_{m=1}^{M} \gamma^{t_m} \mathbb{E}_{a_{t_m}^{\mathrm{gt}} \sim \pi_{\mathrm{gt}}, a_{t_m} \sim \pi} (Q(s_{t_m}, z_{t_m+1}, a_{t_m}^{\mathrm{gt}}) - Q(s_{t_m}, z_{t_m+1}, a_{t_m})).$$

Now averaging over the context realizations proves the result. ☐

# C    ALGORITHM DETAILS

There are three main components in our algorithm: the generative model derivation (HDP-C-MDP), the model learning algorithm with probabilistic inference and the control algorithms. We firstly briefly comment on each on these components to give an overview of the results and then explain our main contributions to each.

In order to learn the model of the context transitions, we choose the Bayesian approach and we employ Hierarchical Dirichlet Processes (HDP) as priors for context transitions inspired by time-series modeling and analysis tools reported by Fox et al. (2008a;b). We improve the model by proposing a context spuriosity measure allowing for reconstruction of ground truth contexts. We then derive a model learning algorithm using probabilistic inference. Having a model, we can take off-the-shelf frameworks such as (Pineda et al., 2021), which can include a Model Predictive Control (MPC) approach using Cross-Entropy Minimization (CEM) (cf. Chua et al. (2018) and Appendix C.5), or a policy-gradient approach Soft-actor critic (cf. Haarnoja et al. (2018) and Appendix C.6), which are both well-suited for model-based reinforcement learning. While MPC can be directly applied to our model, for policy-based control we needed to derive the representation of the optimal policy and prove the dynamic programming principle for our C-MDP (see Theorem 2 in the main text and its proof in Appendix B.1). We summarize our model-based approach in Algorithm A1.

---

**Algorithm A1:** Learning to Control HDP-C-MDP

**Input:** $\varepsilon_{\mathrm{distill}}$ - distillation threshold, $N_{\mathrm{warm}}$ - number of trajectories for warm start, $N_{\mathrm{traj}}$ - number of newly collected trajectories per epoch, $N_{\mathrm{epochs}}$ - number of training epochs, AGENT - policy gradient or MPC agent

Initialize AGENT with RANDOM AGENT, $\mathcal{D} = \emptyset$;

**for** $i = 1, \ldots, N_{\mathrm{epochs}}$ **do**

    Sample $N_{\mathrm{traj}}$ ($N_{\mathrm{warm}}$ if $i = 1$) trajectories from the environment with AGENT;

    Set $\mathcal{D}_{\mathrm{new}} = \{(\boldsymbol{s}^i, \boldsymbol{a}^i)\}_{i=1}^{N_{\mathrm{traj}}}$, where $\boldsymbol{s}^i = \{\boldsymbol{s}_t^i\}_{t=-1}^T$ and $\boldsymbol{a}^i = \{\boldsymbol{a}_t^i\}_{t=-1}^T$ are the state and action sequences in the $i$-th trajectory. Set $\mathcal{D} = \mathcal{D} \cup \mathcal{D}_{\mathrm{new}}$;

    Update generative model parameters by gradient ascent on ELBO in Equation 4;

    Perform context distillation with $\varepsilon_{\mathrm{distill}}$;

    **if** AGENT *is* POLICY **then**

        Sample trajectories for policy update from $\mathcal{D}$;

        Recompute the beliefs using the model for these trajectories;

        Update policy parameters

    **end**

**end**

**return** AGENT

---

## C.1    SLIGHTLY MORE DETAILS ON THE HIERARCHICAL DIRICHLET PROCESSES

*A Dirichlet process (DP),* denoted as $\mathbf{DP}(\gamma, H)$, is characterized by a concentration parameter $\gamma$ and a base distribution $H(\lambda)$ defined over the parameter space $\Theta$. A sample $G$ from $\mathbf{DP}(\gamma, H)$ is a probability distribution satisfying $(G(A_1), ..., G(A_r)) \sim \mathbf{Dir}(\gamma H(A_1), ..., \gamma H(A_r))$ for every finite measurable partition $A_1, ..., A_r$ of $\Theta$, where $\mathbf{Dir}$ denotes the Dirichlet distribution. Sampling $G$ is often performed using the stick-breaking process (Sethuraman, 1994) and constructed by randomly mixing atoms independent and identically distributed samples $\boldsymbol{\theta}_k$ from $H$:

$$\nu_k \sim \mathbf{Beta}(1, \gamma), \quad \beta_k = \nu_k \prod_{i=1}^{k-1}(1 - \nu_i), \quad G = \sum_{k=1}^{\infty} \beta_k \delta_{\boldsymbol{\theta}_k}, \tag{A3}$$

where $\delta_{\boldsymbol{\theta}_k}$ is the Dirac distribution at $\boldsymbol{\theta}_k$, and the resulting distribution of $\boldsymbol{\beta} = (\beta_1, ...\beta_\infty)$ is called GEM$(\gamma)$ for Griffiths-Engen-McCloskey (Teh et al., 2006). The discrete nature of $G$ motivates the application of DP as a non-parametric prior for mixture models with an infinite number of atoms $\boldsymbol{\theta}_k$. We note that the stick-breaking procedure assigns progressively smaller values to $\beta_k$ for large $k$, thus encouraging a smaller number of meaningful atoms.

*The Hierarchical Dirichlet Process (HDP)* is a group of DPs sharing a base distribution, which itself is a sample from a DP: $G \sim \mathbf{DP}(\gamma, H)$, $G_j \sim \mathbf{DP}(\alpha, G)$ for all $j = 0, 1, 2, \ldots$ (Teh et al.,

2006). The distribution $G$ guarantees that all $G_j$ inherit the same set of atoms, i.e., atoms of $G$, while keeping the benefits of DPs in the distributions $G_j$. HDPs have received a significant attention in the literature (Teh et al., 2006; Fox et al., 2008b;a) with various applications including Markov chain modeling.

In our case, the atoms $\{\boldsymbol{\theta}_k\}$ forming the context set $\widetilde{\mathcal{C}}$ are sampled from $H(\lambda)$. It can be shown that a random draw $G_j$ from $\mathbf{DP}(\alpha, G)$ can be done using $\widetilde{\boldsymbol{\rho}}_j \sim \mathrm{GEM}(\alpha)$ and $\widetilde{\boldsymbol{\theta}}_k \sim G$. However, since $\widetilde{\boldsymbol{\theta}}_k$ is sampled from $\widetilde{\mathcal{C}}$, $G_j$ is also a distribution over $\widetilde{\mathcal{C}}$ and

$$G_j = \sum_{k=0}^{\infty} \widetilde{\rho}_{jk} \delta_{\widetilde{\boldsymbol{\theta}}_k} = \sum_{k=0}^{\infty} \rho_{jk} \delta_{\boldsymbol{\theta}_k},$$

for some $\boldsymbol{\rho}_{jk}$, which can be sampled using another stick-break construction (Teh et al., 2006). We consider its modified version introduced by Fox et al. (2011):

$$\mu_{jk} \mid \alpha, \kappa, \beta \sim \mathbf{Beta}\left(\alpha\beta_k + \kappa\tilde{\delta}_{jk}, \; \alpha + \kappa - \left(\sum_{i=1}^{k} \alpha\beta_i + \kappa\tilde{\delta}_{ji}\right)\right), \;\; \rho_{jk} = \mu_{jk} \prod_{i=1}^{k-1}(1 - \mu_{ji}),$$
(A4)

where $k \geq 1$, $j \geq 0$, $\tilde{\delta}_{jk}$ is the Kronecker delta, the parameter $\kappa \geq 0$, called the sticky factor, modifies the transition matrix priors encouraging self-transitions. The sticky factor serves as another measure of regularization reducing the average number of transitions. Thus $\mathbf{DP}(\alpha, G)$ can serve as the prior for the initial context distribution $\boldsymbol{\rho}_0$ and each row $\boldsymbol{\rho}_j$ in the transition matrix $\boldsymbol{R}$.

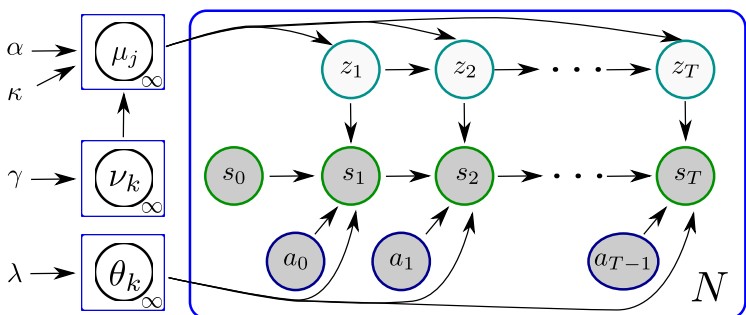

Figure A3: A probabilistic model for C-MDP with Markovian context

In summary, our probabilistic model is constructed in Equations A1,A3,A4 and illustrated in Figure A3 as a graphical model. We stress that the HDP in its stick-breaking construction assumes that $|\widetilde{\mathcal{C}}|$ is infinite and countable. In practice, however, we make an approximation and set $|\widetilde{\mathcal{C}}| = K$ with a large enough $K$.

## C.2 VARIATIONAL INFERENCE FOR PROBABILISTIC MODELING

Recall that our context MDP is represented as follows

$$z_{t+1} \mid z_t, \{\boldsymbol{\rho}_j\}_{j=1}^{\infty} \sim \mathrm{Mul}(\boldsymbol{\rho}_{z_t}), \quad z_0 \mid \rho_0 \sim \mathrm{Mul}(\boldsymbol{\rho}_0),$$
$$\boldsymbol{\theta}_k \mid \lambda \sim H(\lambda),$$
$$\boldsymbol{s}_t \mid \boldsymbol{s}_{t-1}, \boldsymbol{a}_{t-1}, z_t, \{\boldsymbol{\theta}_k\}_{k=1}^{\infty} \sim p(\boldsymbol{s}_t | \boldsymbol{s}_{t-1}, \boldsymbol{a}_{t-1}, \boldsymbol{\theta}_{z_t}),$$

and we depict our generative model as a graphical one in Figure A3. Also recall that the distributions $\boldsymbol{\rho}_j$ have the following priors:

$$\rho_{jk} = \mu_{jk} \prod_{i=1}^{k-1}(1 - \mu_{ji}), \quad \mu_{jk} \mid \alpha, \kappa, \beta \sim \mathbf{Beta}\left(\alpha\beta_k + \kappa\tilde{\delta}_{jk}, \alpha + \kappa - \left(\sum_{i=1}^{k} \alpha\beta_i + \kappa\tilde{\delta}_{ji}\right)\right),$$

$$\nu_k \mid \gamma \sim \mathbf{Beta}(1, \gamma), \qquad \beta_k = \nu_k \prod_{i=1}^{k-1}(1 - \nu_i),$$
(A5)

where $k \in \mathbb{N}^{\geq 1}, j \in \mathbb{N}^{\geq 0}, t \in \mathbb{N}^{\geq 0}$ and $\tilde{\delta}_{jk}$ is the Kronecker delta function.

We aim to find a variational distribution $q(\boldsymbol{\nu}, \boldsymbol{\mu}, \boldsymbol{\theta})$ to approximate the true posterior $p(\boldsymbol{\nu}, \boldsymbol{\mu}, \boldsymbol{\theta}|\mathcal{D})$, where $\mathcal{D} = \{(\boldsymbol{s}^i, \boldsymbol{a}^i)\}_{i=1}^N$ is a data set, $\boldsymbol{s}^i = \{\boldsymbol{s}_t^i\}_{t=-1}^T$ and $\boldsymbol{a}^i = \{\boldsymbol{a}_t^i\}_{t=-1}^T$ are the state and action sequences in the $i$-th trajectory. This is achieved by minimizing $\mathcal{KL}(q(\boldsymbol{\nu}, \boldsymbol{\mu}, \boldsymbol{\theta}) \,||\, p(\boldsymbol{\nu}, \boldsymbol{\mu}, \boldsymbol{\theta}|\mathcal{D}))$, or equivalently, maximizing the evidence lower bound (ELBO):

$$\text{ELBO} = \mathbb{E}_{q(\boldsymbol{\mu}, \boldsymbol{\theta})} \left[ \sum_{i=1}^N \log p(\boldsymbol{s}^i|\boldsymbol{a}^i, \boldsymbol{\mu}, \boldsymbol{\theta}) \right] - \mathcal{KL}(q(\boldsymbol{\nu}, \boldsymbol{\mu}, \boldsymbol{\theta}) \,||\, p(\boldsymbol{\nu}, \boldsymbol{\mu}, \boldsymbol{\theta})).$$

The variational distribution above involves infinite-dimensional random variables $\boldsymbol{\nu}, \boldsymbol{\mu}, \boldsymbol{\theta}$. To reach a tractable solution, we set $|\tilde{\mathcal{C}}| = K$ and exploit a mean-field truncated variational distribution (Blei et al., 2006; Hughes et al., 2015; Bryant & Sudderth, 2012). We construct the following variational distributions:

$$q(\boldsymbol{\nu}, \boldsymbol{\mu}, \boldsymbol{\theta}) = q(\boldsymbol{\nu})q(\boldsymbol{\mu})q(\boldsymbol{\theta}), \; q(\boldsymbol{\theta}|\hat{\boldsymbol{\theta}}) = \prod_{k=1}^K \delta(\boldsymbol{\theta}_k|\hat{\boldsymbol{\theta}}_k), \; q(\boldsymbol{\nu}|\hat{\boldsymbol{\nu}}) = \prod_{k=1}^{K-1} \delta(\nu_k|\hat{\nu}_k), \; q(\nu_K = 1) = 1,$$

$$q(\boldsymbol{\mu}|\hat{\boldsymbol{\mu}}) = \prod_{j=0}^K \prod_{k=1}^{K-1} \mathbf{Beta}\left(\mu_{jk}\middle|\hat{\mu}_{jk}, \hat{\mu}_j - \sum_{i=1}^k \hat{\mu}_{ji}\right), \quad q(\mu_{jK} = 1) = 1,$$

(A6)

where the hatted symbols represent free parameters. Random variables not shown in (A6) are conditionally independent of the data, and thus can be discarded from the problem.

We maximize ELBO using stochastic gradient ascent. In particular, given a sub-sampled batch $\mathcal{B} = \{(\boldsymbol{s}^i, \boldsymbol{a}^i)\}_{i=1}^B$, the gradient of ELBO is estimated as:

$$\nabla_{\hat{\boldsymbol{\nu}}, \hat{\boldsymbol{\mu}}, \hat{\boldsymbol{\theta}}} \text{ELBO} = \frac{N}{B} \sum_{i=1}^B \nabla_{\hat{\boldsymbol{\mu}}, \hat{\boldsymbol{\theta}}} \mathbb{E}_{q(\boldsymbol{\mu})} \left[ \log p(\boldsymbol{s}^i|\boldsymbol{a}^i, \boldsymbol{\mu}, \hat{\boldsymbol{\theta}}) \right] - \nabla_{\hat{\boldsymbol{\nu}}, \hat{\boldsymbol{\mu}}} \mathbb{E}_{q(\boldsymbol{\nu})}[\mathcal{KL}(q(\boldsymbol{\mu}) \,||\, p(\boldsymbol{\mu}|\boldsymbol{\nu}))]$$

$$+ \nabla_{\hat{\boldsymbol{\nu}}} \log p(\hat{\boldsymbol{\nu}}) + \nabla_{\hat{\boldsymbol{\theta}}} \log p(\hat{\boldsymbol{\theta}}),$$

where we apply implicit reparameterization method (Figurnov et al., 2018; Jankowiak & Obermeyer, 2018) for gradients with respect to the expectations over Beta distributions. For computing the gradient with respect to the likelihood term $\log p(\boldsymbol{s}^i|\boldsymbol{a}^i, \boldsymbol{\mu}, \hat{\boldsymbol{\theta}})$, we exploit a message passing algorithm to integrate out the context indexes $z_{1:T}^i$. We present the details of the gradient computations in what follows.

**Gradient of** $\log p(\boldsymbol{s}^i|\boldsymbol{a}^i, \boldsymbol{\mu}, \hat{\boldsymbol{\theta}})$**.** We drop the dependency on $\boldsymbol{\mu}, \hat{\boldsymbol{\theta}}$ in the following derivations. We have:

$$\nabla \log p(\boldsymbol{s}^i|\boldsymbol{a}^i) = \mathbb{E}_{p(\boldsymbol{z}^i|\boldsymbol{s}^i, \boldsymbol{a}^i)} \left[ \nabla \log p(\boldsymbol{s}^i|\boldsymbol{a}^i) \right] = \mathbb{E}_{p(\boldsymbol{z}^i|\boldsymbol{s}^i, \boldsymbol{a}^i)} \left[ \nabla \log \frac{p(\boldsymbol{s}^i, \boldsymbol{z}^i|\boldsymbol{a}^i)}{p(\boldsymbol{z}^i|\boldsymbol{s}^i, \boldsymbol{a}^i)} \right]$$

$$= \mathbb{E}_{p(\boldsymbol{z}^i|\boldsymbol{s}^i, \boldsymbol{a}^i)} \left[ \nabla \log p(\boldsymbol{s}^i, \boldsymbol{z}^i|\boldsymbol{a}^i) \right] - \mathbb{E}_{p(\boldsymbol{z}^i|\boldsymbol{s}^i, \boldsymbol{a}^i)} \left[ \nabla \log p(\boldsymbol{z}^i|\boldsymbol{s}^i, \boldsymbol{a}^i) \right],$$

where $\boldsymbol{z}^i = \{z_t^i\}_{t=1}^T$ is the context index sequence. Since the second term equals zero, we have:

$$\nabla \log p(\boldsymbol{s}^i|\boldsymbol{a}^i) = \mathbb{E}_{p(\boldsymbol{z}^i|\boldsymbol{s}^i, \boldsymbol{a}^i)}[\nabla \log p(\boldsymbol{s}^i, \boldsymbol{z}^i|\boldsymbol{a}^i)]$$

$$= \mathbb{E}_{p(z_1^i|\boldsymbol{s}^i, \boldsymbol{a}^i)}[\nabla \log p(\boldsymbol{s}_1^i|\boldsymbol{s}_0^i, \boldsymbol{a}_0^i, z_1^i)p(z_1^i)] +$$

$$+ \sum_{t=2}^T \mathbb{E}_{p(z_{t-1}^i, z_t^i|\boldsymbol{s}^i, \boldsymbol{a}^i)}[\nabla \log p(\boldsymbol{s}_t^i|\boldsymbol{s}_{t-1}^i, \boldsymbol{a}_{t-1}^i, z_t^i)p(z_t^i|z_{t-1}^i)].$$

Context index posteriors $p(z_0^i|\boldsymbol{s}^i, \boldsymbol{a}^i)$ and $p(z_{t-1}^i, z_t^i|\boldsymbol{s}^i, \boldsymbol{a}^i)$ required to compute the above expectation can be obtained by the message passing algorithm. The forward pass can be written as:

$$m_f(z_1^i) = p(z_1^i, \boldsymbol{s}_1^i|\boldsymbol{s}_0^i, \boldsymbol{a}_0^i) = p(\boldsymbol{s}_1^i|\boldsymbol{s}_0^i, \boldsymbol{a}_0^i, z_1^i)p(z_1^i)$$

$$m_f(z_t^i) = p(z_t^i, \boldsymbol{s}_{1:t}^i|\boldsymbol{s}_0^i, \boldsymbol{a}_{0:t-1}^i) = \sum_{z_{t-1}^i} p(z_t^i, z_{t-1}^i, \boldsymbol{s}_{1:t-1}^i, \boldsymbol{s}_t^i|\boldsymbol{s}_0^i, a_{0:t-1}^i)$$

$$= p(\boldsymbol{s}_t^i|\boldsymbol{s}_{t-1}^i, \boldsymbol{a}_{t-1}^i, z_t^i) \sum_{z_{t-1}^i} p(z_t^i|z_{t-1}^i)m_f(z_{t-1}^i).$$

The backward pass can be written as:

$$m_b(z_T^i) = 1$$

$$m_b(z_{t-1}^i) = p(\boldsymbol{s}_{t:T}|\boldsymbol{s}_{t-1}^i, \boldsymbol{a}_{t-1:T}^i, z_{t-1}^i) = \sum_{z_t^i} p(\boldsymbol{s}_t^i|\boldsymbol{s}_{t-1}^i, \boldsymbol{a}_{t-1}^i, z_t^i)p(z_t^i|z_{t-1}^i)m_b(z_t^i).$$

Combining the forward and backward messages, we have:

$$p(z_1^i|\boldsymbol{s}^i, \boldsymbol{a}^i) \propto p(z_1^i, \boldsymbol{s}_{1:T}^i|\boldsymbol{s}_0^i, \boldsymbol{a}^i) = m_f(z_1^i)m_b(z_1^i)$$

$$p(z_{t-1}^i, z_t^i|\boldsymbol{s}^i, \boldsymbol{a}^i) \propto p(z_{t-1}^i, z_t^i, \boldsymbol{s}_{1:t-1}^i, \boldsymbol{s}_t^i, \boldsymbol{s}_{t+1}^i|\boldsymbol{s}_0^i, \boldsymbol{a}^i) =$$

$$= m_f(z_{t-1}^i)p(z_t^i|z_{t-1}^i)p(\boldsymbol{s}_t^i|\boldsymbol{s}_{t-1}^i, \boldsymbol{a}_{t-1}^i, z_t^i)m_b(z_t^i).$$

The forward pass estimates the posterior context distribution at time $t$ give the past observations and actions (i.e., for $k \leq t$), which is similar to a filtering process. The backward pass estimates the context distribution at time $t$ give the future observations and actions (i.e., for $k \geq t$). Combining both passes allows to compute the context distribution at time $t$ given the whole trajectory.

**Gradient of ELBO**

$$\nabla_{\hat{\boldsymbol{\nu}}} \text{ELBO} = -\nabla_{\hat{\boldsymbol{\nu}}} \mathbb{E}_{q(\boldsymbol{\nu})}[\mathcal{KL}(q(\boldsymbol{\mu}) \,||\, p(\boldsymbol{\mu}|\boldsymbol{\nu}))] - \nabla_{\hat{\boldsymbol{\nu}}} \mathcal{KL}(q(\boldsymbol{\nu}) \,||\, p(\boldsymbol{\nu})),$$

$$\nabla_{\hat{\boldsymbol{\mu}}} \text{ELBO} = \frac{N}{B} \sum_{i=1}^B \nabla_{\hat{\boldsymbol{\mu}}} \mathbb{E}_{q(\boldsymbol{\mu})} \left[ \log p(\boldsymbol{s}^i|\boldsymbol{a}^i, \boldsymbol{\mu}, \hat{\boldsymbol{\theta}}) \right] - \mathbb{E}_{q(\boldsymbol{\nu})}[\nabla_{\hat{\boldsymbol{\mu}}}\mathcal{KL}(q(\boldsymbol{\mu}) \,||\, p(\boldsymbol{\mu}|\boldsymbol{\nu}))], \quad \text{(A7)}$$

$$\nabla_{\hat{\boldsymbol{\theta}}} \text{ELBO} = \frac{N}{B} \sum_{i=1}^B \mathbb{E}_{q(\boldsymbol{\mu})} \left[ \nabla_{\hat{\boldsymbol{\theta}}} \log p(\boldsymbol{s}^i|\boldsymbol{a}^i, \boldsymbol{\mu}, \hat{\boldsymbol{\theta}}) \right] + \nabla_{\hat{\boldsymbol{\theta}}} \log p(\hat{\boldsymbol{\theta}}),$$

where the terms in blue involve differentiating an expectation over Beta distributions, which we compute by adopting the implicit reparameterization (Figurnov et al., 2018; Jankowiak & Obermeyer, 2018). Considering a general case where $x \sim p_\phi(x)$ and the cumulative distribution function (CDF) of $p_\phi(x)$ is $F_\phi(x)$, it has been shown that:

$$\nabla_\phi \mathbb{E}_{p_\phi(x)}[f_\phi(x)] = \mathbb{E}_{p_\phi(x)}[\nabla_\phi f_\phi(x) + \nabla_x f_\phi(x)\nabla_\phi x], \quad \nabla_\phi x = -\frac{\nabla_\phi F_\phi(x)}{p_\phi(x)}.$$

### C.3 JUSTIFICATION FOR THE VARIATIONAL DISTRIBUTIONS

Here, we provide both intuitive and empirical justifications for the choice of variational distributions in (A6).

The mean-field approximation is mainly based on the tractability consideration where a reparameterizable variational distribution is required for gradient estimation (Blei et al., 2017). Besides, the truncation level is set to $K$, which reduces an infinite dimensional problem to a finite one.

The intuition of choosing the point estimation for $q(\boldsymbol{\nu})$ is the following: The $q(\boldsymbol{\mu})$ in (A6) induces a variational distribution $q(\boldsymbol{\rho})$, following $\rho_{jk} = \mu_{jk} \prod_{i=1}^{k-1}(1 - \mu_{ji})$ in (A5). When observing reasonable amount of trajectories, the optimal $q^*(\boldsymbol{\rho})$ should center around the ground-truth initial context distribution and the context transition. The HDP prior in our generative model specifies that $\boldsymbol{\rho}_j|\alpha, \boldsymbol{\beta} \sim \mathbf{DP}(\alpha, \boldsymbol{\beta})$ (Teh et al., 2006), which means $\boldsymbol{\beta}$ serves as the expectation of the initial context distribution and each row in the context transition. Intuitively, the optimal $q^*(\boldsymbol{\beta})$, which is induced from $q^*(\boldsymbol{\nu})$, should center around the stationary distribution of the context chain. Therefore, each factor $q^*(\nu_k)$ in the optimal $q^*(\boldsymbol{\nu})$ is supposed to be uni-modal, and thus a point estimation could be a reasonable simplification. This intuition was supported by our computing $\boldsymbol{\beta}$ and stationary distributions during training resulting in similar distilled Markov chains.

We then conduct a simple numerical study to verify the intuition above, which reveals another problem when using a full Beta variational distribution for $\boldsymbol{\nu}$. Consider the following context Markov chain:

$$\boldsymbol{\rho}_0 = \begin{pmatrix} 0.6 \\ 0.4 \end{pmatrix}, \qquad \boldsymbol{R} = \begin{pmatrix} \boldsymbol{\rho}_1^T \\ \boldsymbol{\rho}_2^T \end{pmatrix} = \begin{pmatrix} 0.7 & 0.3 \\ 0.2 & 0.8 \end{pmatrix}.$$

We set hyper-parameters in (A5) as $\gamma = \alpha = 1, \kappa = 0, K = 2$ and assume that, with sampled trajectories $\mathcal{D}$, the learned optimal $q^*(\boldsymbol{\mu})$ is given by:

$$q^*(\mu_{01})q^*(\mu_{11})q^*(\mu_{21}) = \mathbf{Beta}(\mu_{01}|13.8, 9.2)\,\mathbf{Beta}(\mu_{11}|14.0, 6.0)\,\mathbf{Beta}(\mu_{21}|3.0, 12.0).$$

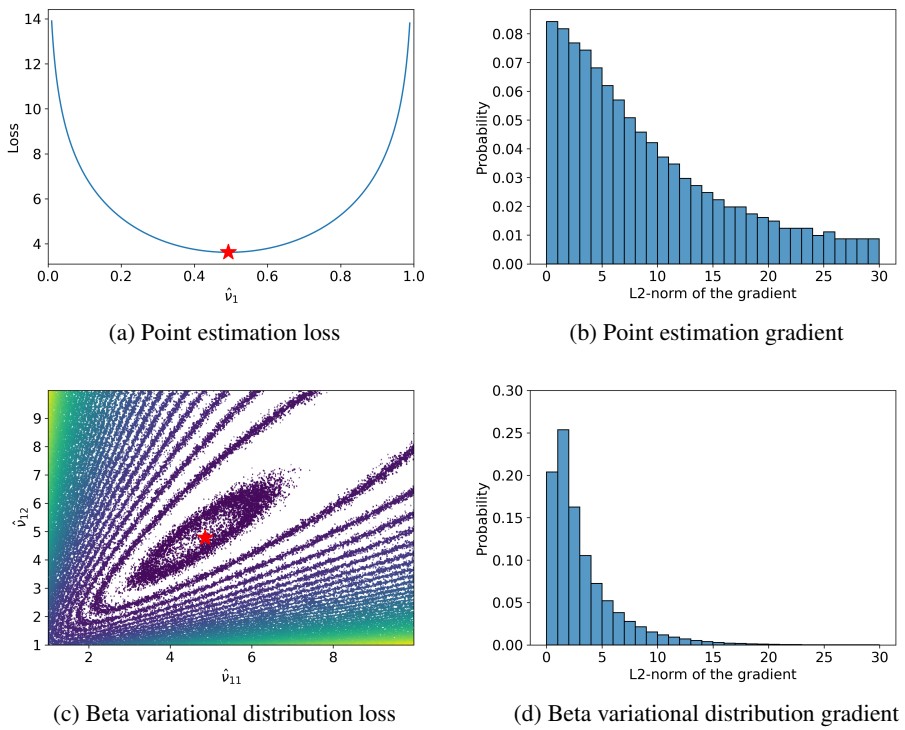

(a) Point estimation loss

(b) Point estimation gradient

(c) Beta variational distribution loss

(d) Beta variational distribution gradient

Figure A4: Loss function (A8) and the empirical distribution of its gradient under two configurations of $q(\boldsymbol{\nu})$

Recall that $\rho_{j1} = \mu_{j1}$ in (A5). The assumed $q^*(\mu_{jk})$ has the mean at its ground-truth value and the variance $0.01$. According to (A7), maximizing ELBO w.r.t $q(\boldsymbol{\nu})$ is equivalent to:

$$\min_{q(\boldsymbol{\nu})} \mathbb{E}_{q(\boldsymbol{\nu})}[\mathcal{KL}\left(q^*(\boldsymbol{\mu})\,||\,p(\boldsymbol{\mu}|\boldsymbol{\nu})\right)] + \nabla_{\hat{\nu}}\,\mathcal{KL}\left(q(\boldsymbol{\nu})\,||\,p(\boldsymbol{\nu})\right). \tag{A8}$$

where $q(\boldsymbol{\mu})$ is fixed to the assumed optima $q^*(\boldsymbol{\mu})$.

We investigate two configurations of $q(\boldsymbol{\nu})$: (1) a point estimation where $q(\boldsymbol{\nu}|\hat{\nu}) = \delta(\nu_1|\hat{\nu}_1)$; (2) a full Beta variational distribution where $q(\boldsymbol{\nu}|\hat{\boldsymbol{\nu}}) = \mathbf{Beta}(\nu_1|\hat{\nu}_{11}, \hat{\nu}_{12})$. The results are shown in Figure A4. The optimal point estimation is $q^*(\boldsymbol{\nu}) = \delta(\nu_1|0.493)$ (labeled by the red star in Figure A4(a)), while the optimal Beta distribution in Figure A4(c) is $q^*(\boldsymbol{\nu}) = \mathbf{Beta}(\nu_1|4.86, 4.78)$. The Beta optimum is uni-modal and its mode $0.505$ is very close to the point estimation $0.493$, which is consistent with our intuition. Comparing Figure A4(b) with A4(d), we observe that the point estimation can provide gradients better suited for optimization while using a Beta variational distribution potentially suffers from vanishing gradients. We observed this phenomenon in our model learning experiments as well.

Since the transition function $p(\boldsymbol{s}_t|\boldsymbol{s}_{t-1}, \boldsymbol{a}_{t-1}, \boldsymbol{\theta}_{z_t})$ is modeled by neural networks, it is generally hard to predict any property of the true posterior of $\boldsymbol{\theta}$ and choose an appropriate variational distribution. Bayesian neural network literature attempt to tackle this problem (Welling & Teh, 2011; Blundell et al., 2015; Kingma et al., 2015; Gal & Ghahramani, 2016; Ritter et al., 2018). However, most methods have considerably high computational complexity and it is not trivial to evaluate the quality of generated probabilistic prediction. In this work, we explicitly assume the distribution of $\boldsymbol{s}_t$ whose parameters are fitted by neural networks. The transition model is still capable of generating probabilistic prediction with a point estimation of $\boldsymbol{\theta}$. This assumption/simplification is followed by many model-based RL works.

### C.4 CONTEXT DISTILLATION

At every iteration of model learning, we can extract MAP parameter estimates $\{\boldsymbol{\theta}_k\}_{k=1}^K$ and approximated posteriors of $\boldsymbol{\rho}_0$ and $\boldsymbol{R} = [\boldsymbol{\rho}_1, ..., \boldsymbol{\rho}_K]$, which are induced from $q(\boldsymbol{\mu})$. Let us also define

---

**Algorithm A2:** Context distillation

---

**Inputs:** $\varepsilon_{\text{distil}}$ - distillation threshold; $\bar{R}$ - expected context transition matrix; $\hat{\beta}$ - weights of HDP's base distribution

Determine the distillation vector $v$ using one of the following choices:

(a) stationary distribution of the chain, $v$ such that $v = v\bar{R}$;

(b) weights of HDP's base distribution, $v = \hat{\beta}$.

Determine distilled context indexes $\mathcal{I}_1$ and spurious context indexes $\mathcal{I}_2$ as follows:

$\mathcal{I}_1 = \{i | v_i \geq \varepsilon_{\text{distil}}\}, \mathcal{I}_2 = \{i | v_i < \varepsilon_{\text{distil}}\}$;

Compute $\hat{R}$ as follows:

**if** AGENT *is* MPC **then**

$\quad \hat{R} = \bar{R}_{\mathcal{I}_1, \mathcal{I}_1} + \bar{R}_{\mathcal{I}_1, \mathcal{I}_2}(I - \bar{R}_{\mathcal{I}_2, \mathcal{I}_2})^{-1}\bar{R}_{\mathcal{I}_2, \mathcal{I}_1}$.

**end**

**else if** AGENT *is* POLICY **then**

$\quad \hat{R} = \begin{pmatrix} \bar{R}_{\mathcal{I}_1, \mathcal{I}_1} + \bar{R}_{\mathcal{I}_1, \mathcal{I}_2}(I - \bar{R}_{\mathcal{I}_2, \mathcal{I}_2})^{-1}\bar{R}_{\mathcal{I}_2, \mathcal{I}_1} & 0 \\ (I - \bar{R}_{\mathcal{I}_2, \mathcal{I}_2})^{-1}\bar{R}_{\mathcal{I}_2, \mathcal{I}_1} & 0 \end{pmatrix}$.

**end**

**return** $\hat{R}$ - distilled probability transition matrix.

---

the expected context initial distribution $\bar{\rho}_0 = \mathbb{E}_{q(\mu)}[\rho_0]$ and the expected context transition matrix $\bar{R} = \mathbb{E}_{q(\mu)}[R]$. These MAP estimates are used during training as well as testing for sampling the values $z$. Hence distilling $\bar{R}$ has an effect on training as well as testing.

Our distillation criterion is based on the values of the stationary distribution of the context Markov chain. Recall that one can compute the stationary distribution $\rho^\infty$ by solving $\rho^\infty = \rho^\infty \bar{R}$. Now the meaningful context indexes $\mathcal{I}_1 = \{i | \rho_i^\infty \geq \varepsilon_{\text{distil}}\}$ and spurious context indexes $\mathcal{I}_2 = \{i | \rho_i^\infty < \varepsilon_{\text{distil}}\}$ can be chosen using a distillation threshold $\varepsilon_{\text{distil}}$. Then, we distill the learned contexts by simply discarding $\hat{\theta}_{\mathcal{I}_2}$. Meanwhile, the context Markov chain also needs to be reduced. For $\bar{\rho}_0$, we gather those dimensions indexed by $\mathcal{I}_1$ into a new vector $\hat{\rho}_0$ and re-normalize $\hat{\rho}_0$. In addition, $\bar{R}$ can be reduced to $\hat{R}$ following the Theorem 1 in the main text.

Perhaps, a less rigorous, but definitely a simpler approach is choosing the index sets $\mathcal{I}_1$ and $\mathcal{I}_2$ using $\hat{\beta}$ — a MAP estimation of $\beta$ computed using $\hat{\nu}$. Since optimizing the $\mathcal{KL}(q(\mu) \| p(\mu | \hat{\nu}))]$ term in ELBO is essentially driving the posterior of $\rho_{0:K}$ toward $\hat{\beta}$. Therefore, $\hat{\beta}$ can be seen as a 'summary' distribution over contexts and we can consider the $k$-th context as a redundancy if $\hat{\beta}_k$ is small. It is not clear if $\hat{\beta}$ has a direct relation to the stationary distribution of the Markov chain with the transition probability $\hat{R}$. However, we have observed that the magnitudes of the entries of $\hat{\beta}$ and $p^\infty$ are correlated. Hence, in order to avoid computing an eigenvalue decomposition at every context estimation one can employ distillation using $\hat{\beta}$.

Both approaches are summarized in Algorithm A2. For policy optimization, we actually need to keep the number of contexts constant as dealing with changing state-belief space can be challenging during training. Therefore, the transition matrix $\hat{R}$ has the same dimensions as $\bar{R}$, where the transition probabilities between spurious contexts and from meaningful to spurious context are set to zero. We can still remove the spurious contexts after training both from the model and the policy.

## C.5 MPC USING CROSS-ENTROPY METHOD

This procedure is gradient-free, which benefits lower-dimensional settings, but could be less efficient in higher dimensional environments. It is also known to be more efficient than random shooting methods (Chua et al., 2018). The idea of this approach is quite simple and sketched in Algorithm A3. At the state $s_t$ with an action plan $\{a_k\}_{k=t}^{t+H}$, we sample plan updates $\{\delta_k^i\}_{k=t}^{t+H}$. We then roll-out trajectories and compute the average returns for each plan $\{a_k + \delta_k^i\}_{k=t}^{t+H}$. We pick $N_{\text{elite}}$ best performing action plans $\{\{a_k + \delta_k^{i_j}\}_{k=t}^{t+H}\}_{j=1}^{N_{\text{elite}}}$, compute the empirical mean $\tilde{\mu}_k$ and variance $\tilde{\Sigma}_k$ of the elite plan updates $\delta_k^{i_j}$, and then compute the updates on the action plan distribution as follows:

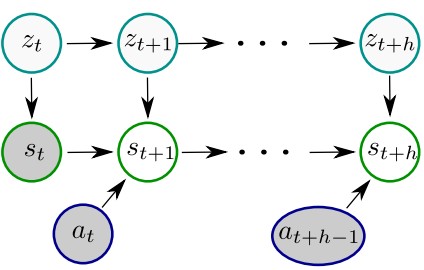

Figure A5: Graphical model for the MPC problem

$$\mu_k := (1 - l_r)\mu_k + l_r\tilde{\mu}_k,$$
$$\Sigma_k := (1 - l_r)\Sigma_k + l_r\tilde{\Sigma}_k, \tag{A9}$$

where $l_r$ is the learning rate.

---

**Algorithm A3:** MPC based on CEM

**Inputs:** $\{a_k\}_{k=t}^{t+H}$, $\{\mu_k\}_{k=t}^{t+H}$, $\{\Sigma_k\}_{k=t}^{t+H}$, $N_{\text{epochs}}$, $N_{\text{elite}}$, $N_{\text{traces}}$, $N_{\text{pops}}$, $l_r$, $s$, $H$
**for** $j = 0, \ldots, N_{\text{epochs}}$ **do**
 Sample action plan updates $\{\{\delta_k^i\}_{k=t}^{t+H}\}_{i=1}^{N_{\text{pops}}}$, where $\delta_k^i \sim \mathcal{N}(\mu_k, \Sigma_k)$;
 Roll-out $N_{\text{traces}}$ for each update plan;
 Compute the returns $1/N_{\text{traces}} \sum_{p=1}^{N_{\text{traces}}} \sum_{k=t}^{t+H} r(s_k^p, a_k + \delta_k^i)$ with $s_t^p = s$ for all $p$;
 Pick $N_{\text{elite}}$ best performing action plans;
 Update the sampling distributions $\{\mu_k\}_{k=t}^{t+H}$, $\{\Sigma_k\}_{k=t}^{t+H}$ as in (A9).
**end**

---

## C.6 SOFT-ACTOR CRITIC

We reproduce the summary of the soft-actor critic algorithm by Achiam (2018), which we found very accessible. The soft-actor critic algorithm aims at solving a modified RL problem with an entropy-regularized objective:

$$\pi = \arg\max_\pi \mathbb{E}_{\tau \sim \pi} \left[ \sum_{t=0}^{T} \gamma^t r(s_t, a_t) + \alpha H(\pi(\cdot|s_t)) \right],$$

where $H(P) = -\mathbb{E}_{x \sim P}[\log(P(x))]$ and $\alpha$ is called the temperature parameter. The entropy regularization modifies the Bellman equation for this problem as follows:

$$Q^\pi(s, a) = \mathbb{E}_{a' \sim \pi, s' \sim p(\cdot|s,a)}[r(s, a) + \gamma(Q^\pi(s', a') - \alpha \log \pi(a'|s'))].$$

The algorithm largely follows the standard actor-critic framework for updating value functions and policy, with a few notable changes. First, two Q functions are used in order to avoid overestimation of the value functions . In particular, the loss for value learning is as follows:

$$L_{\text{value,i}}(\phi_i, \mathcal{D}) = \mathbb{E}_{(s,a,r,s',d) \sim \mathcal{D}}\left[(Q_{\phi_i}(s, a) - y(r, s', d))^2\right], \tag{A10}$$

$$y(r, s', d) = r + \gamma(1 - d)\left(\min_{j=1,2} Q_{\phi_{\text{targ,j}}}(s', a') - \alpha \log \pi_\psi(a'|s')\right). \tag{A11}$$

For policy updates the reparameterization trick is used allowing for differentiation of the policy. Namely, the policy loss function is as follows:

$$L_{\text{policy}}(\psi, \mathcal{D}) = -\mathbb{E}_{s \sim D, \xi \sim \mathcal{N}(0,I)} \min_{j=1,2} Q_{\phi_j}(s', \tilde{a}_\psi) - \alpha \log \pi_\psi(\tilde{a}_\psi|s'), \tag{A12}$$

$$\tilde{a}_\psi = \tanh(\mu_\psi + \sigma_\psi \odot \xi), \quad \xi \sim \mathcal{N}(0, I). \tag{A13}$$

---

**Algorithm A4:** Soft-actor critic (basic version)

---

**Inputs:** $N_{\text{epochs}}$ - number of epochs, $N_{\text{upd}}$ - number of gradient updates per epochs,
  $N_{\text{target-freq}}$ - target value function update frequency, $N_{\text{samples}}$ - number of steps per epoch, $l_r$,
  $w$ - learning rates
$N_{\text{total-upd}} = 0$.
Initialize parameters $\boldsymbol{\psi}, \boldsymbol{\phi}_i, \boldsymbol{\phi}_{\text{target,i}} = \boldsymbol{\phi}_i$;
**for** $j = 0, \ldots, N_{\text{epochs}}$ **do**
  Sample $N_{\text{samples}}$ steps from the environment with $\boldsymbol{a} \sim \pi_{\boldsymbol{\psi}}(\cdot|\boldsymbol{s})$ resulting in the buffer update
    $\mathcal{D}_{\text{new}} = \{(\boldsymbol{s}_i, \boldsymbol{a}_i, \boldsymbol{s}'_i, \boldsymbol{r}_i, \boldsymbol{d}_i)\}_{i=1}^{N_{\text{samples}}}$;
  Set $\mathcal{D} = \mathcal{D}_{\text{new}} \cup \mathcal{D}$;
  Sample a batch $\mathcal{B}$ from the buffer $\mathcal{D}$;
  **for** $k = 0, \ldots, N_{\text{upd}}$ **do**
    $N_{\text{total-upd}} \leftarrow N_{\text{total-upd}} + 1$;
    Update parameters of the value functions $\boldsymbol{\phi}_i \leftarrow \boldsymbol{\phi}_i - l_r \nabla_{\boldsymbol{\phi}_i} L_{\text{value,i}}(\boldsymbol{\phi}_i, \mathcal{B})$;
    Update parameters of the policy $\boldsymbol{\psi} \leftarrow \boldsymbol{\psi} - l_r \nabla_{\boldsymbol{\psi}} L_{\text{policy}}(\boldsymbol{\psi}, \mathcal{B})$;
    **if** $\text{mod}(N_{\text{total-upd}}, N_{\text{target-freq}}) = 0$ **then**
      | Update parameters of the target value function $\boldsymbol{\phi}_{\text{target,i}} \leftarrow w\boldsymbol{\phi}_{\text{target,i}} + (1-w)\boldsymbol{\phi}_i$;
    **end**
  **end**
**end**
**return** $\pi_{\boldsymbol{\psi}}$

---

# D   EXPERIMENT DETAILS

## D.1   LEARNING ALGORITHMS

**Model learning**   We implemented the model learning using the package Pyro (Bingham et al., 2018), which is designed for efficient probabilistic programming. Pyro allows for automatic differentiation, i.e., we do not need to explicitly implement message passing and reparametrized gradients for the ELBO gradient computation. We still need, however, a forward message pass to compute the belief estimate, e.g., to perform filtering on the variable $z_t$ when needed.

**PPO with an RNN model**   We modified an implementation of PPO by Kostrikov (2018) to account for our belief model. In our implementation, the RNN with a hidden state $\boldsymbol{h}$ at time $t$ is taking the inputs $\boldsymbol{h}_{t-1}, \boldsymbol{s}_{t-1}, \boldsymbol{a}_{t-1}$, while producing the output $\boldsymbol{h}_t$. What is left is to project the hidden state onto the belief space using a decoder, which we have chosen as $\widehat{\boldsymbol{b}}_t = \text{softmax}(\boldsymbol{W}\boldsymbol{h}_t)$, where the length of the vector $\widehat{\boldsymbol{b}}_t$ is equal to the number of contexts. The architecture is depicted in Figure A6(a). Note that one can see the RNN and the decoder architecture as a model for the sufficient information state for the POMDP. We have experimented with different architectures, e.g., projecting to a larger belief space to account for spurious contexts, removing the decoder altogether etc. These architectures, however, did not yield reasonable results.

**GPMM.**   We took the implementation by Xu et al. (2020), which is able to swing-up the pole attached to the cart and adapt to environments with different parameters.

**SAC.**   We based our implementation largely on (Tandon, 2018) with some inspiration from (Yarats & Kostrikov, 2020). We use two architectures: full information policy (see Figure A6(c)) and model-based policy (see Figure A6(b)). The full information policy is using one hot encoded true context and is, therefore, used as a reference for the best case performance only.

**CEM-MPC.**   We implemented the algorithm from scratch in PyTorch.

## D.2   ENVIRONMENTS AND THEIR MODELS

**Cart-Pole Swing Up**   We largely followed the setting introduced by Xu et al. (2020), that is we set the maximum force magnitude to 20, time interval to 0.04, and time horizon to 100. We took

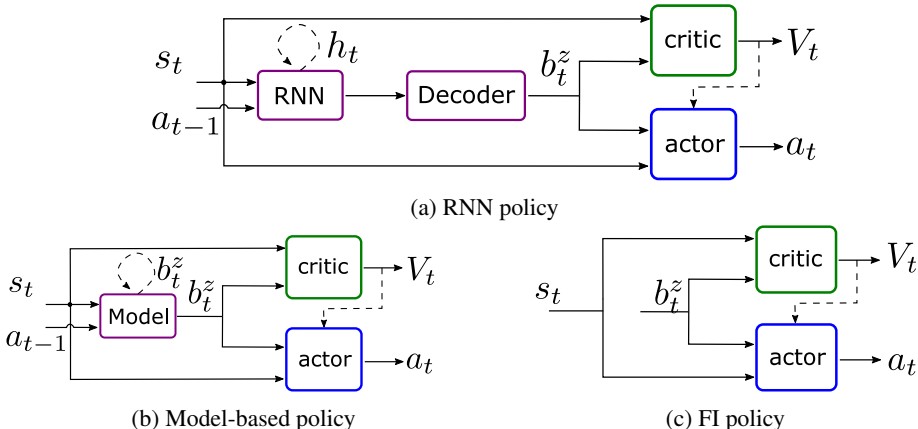

(a) RNN policy

(b) Model-based policy

(c) FI policy

Figure A6: Policy architectures

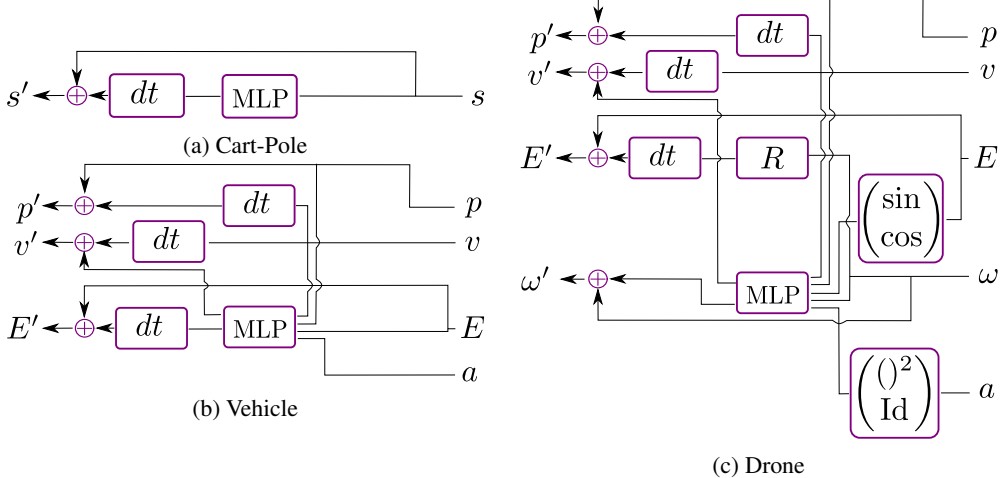

(a) Cart-Pole

(b) Vehicle

(c) Drone

Figure A7: Structure of the neural networks predicting the mean of the transition probability. The blocks $R$ and $dt$ stand for multiplication with the rotation matrix $R$ and discretization time $dt$. MLP denotes a multi-layer perceptron. $E$ stands for Eurler angles (pitch, roll, yaw), $p$, $v$, and $w$ stands for position, velocity and angular velocity in the world frame. Operator $\cdot'$ stands for the next time step

the implementation by Lovatto (2019) and modified it to fit our context MDP setting. The states of the environment are the position of the mass ($p$), velocity of the mass ($v$), deviation angle of the pole from the top position ($\theta$) and angular velocity ($\dot{\theta}$). For GPMM we replaced $\theta$ with $\sin(\theta)$ and $\cos(\theta)$ as was done by Xu et al. (2020). We set the reward function to $\cos(\theta)$, where $\theta$ is the deviation from the top position. Our transition model predicts the mean change between the next and the current states and its variance as it is common in model-based RL. Therefore, the structure of our neural network model for mean prediction is $s' = s + \mathrm{MLP}(s, a)$, where $s'$ is the successor state and MLP is a multi-layer perceptron predicting $s' - s$. The variance in the transition model is a trained parameter. The structure of the neural network predicting the mean of the transition model is depicted in Figure A7(a).

**Drone Take-off** The drone environment (Panerati et al., 2021) has 12 states: position in the world frame ($p_x$, $p_y$ and $p_z$), yaw, pitch, roll angles ($\psi$, $\theta$ and $\phi$), velocities in the world frame ($v_x$, $v_y$ and $v_z$) and angular velocities in the world frame ($\omega_x$, $\omega_y$ and $\omega_z$). The prediction of the transition model is similar to the cart-pole model with one notable exception: the neural network for the mean prediction has additional structure. Note that we can estimate spacial positions, roll, pitch and yaw

angles given position and angular velocities using crude but effective formulae:

$$\begin{pmatrix} \Delta p_x \\ \Delta p_y \\ \Delta p_z \end{pmatrix} \approx dt \cdot \begin{pmatrix} v_x \\ v_y \\ v_z \end{pmatrix}, \begin{pmatrix} \Delta \phi \\ \Delta \theta \\ \Delta \psi \end{pmatrix} \approx dt \cdot \underbrace{\begin{pmatrix} 1 & \sin(\psi)\tan(\theta) & \cos(\phi)\tan(\theta) \\ 0 & \cos(\psi) & -\sin(\psi) \\ 0 & \dfrac{\sin(\psi)}{\cos(\theta)} & \dfrac{\cos(\psi)}{\cos(\theta)} \end{pmatrix}}_{R} \begin{pmatrix} \omega_x \\ \omega_y \\ \omega_z \end{pmatrix},$$

where the formula for angular velocities can be found, for example, in Hover & Triantafyllou (2009). We will refer to the matrix $R$ as the rotation matrix with a slight abuse of notation. We also choose special features for the MLP: angular velocities, velocities, sines and cosines of the Euler angles ($E$), actions and actions squared. Using these expression we impose the structure on the neural network depicted in Figure A7(c).

Table A1: Hyper-parameters for model learning.

| | | Cart-Pole Swing-Up | Intersection | Drone Take-Off |
|---|---|---|---|---|
| Model Prior | $K$ | $[4, 5, 6, 8, 10, 20]$ | 10 | 10 |
| | $\gamma$ | 2 | 2 | 1 |
| | $\alpha$ | $1 \cdot 10^3$ | $1 \cdot 10^3$ | $5 \cdot 10^3$ |
| | $\kappa$ | $3 \cdot K/5$ | 6 | 3 |
| | $\text{std}_\theta$ | 0.1 | 0.1 | 0.1 |
| | transition cool-off | 5 | 5 | 5 |
| | Network dimensions | $\{6, 128, 4\}$ | $\{6, 64, 4\}$ | $\{20, 128, 6\}$ |
| | Activations | ReLU | ReLU | ReLU |
| | Optimizer | Clipped Adam | Clipped Adam | Clipped Adam |
| | Learning rates $\{\theta, \rho, \nu\}$ | $\{5 \cdot 10^{-3}, 10^{-2}, 10^{-2}\}$ | $\{5 \cdot 10^{-3}, 10^{-2}, 10^{-2}\}$ | $\{5 \cdot 10^{-3}, 10^{-2}, 10^{-2}\}$ |

Table A2: Hyper-parameters for SAC experiments.

| | | Cart-Pole Swing-Up | Intersection | Drone Take-Off |
|---|---|---|---|---|
| Runner | # roll-outs at warm-start | 100 | 200 | 200 |
| | # roll-outs per iteration | 1 | 1 | 1 |
| | # model iterations at warm-start | 500 | 500 | 500 |
| | # model iterations per epoch | 500 | 200 | 200 |
| | # agent updates at warm start | 1000 | 1000 | 100 |
| | # agent updates per epoch | 200 | 100 | 150 |
| | # epochs | 500 | 500 | 500 |
| | Model frequency update | 100 | 100 | 80 |
| | Model batch size | 20 | 50 | 50 |
| | Agent batch size | 256 | 256 | 256 |
| Prior | Training distillation threshold | 0.1 | 0.05 | 0.02 |
| | Testing distillation threshold | 0.1 | 0.05 | 0.02 |
| SAC | Policy network dimensions | $\{4, 256, 2\}$ | $\{12, 256, 256, 4\}$ | $\{12, 256, 4\}$ |
| | Policy networks activations | ReLU | ReLU | ReLU |
| | Value network layer dims | $\{4, 256, 1\}$ | $\{12, 256, 256, 1\}$ | $\{12, 256, 1\}$ |
| | Value networks activations | ReLU | ReLU | ReLU |
| | Target entropy | $-0.05$ | $-0.01$ | $-0.1$ |
| | Initial temperature | 0.8 | 0.2 | 0.6 |
| | Discount factor | 0.99 | 0.999 | 0.999 |
| | Target value fn update freq | 4 | 4 | 4 |
| Optimization | Optimizer | Adam | Adam | Adam |
| | Policy learning rate | $3 \cdot 10^{-4}$ or $7 \cdot 10^{-4}$ | $5 \cdot 10^{-4}$ | $3 \cdot 10^{-4}$ |
| | Value function learning rate | $3 \cdot 10^{-4}$ or $7 \cdot 10^{-4}$ | $5 \cdot 10^{-4}$ | $3 \cdot 10^{-4}$ |
| | Temperature learning rate | $5 \cdot 10^{-5}$ or $7 \cdot 10^{-5}$ | $1 \cdot 10^{-4}$ | $1 \cdot 10^{-4}$ |
| | Linear Learning Decay | True | True | True |
| | Weight Decay | $10^{-8}$ | $10^{-6}$ | $10^{-6}$ |

**Left turn on the Intersection in Highway Environment** We take the environment by Leurent (2018), but use the modifications made by Xu et al. (2020) including the overall reward function structure. We, however, do not penalize the collisions. and we increase the episode time from 40 to

Table A3: Hyper-parameters for MPC experiments.

| | | Cart-Pole Swing-Up | Intersection | Drone Take-Off |
|---|---|---|---|---|
| **Runner** | # roll-outs at warm-start | 100 | 200 | 200 |
| | # roll-outs per iteration | 20 | 20 | 20 |
| | # model iterations at warm-start | 500 | 500 | 500 |
| | # model iterations per epoch | 500 | 50 | 60 |
| | # epochs | 10 | 3 | 3 |
| | Model batch size | 100 | 100 | 50 |
| **Prior** | Training distillation threshold | 0.1 | 0 | 0 |
| | Testing distillation threshold | 0.02 | 0 | 0.02 |

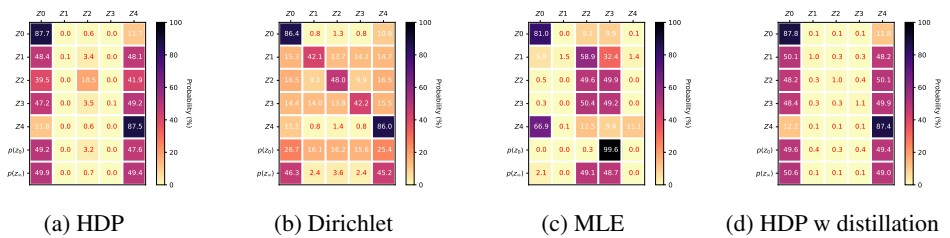

(a) HDP      (b) Dirichlet      (c) MLE      (d) HDP w distillation

Figure A8: Transition matrices, initial $p(z_0)$ and stationary $p(z_\infty)$ distributions of the learned context models in the Cart-Pole Swing-Up Experiment for Result A. $Z0 - Z4$ stand for the learned contexts. Reproduction of Figure 2 from the main text.

100 time steps. We again predicted the difference between current and next steps for the mean, and used the simplified model for the position, i.e., $\Delta p_x \approx dt v_x$, $\Delta p_y \approx dt v_y$ for both social and ego vehicles.

## D.3   HYPER-PARAMETERS

All the hyper-parameters are presented in Tables A1, A2 and A3. For model learning experiments we used 500 trajectory roll-outs and 500 epochs for optimization. In the cart-pole environment we used the higher learning rate for hard failure experiments when $\chi < 0$ and used the lower learning rate for the soft failure experiments $\chi > 0$. We use the weight decay to avoid gradient explosion in the value functions and the policies. Similarly, Clipped Adam optimizer (available in Pyro) was used to avoid gradient explosion in model learning.

## E   ADDITIONAL EXPERIMENTS

### E.1   HDP IS AN EFFECTIVE PRIOR FOR LEARNING AN ACCURATE AND INTERPRETABLE MODEL

We plot the time courses of the context evolution and the ground truth context evolution in Figure A9. As the results in Figure A8 (reproduction of Figure 2 in the main text) suggested the MLE method did not provide an accurate context model, while both DP and HDP priors provided models for reconstructing the true context cardinality after distillation. The difference was the choice of the distillation threshold, which had to be significantly higher for the DP prior. This experiment indicates that DP prior can be a good tool for modeling context transitions, but HDP provides sharper model fit and a more interpretable model.

For completeness, we performed the same experiment, but with a different seed and setting $\varepsilon_{\text{distil}} = 0.1$ during training. We plot the results in Figures A10 and A11. In this case, all models (including the MLE method) coupled with distillation provided an accurate estimate of the context evolution. This suggests that the optimization profile for the MLE method has many local minima (peaks and troughs in ELBO), which we can be trapped in given an unlucky seed.

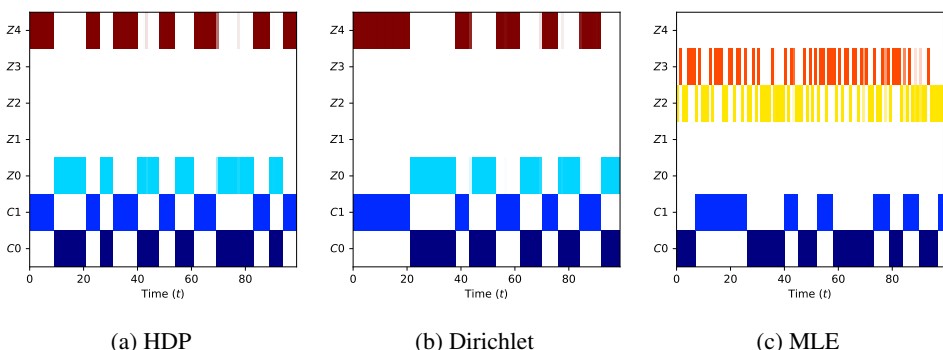

(a) HDP        (b) Dirichlet        (c) MLE

Figure A9: Time courses the learned context models in Cart-Pole Swing-Up Experiment. "Unlucky' random seed for MLE was used. $C0$ and $C1$ stand for the ground true contexts, while $Z0 - Z4$ are the learned contexts. Reproduction of Figure 3 from the main text.

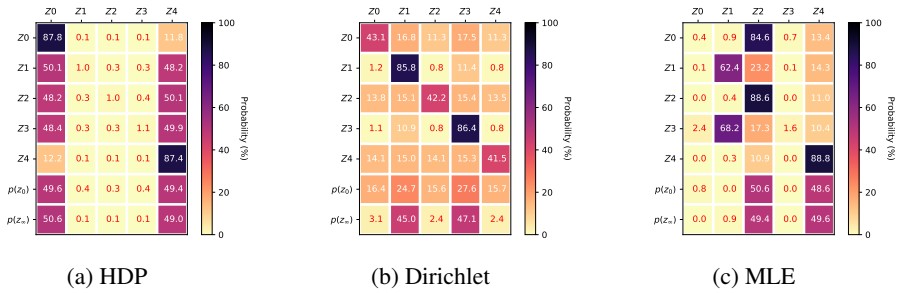

(a) HDP        (b) Dirichlet        (c) MLE

Figure A10: Transition matrices, initial $p(z_0)$ and stationary $p(z_\infty)$ distributions of the learned context models in Cart-Pole Swing-Up Experiment. Distillation during training and a "lucky" seed were used. $Z0 - Z4$ are the learned contexts.

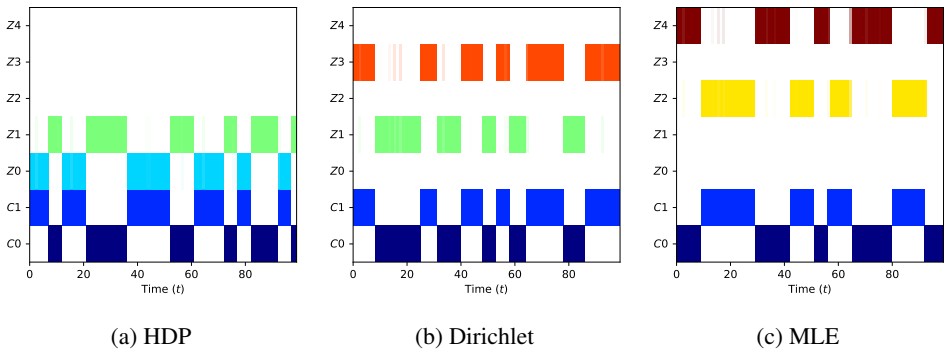

(a) HDP        (b) Dirichlet        (c) MLE

Figure A11: Time courses the learned context models in Cart-Pole Swing-Up Experiment. Distillation during training and a "lucky" seed were used. $C0$ and $C1$ stand for the ground true contexts, while $Z0 - Z4$ are the learned contexts.

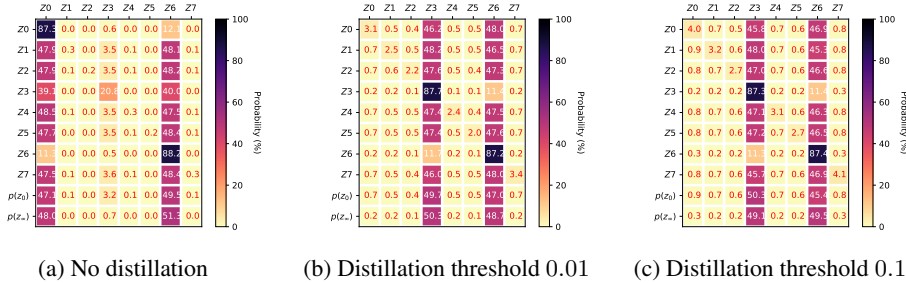

(a) No distillation      (b) Distillation threshold 0.01      (c) Distillation threshold 0.1

Figure A12: Influence of distillation during training in Cart-Pole Swing-Up Experiment with $|\widetilde{\mathcal{C}}| = 8$.

Table A4: Comparing the probability mass of the third most probable state in the stationary distribution. We vary the cardinality of the estimated context set $\widetilde{\mathcal{C}}$ and the distillation threshold $\varepsilon_{\text{distil}}$. Red indicates underestimation of the distillation threshold. Reproduction of Table 1 from the main text.

| $\varepsilon_{\text{distil}} \downarrow$ $\quad$ $|\widetilde{\mathcal{C}}| \rightarrow$ | 4 | 5 | 6 | 8 | 10 | 20 | 100 |
|---|---|---|---|---|---|---|---|
| **0** | 8.58e-03 | 7.06e-03 | 3.71e-03 | 6.85e-03 | 2.20e-03 | 2.25e-02 | 1.37e-01 |
| **0.01** | 1.06e-03 | 1.24e-03 | 1.37e-03 | 2.19e-03 | 2.56e-03 | 1.60e-02 | 8.84e-03 |
| **0.1** | 1.21e-03 | 1.54e-03 | 1.70e-03 | 2.80e-03 | 3.54e-03 | 9.86e-03 | 5.03e-02 |

### E.2 DISTILLATION ACTS AS A REGULARIZER

After the first experiment, we noticed that the context $Z2$ has a low probability mass in stationarity, but a high probability of self-transition (see Figure A8 - reproduction of Figure 2 in the main text). This suggest that spurious transitions can happen, while highly unlikely. We speculate that the learning algorithm tries to fit the uncertainty in the model (e.g., due to unseen data) to one context. This can lead to over-fitting and unwanted side-effects. Results in Figure A8 (reproduction of Figure 2 in the main text) suggest that distillation during training can act as a regularizer when we used a high enough threshold $\varepsilon_{\text{distil}} = 0.1$. We further validate our findings by changing the context number upper bound $|\widetilde{\mathcal{C}}|$ between 4 and 20. In particular, we proceed by presenting the results for varying $|\widetilde{\mathcal{C}}|$ (taking values 4, 5, 6, 8, 10, 20 and 100) and the distillation threshold $\varepsilon_{\text{distil}}$ (taking values 0, 0.01, and 0.1). Note that we distill during training and we refer to the transition matrix for the distilled Markov chain as the distilled transition matrix. First, consider the results in Figure A12, where we plot the learned MAP estimates of the transition matrices with $|\widetilde{\mathcal{C}}| = 8$. Note that using distillation with both thresholds prevents overfitting to one context. One could argue that instead of context distillation during training one could simply use distillation after training. This approach, however, can lead to emergence of a spurious context with a large probability of a self-transition raising the possibility of inaccurate model predictions. Furthermore, the large number of spurious contexts can lead to a large probability mass concentrated in one of them in stationarity. Indeed, consider Table A4, where we plot the context with third largest probability mass. In particular, for $|\widetilde{\mathcal{C}}| = 20$ the probability mass values for this context are larger than 0.01. This indicates a small but not insignificant possibility of a transition to this context, if the distillation does not remove it.

We verify that the distilled transition matrices for various cardinalities $|\widetilde{\mathcal{C}}|$ are close to each other. Let $\hat{\boldsymbol{R}}_K$ denote the estimated transition matrix for $|\widetilde{\mathcal{C}}| = K$ with only two contexts chosen a posteriori. We use the following metric for comparison

$$\delta_K = \frac{\|\hat{\boldsymbol{R}}_K - \hat{\boldsymbol{R}}_5\|_1}{\|\hat{\boldsymbol{R}}_5\|_1}.$$

That is, we compare all the transition matrices to the case of $|\widetilde{\mathcal{C}}| = 5$. In Table A5 we present the results, which indicate that the estimated transition matrices are quite close to each other. Furthermore, the distillation during training helps to recover the true context regardless of the upper bound $|\widetilde{\mathcal{C}}|$.

Table A5: Comparing the estimated transition matrices using the metric $\delta_{|\widetilde{\mathcal{C}}|} = \frac{\|\hat{\boldsymbol{R}}_{|\widetilde{\mathcal{C}}|} - \hat{\boldsymbol{R}}_5\|_1}{\|\hat{\boldsymbol{R}}_5\|_1}$. We vary the cardinality $\widetilde{\mathcal{C}}$ and the distillation threshold $\varepsilon_{\text{distil}}$.

| $\varepsilon_{\text{distil}} \downarrow$ $\quad |\widetilde{\mathcal{C}}| \rightarrow$ | 4 | 5 | 6 | 8 | 10 | 20 | 100 |
|---|---|---|---|---|---|---|---|
| **0** | 8.26e-03 | 0 | 7.69e-03 | 2.79e-03 | 1.29e-02 | 2.89e-02 | 1.42e-01 |
| **0.01** | 6.08e-03 | 0 | 6.41e-03 | 8.88e-03 | 3.61e-03 | 2.14e-02 | 1.77e-02 |
| **0.1** | 2.09e-03 | 0 | 1.66e-03 | 4.87e-03 | 1.38e-02 | 2.19e-02 | 2.60e-02 |

To summarize, our experiments suggest that **the context set cardinality $|\widetilde{\mathcal{C}}|$ can be confidently overestimated** and the context set can be reconstructed using our distillation procedure.

### E.3 CONTEXT CARDINALITY VS MODEL COMPLEXITY

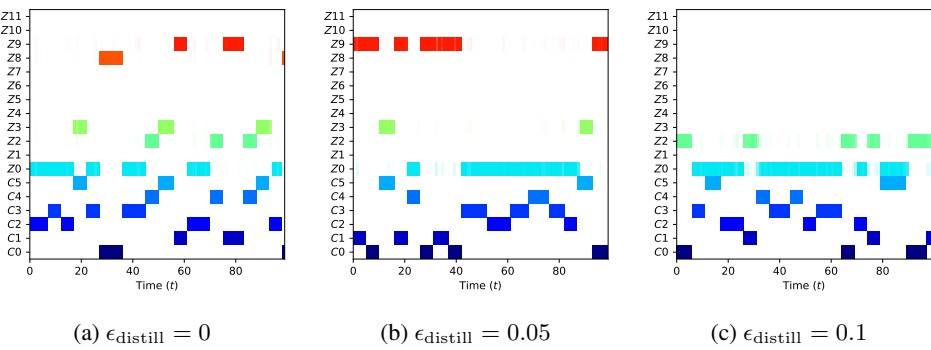

(a) $\epsilon_{\text{distill}} = 0$        (b) $\epsilon_{\text{distill}} = 0.05$        (c) $\epsilon_{\text{distill}} = 0.1$

Figure A13: Time courses of the learned context models for various distillation thresholds where $\mathcal{C} = \{\{-1\}, \{-0.5\}, \{0.05\}, \{0.1\}, \{0.5\}, \{1\}\}$. We deduce that the learned context sets: $\widehat{\mathcal{C}}_0 = \{\{-1\}, \{-0.5\}, \{0.05, 0.1\}, \{0.5\}, \{1\}\}$, $\widehat{\mathcal{C}}_{0.05} = \{\{-1, -0.5\}, \{0.05, 0.1, 0.5\}, \{1\}\}$, $\widehat{\mathcal{C}}_{0.1} = \{\{-1, -0.5\}, \{0.05, 0.1, 0.5, 1\}\}$.

Next we demonstrate that our algorithm can merge similar context automatically using an appropriate distillation threshold during training. We increased the force magnitude by two-fold in order to create a large number of distinct contexts and chosen the context set as $\mathcal{C} = \{\{-1\}, \{-0.5\}, \{0.5\}, \{0.05\}, \{0.1\}, \{0.5\}, \{1\}\}$ and vary the distillation threshold $\epsilon_{\text{distill}}$ setting it to 0, 0.05 and 0.1 during training. In order to get the estimated context sets we distilled one more time after training with $\epsilon_{\text{distill}} = 0.02$. We set a sufficiently high context cardinality estimate $K = 12$. Results in Figure A13 suggest that the learned contexts sets for various distillation threshold are as follows:

$$\widehat{\mathcal{C}}_0 = \{\{-1\}, \{-0.5\}, \{0.05, 0.1\}, \{0.5\}, \{1\}\},$$
$$\widehat{\mathcal{C}}_{0.05} = \{\{-1, -0.5\}, \{0.05, 0.1, 0.5\}, \{1\}\},$$
$$\widehat{\mathcal{C}}_{0.1} = \{\{-1, -0.5\}, \{0.05, 0.1, 0.5, 1\}\}.$$

This indicates the ability to change the model structure using the distillation threshold. Furthermore, the trade-off between continuous and discrete dynamics is decided automatically using the distillation threshold.

### E.4 BREAKING THE ASSUMPTIONS IN HIDDEN MARKOV MODELS

One of the major assumptions in our framework is the Markovian nature of the switches, which can limit its applicability. We conduct two simple experiments to assess if these assumptions can be broken without detrimental effects. First, we model the context transitions as a process, where the next context depends on the previous context rather than the current context. This process is clearly

non-Markovian. However, results in Figure A14 indicate that our model successfully predicts the correct contexts in the context realization. Second, we make the context state dependent. In particular, we have:

$$c = 0, \quad \text{if pos} \in [0, 0.1), \qquad [-0.1, -0.2), \quad [0.2, 0.3), \qquad [-0.3, -0.4), \quad \cdots$$
$$c = 1, \quad \text{if pos} \in [0, -0.1), \quad [0.1, 0.2), \qquad [-0.2, -0.3), \quad [0.3, 0.4), \qquad \cdots,$$

where pos is the position of the cart. Again our model handles this case as the results in Figure A15 suggest.

So how our Markovian model handles predictions for non-Markovian models? In both cases, the key is comparing context estimations/predictions rather than context models, which are almost surely incorrect. However, the predictive power of our model is still preserved since it relies heavily on the observed state history. This allowed us to correctly estimate the contexts in these experiments.

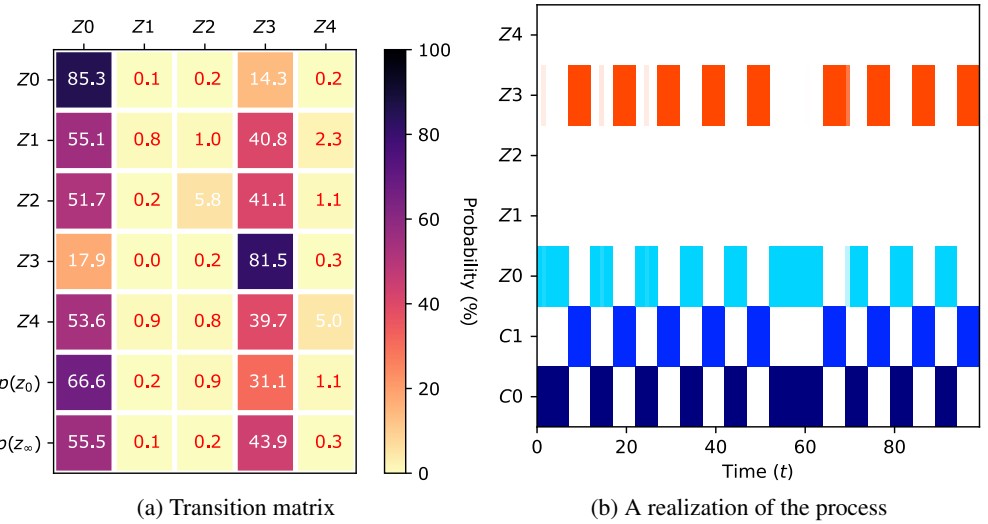

(a) Transition matrix      (b) A realization of the process

Figure A14: Learning non-Markovian transitions.

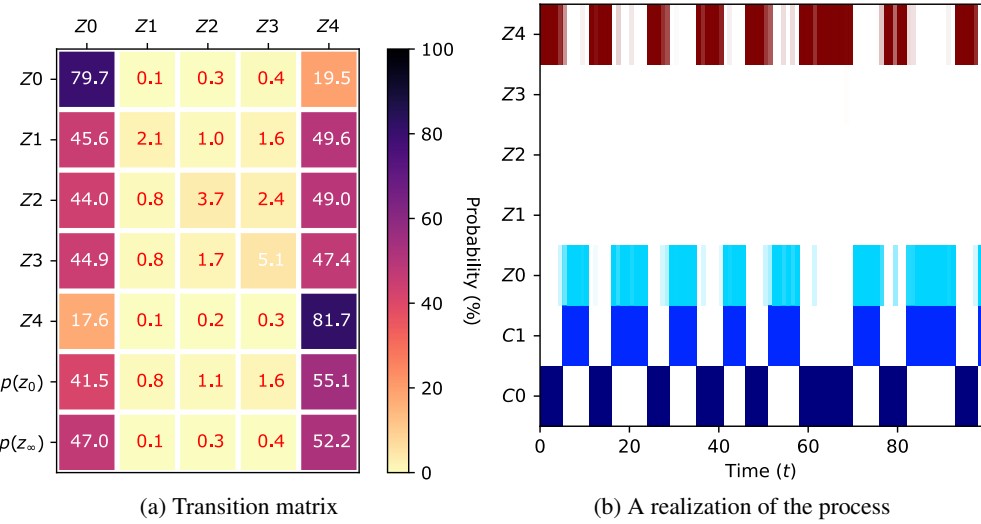

(a) Transition matrix      (b) A realization of the process

Figure A15: Learning state-dependent transitions.

### E.5   LEARNING NEW CONTEXTS

As we mentioned above we can remove the new contexts following the proposed distillation procedure. However, our model is flexible enough to add new (unseen) contexts without learning the state

transition model from scratch. We design the following experimental procedure to demonstrate this ability: (1) train the model on dataset $\mathcal{D}_1$ which contains two contexts ($C0$ and $C1$); (2) reset free parameters in the variational distribution $q(\boldsymbol{\nu}|\hat{\boldsymbol{\nu}})$ and $q(\boldsymbol{\mu}|\hat{\boldsymbol{\mu}})$ while preserving $q(\boldsymbol{\theta}|\hat{\boldsymbol{\theta}})$; (3) re-train the model on dataset $\mathcal{D}_2$ which contains the two original contexts and two new contexts ($C2$ and $C3$). We present the training results on the sets $\mathcal{D}_1$ and $\mathcal{D}_2$ in Figure A16 and A17, respectively. These results suggest that our model is able to learn new contexts ($C2$ and $C3$ in Figure A17), while preserving the original contexts ($C0$ and $C1$ in both Figures A16 and A17).

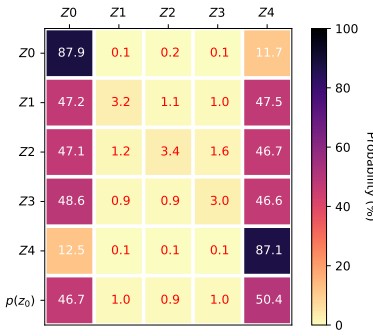

(a) Learned context model on $\mathcal{D}_1$

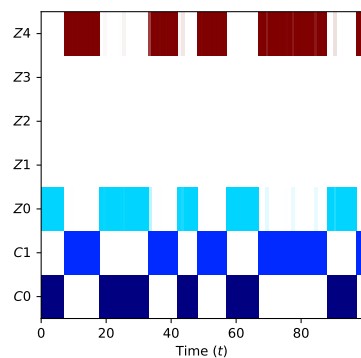

(b) Learned context transitions on $\mathcal{D}_1$

Figure A16: Learning the context on the set $\mathcal{D}_1$. $C0$ and $C1$ stand for the ground true contexts, while $Z0$-$Z4$ are the learned contexts.

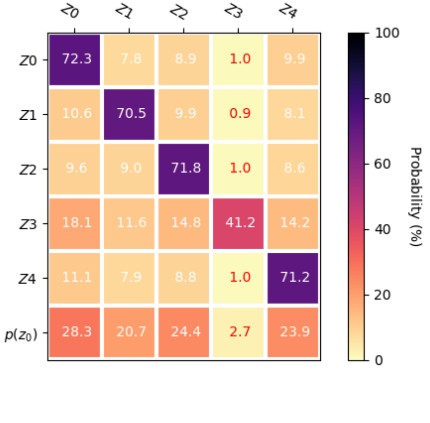

(a) Context model

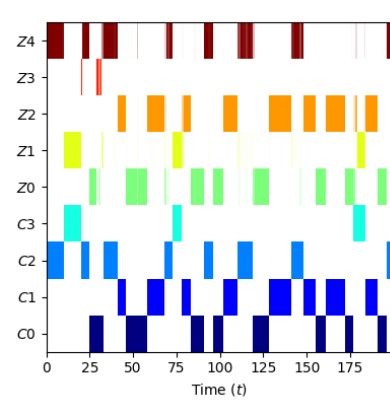

(b) Context transition

Figure A17: Expanding the model by learning new contexts on the set $\mathcal{D}_2$. $C0$-$C3$ stand for the ground true contexts, while $Z0$-$Z4$ are the learned contexts.

## E.6 COMPARING TO POMDP AND C-MDP METHODS

We chosen the context set as $\mathcal{C} = \{-1, \chi\}$ for $\chi \in \{-1, 0.1, 0.3, 0.5\}$. However, for $\chi > 0$, we again increased the force magnitude by two-fold. For SAC-based algorithms we present the statistics for 50 episodes and 3 seeds. To avoid unfair comparison to POMDP and continual RL methods, we pick their best performance and explain why these methods are not well-suited for our problem. For GPMM we run ten separate experiments with 15 episode each and picked three best runs and the best episode performance. For RNN-PPO we run three experiments, but picked the best learned policy over time and over the runs. Furthermore, we use the RNN in a more favorable setting as we assume that the number of contexts is known and can be hard coded in the RNN architecture. We present our experimental results in Table A6. While it seems that RNN-PPO learns to swing-up correctly for

| failure → algo ↓ | hard | soft $\alpha = 0.1$ | soft $\alpha = 0.3$ | soft $\alpha = 0.5$ |
|---|---|---|---|---|
| FI-SAC | $84.50 \pm 1.79$ | $\mathbf{76.63 \pm 8.54}$ | $\mathbf{84.75 \pm 3.07}$ | $\mathbf{86.92 \pm 1.03}$ |
| C-SAC | $85.38 \pm 1.64$ | $\mathbf{76.80 \pm 8.91}$ | $\mathbf{86.76 \pm 2.88}$ | $\mathbf{88.35 \pm 1.30}$ |
| C-CEM | $\mathbf{87.63 \pm 0.14}$ | $60.15 \pm 25.91$ | $83.15 \pm 7.72$ | $\mathbf{89.08 \pm 1.90}$ |
| DNNMM | $-7.73 \pm 17.62$ | $2.87 \pm 15.18$ | $28.24 \pm 22.87$ | $66.35 \pm 19.67$ |
| ANPMM | $-3.35 \pm 15.64$ | $8.07 \pm 16.48$ | $32.08 \pm 19.54$ | $57.22 \pm 25.55$ |
| GPMM | $3.50 \pm 18.59$ | $3.55 \pm 7.83$ | $10.64 \pm 16.10$ | $49.61 \pm 19.13$ |
| RNN-PPO | $-0.17 \pm 18.06$ | $64.10 \pm 21.37$ | $74.58 \pm 20.66$ | $67.01 \pm 8.52$ |

Table A6: Mean $\pm$ standard deviation for: our algorithms (C-SAC, C-CEM), continual learning algos (GPMM, DNNMM, ANPMM), a POMDP algo (RNN-PPO), and SAC with a known context (FI-SAC). For soft failure experiments, we have increased the maximum applicable force by the factor of two. Reproduction of Table 2 from the main text.

soft failures and GPMM almost learns to swing-up for $\chi = 0.5$, the learned belief models indicate that this not so. In fact, plotting the evolution of the belief for RNN-PPO and the context evolution for GPMM illustrates that the algorithms do not learn the context model (see Figure 4 in the main text). In order to illustrate that it is not Gaussian Processes that cause failure in GPMM, we replace the Gaussian Process mixture with Deep Neural Network and Attentive Neural Process (Qin et al., 2019; Kim et al., 2019) mixtures (DNNMM and ANPMM, respectively) using the code from Xu et al. (2020). The results in this case are similar to the GPMM case, i.e., we manage to get reasonable rewards for $\alpha = 0.5$, but fail for other cases.

### E.7 EXPERIMENTS WITH A LARGER NUMBER OF CONTEXTS

We further test our approach by introducing a larger number of contexts. We have the following parameter sets

$$c_{\text{friction}} = [0, 0.1], \ c_{\text{gravity}} = [9.82, 50], \ c_{\text{cart mass}} = [0.5, 5], \ c_{\text{max force}} = [20, 40],$$

with 16 parameters in total. The contexts for our experiment are as follows:

| | | | | |
|---|---|---|---|---|
| $C_0:$ | $c_{\text{friction}} = 0.1,$ | $c_{\text{gravity}} = 9.82,$ | $c_{\text{cart mass}} = 0.5,$ | $c_{\text{max force}} = 40,$ |
| $C_1:$ | $c_{\text{friction}} = 0.1,$ | $c_{\text{gravity}} = 9.82,$ | $c_{\text{cart mass}} = 0.5,$ | $c_{\text{max force}} = 20,$ |
| $C_2:$ | $c_{\text{friction}} = 0.1,$ | $c_{\text{gravity}} = 9.82,$ | $c_{\text{cart mass}} = 5,$ | $c_{\text{max force}} = 40,$ |
| $C_3:$ | $c_{\text{friction}} = 0.1,$ | $c_{\text{gravity}} = 9.82,$ | $c_{\text{cart mass}} = 5,$ | $c_{\text{max force}} = 20,$ |
| $C_4:$ | $c_{\text{friction}} = 0.1,$ | $c_{\text{gravity}} = 50,$ | $c_{\text{cart mass}} = 0.5,$ | $c_{\text{max force}} = 40,$ |
| $C_5:$ | $c_{\text{friction}} = 0.1,$ | $c_{\text{gravity}} = 50,$ | $c_{\text{cart mass}} = 0.5,$ | $c_{\text{max force}} = 20,$ |
| $C_6:$ | $c_{\text{friction}} = 0.1,$ | $c_{\text{gravity}} = 50,$ | $c_{\text{cart mass}} = 5,$ | $c_{\text{max force}} = 40,$ |
| $C_7:$ | $c_{\text{friction}} = 0.1,$ | $c_{\text{gravity}} = 50,$ | $c_{\text{cart mass}} = 5,$ | $c_{\text{max force}} = 20,$ |
| $C_8:$ | $c_{\text{friction}} = 0,$ | $c_{\text{gravity}} = 9.82,$ | $c_{\text{cart mass}} = 0.5,$ | $c_{\text{max force}} = 40,$ |
| $C_9:$ | $c_{\text{friction}} = 0,$ | $c_{\text{gravity}} = 9.82,$ | $c_{\text{cart mass}} = 0.5,$ | $c_{\text{max force}} = 20,$ |
| $C_{10}:$ | $c_{\text{friction}} = 0,$ | $c_{\text{gravity}} = 9.82,$ | $c_{\text{cart mass}} = 5,$ | $c_{\text{max force}} = 40,$ |
| $C_{11}:$ | $c_{\text{friction}} = 0,$ | $c_{\text{gravity}} = 9.82,$ | $c_{\text{cart mass}} = 5,$ | $c_{\text{max force}} = 20,$ |
| $C_{12}:$ | $c_{\text{friction}} = 0,$ | $c_{\text{gravity}} = 50,$ | $c_{\text{cart mass}} = 0.5,$ | $c_{\text{max force}} = 40,$ |
| $C_{13}:$ | $c_{\text{friction}} = 0,$ | $c_{\text{gravity}} = 50,$ | $c_{\text{cart mass}} = 0.5,$ | $c_{\text{max force}} = 20,$ |
| $C_{14}:$ | $c_{\text{friction}} = 0,$ | $c_{\text{gravity}} = 50,$ | $c_{\text{cart mass}} = 5,$ | $c_{\text{max force}} = 40,$ |
| $C_{15}:$ | $c_{\text{friction}} = 0,$ | $c_{\text{gravity}} = 50,$ | $c_{\text{cart mass}} = 5,$ | $c_{\text{max force}} = 20.$ |

We learn a model with $K = 20$. We further add a structure on the contexts transition matrix: form every contexts we can switch only to and from two contexts, i.e, from the context $i$ we can switch to/from the context $i - 1$ and the context $i + 1$, where operations on $i$ should be understood as modulo

$K$ (i.e., $-1 \triangleq K-1$, $K \triangleq 0$), i.e.:

$$p(z_i|z_j) = \begin{cases} 0.6 & i = j, \\ 0.2 & j = \mathrm{mod}(i+1, K) \text{ or } j = \mathrm{mod}(i-1, K), \\ 0 & \text{otherwise} \end{cases}$$

Similarly to our previous settings we have a transition cool-off period of $5$ time steps.

As the results in Figures A18(a) and A18(b) suggest, we identify the following meaningful contexts: $Z_1 = \{C_1, C_9\}$, $Z_4 = \{C_6, C_7, C_{14}, C_{15}\}$, $Z_{10} = \{C_4, C_5, C_{12}, C_{13}\}$, $Z_{14} = \{C_2, C_3, C_{10}, C_{11}\}$, $Z_{18} = \{C_0, C_4, C_8\}$. Our algorithm does not distinguish the difference in the values of the friction parameter. The difference in maximum force is not identified for larger masses and stronger gravity, but it can be identified for cart mass $0.5$ and gravity $9.82$. The other contexts are characterized by different gravity values and cart masses. The only exception is the ground truth context $C_4$, which is present in both $Z_{10}$ and $Z_{18}$. This, however, is bound to happen if the ground truth contexts are hard to separate.

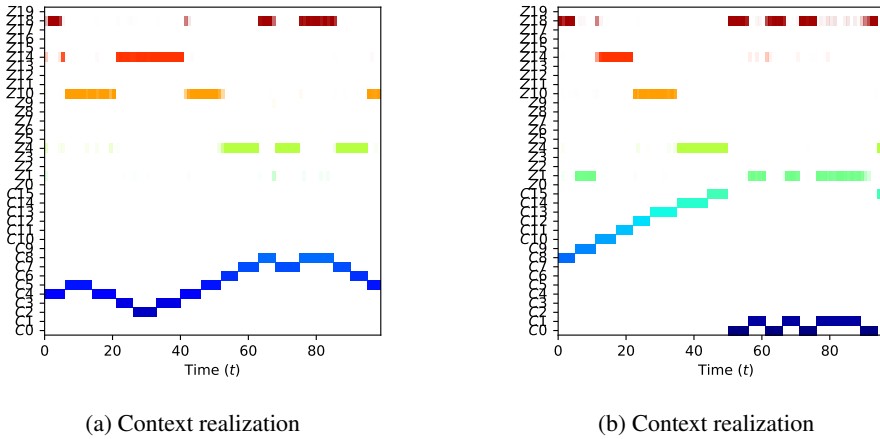

(a) Context realization

(b) Context realization

Figure A18: Learning a context model with $16$ contexts.

