# OpenReview forum: "Reinforcement Learning in Presence of Discrete Markovian Context Evolution "
_ICLR.cc/2022/Conference — ICLR 2022 Poster_

### Official Review · Reviewer_w49w · 2021-10-26

**Correctness:** 4
**Technical Novelty And Significance:** 3
**Empirical Novelty And Significance:** 2
**Recommendation:** 6
**Confidence:** 4

**Main Review:**

This paper considers the important and timely problem of context shift in reinforcement learning. This is a popular area of research at the moment. Recent works such as SODA and DRQ have invigorated the communities interest in domain shift and related problems. The treatment given to the problem here is Bayesian, using HDPs and probabilistic inference to handle the test time context switching. It's nice to see such a treatment, as this was a gap in the current literature.

The domain shift problem is in this paper cast as a contextual MDP (C-MDP). As best as I can tell, a contextual MDP is just an MDP where the state transitions and reward distribution shift around, possibly during training and possibly just at test time. (small note: why does the reward function lack the C subscript if it depends on context?) This machinery seems to play nicely into the eventual HDP model the author's develop. Although, I'm not entirely convinced this additional competing standard is needed. There is already a lot of interesting work on domain shift in robotics. I didn't feel like the new framework was justified, beyond making it easier to recast the problem as probabilistic inference. This is a minor complaint and just an observation. There are always issues with too many competing standards and all that.

It seems that the number of different contexts must be finite and a hand designed heuristic is required to prune connections and avoid spurious correlations. Perhaps this is a standard part of the package whenever you're using HDP. In any case, often the core problem in domain shift is that the learned distributions are not wide enough. And I worry that this sort of limit on the number of contexts might be a hinderance for wider adoption of this model (which I would like to see!) Do we know where the upper bound on problem dimensionality is? There are new domains (distractor control suit) where you can set distractor intensity, in essence producing arbitrarily many distinct contexts for the environment. Is this limit on the number of atoms going to be the first thing that breaks down as the number of contexts scales? Or something else?

The trick used to derive equation (4) is cute. Thank you for calling this to my attention.

There are more clear descriptions in the appendix, but it was not clear to me from reading section 3 what the step by step algorithm is. I could not reproduce these results from the main body of the paper alone.

Can you discuss a little how RNN-PPO relates to RL2, which in essence just fuses the belief state estimate and PPO training steps? There is a general lack of comparison with and concern for both meta learning and domain shift baselines in this paper. Perhaps the authors feel that contextual MDPs is a sufficiently distance problem and prior art in these areas is therefore irrelevant. If that is the case, I would like to see a little further justification. Domain shift in general is a rich area of research (SODA, DRQ). Is this paper justified in ignoring that prior art? Have I fundamentally misunderstood your problem setting?

The interpretability experiments in Result A are strong. I really like seeing results like this.

The choices of Drone and Intesection are a little strange. Are there no Deep Mind Control Suit or similar environments that could work? I haven't seen these environments before in RL, so it is hard for me to evaluate how impressive these results are. A lot of the results are done on cartpole, which is fine for trying to pull apart the method. Although it is a bit concerning because cartpole famously works even when the underlying RL algorithm is broken. I would like to see results on further hard RL environments. In Table 3, the environment is called Highway. At the bottom of page 8 (bold text) it's called Intersection.

I am more than willing to bump my score if some of the questions I asked are answered. It's mostly a few points of clarification.

**Summary Of The Paper:**

This paper considers the problem of MDPs with underlying context variables. As these context variables shift, so too does the environmental MDP an RL agent encounters. In particular, this paper considers domain shift in state transitions and reward functions. A new method which relies on hierarchical dirchlet clustering is introduced. This model, as far as I can tell, uses bayesian probabilistic inference to see what parts of the underlying environment probability distribution are useful across context shifts.

A theorem is provided which draws a relationship between the number of mis-identified context switches and the error in the estimated value function. This quantifies the robustness of the algorithm under sudden context shift.

**Summary Of The Review:**

This paper is an interesting look at domain shift from a bayesian probabilistic modeling perspective. The experiments do a good job of taking apart the design choices, in particular results A and B. I have a few lingering questions about this paper's relationship to prior work in Meta Learning and Domain Shift.

---

> ### Author Response · Authors · 2021-11-15
> **Thank you for a careful consideration of the manuscript and the comments! (response 1)**
>
> We want to thank the reviewer for the careful consideration of the manuscript and their comments. Please see the revised version of the manuscript in the pdf file. All significant changes are in magenta.   We apologize for possible new typos, but we wanted to respond as fast as we could.
>
> We also would like to thank the reviewer for drawing our attention to the domain shift literature. It could be a great application of our techniques, but unfortunately a direct comparison at this stage might be premature as we discuss in what follows.
>
> **(small note: why does the reward function lack the C subscript if it depends on context?)**
>
> Thank you for this comment! In the definition of the C-MDP in Section 2 we do not use C as a subscript to signify dependence on the context but rather assume that all the probability distributions depend on the context. This is the reason why we do not use this subscript for the reward function.
>
> **Although, I'm not entirely convinced this additional competing standard is needed. There is already a lot of interesting work on domain shift in robotics. I didn't feel like the new framework was justified, beyond making it easier to recast the problem as probabilistic inference. This is a minor complaint and just an observation.**
> **There is a general lack of comparison with and concern for both meta learning and domain shift baselines in this paper. Perhaps the authors feel that contextual MDPs is a sufficiently distance problem and prior art in these areas is therefore irrelevant. If that is the case, I would like to see a little further justification. Domain shift in general is a rich area of research (SODA, DRQ). Is this paper justified in ignoring that prior art? Have I fundamentally misunderstood your problem setting?**
>
> Thank you for this comment! We apologize for the confusion, we added minor remarks throughout the paper to highlight the main differences, which we restate for clarity. We assume that the context can change during an episode, in fact, it can change at every time step. Further, we do not observe the context and the boundary between contexts (context switching times). This differentiates our approach from most of the existing methods in meta-RL/ continual RL and the domain shift literature (see Figure A1 in Appendix A for an illustration of different settings).
> The main difficulty in our case is that we need to adapt fast to the new context, which is feasible if we assume that there is a finite number of contexts. Some meta-RL methods can assume infinite cardinality of the context set, however, at the expense of the fast adaptation and restricting to the case when the contexts are independently sampled from a fixed distribution.
>
> Now we address the similarities and difference to SODA ([6]), DRQ ([7]). These works focus on control from pixels, and hence the underlying problem could also be modeled as a POMDP. In this case, the true states $s_t$ (e.g., states, velocities) are observed only through pixels, but the states themselves constitute an MDP (i.e., the next state depends on the previous step and the previous action). The observation layer can shift and change, but the states would not generally shift. In our case, we observe the true state $s_t$ (e.g., states, velocities), however, the one-step transitions depend on the context $z_t$ making our problem a POMDP. Therefore, while similar these frameworks are quite different due to the underlying assumptions, which makes the algorithms mutually inapplicable in general.
>
> It would be interesting to extend our approach to states observed through pixels, and we agree that one could extend our model learning approach to this domain shift application. These considerations, however, go beyond the scope of our paper. Nevertheless we added the discussion on the domain shift papers in our detailed literature review in Appendix A.
>
> References:
>
> [1] RL2 : FAST REINFORCEMENT LEARNING VIA SLOW REINFORCEMENT LEARNING Yan Duan, John Schulman, Xi Chen, Peter L. Bartlett, Ilya Sutskever, Pieter Abbeel
>
> [2]  Annie Xie, James Harrison, and Chelsea Finn. Deep reinforcement learning amidst lifelong non-stationarity.arXiv preprint arXiv:2006.10701, 20
>
> [3] Ilya Kostrikov. Pytorch implementations of reinforcement learning algorithms.https://github.com/ikostrikov/pytorch-a2c-ppo-acktr-gail, 2018
>
> [4] Mengdi Xu, Wenhao Ding, Jiacheng Zhu, Zuxin Liu, Baiming Chen, and Ding Zhao. Task-agnostic online reinforcement learning with an infinite mixture of Gaussian processes. Advances in Neural Information Processing Systems, 33, 2020
>
> [5] Yee Whye Teh, Michael I Jordan, Matthew J Beal, and David M Blei. Hierarchical Dirichlet processes. Journal of the American Statistical Association, 101(476):1566–1581, 2006.
>
> [6] Generalization in Reinforcement Learning by Soft Data Augmentation
> Nicklas Hansen, Xiaolong Wang
>
> [7] Image Augmentation Is All You Need: Regularizing Deep Reinforcement Learning from Pixels. Ilya Kostrikov, Denis Yarats, Rob Fergus

---

> ### Author Response · Authors · 2021-11-15
> **Response 2 (continuation of Response 1 below)**
>
> **It seems that the number of different contexts must be finite and a hand designed heuristic is required to prune connections and avoid spurious correlations. Perhaps this is a standard part of the package whenever you're using HDP.**
>
> **In any case, often the core problem in domain shift is that the learned distributions are not wide enough. And I worry that this sort of limit on the number of contexts might be a hinderance for wider adoption of this model (which I would like to see!) Do we know where the upper bound on problem dimensionality is?**
>
> Thank you for these comments! Our distillation method can be omitted (i.e., used with $\varepsilon_{\rm distill}=0$), but it can be very useful. We can still keep the spurious contexts, but this could decrease performance through mislabeling the contexts.
>
> The number of contexts must be finite, since we perform the gradient ascent over ELBO requiring the finite number of parameters. However, using a different learning method or a different HDP sampling approach (e.g., Chinese restaurant franchise) can potentially result in a different solution (see for example [5]) more amenable for the infinite context set cardinality assumption.
>
> However, even using our algorithm, we can always add further contexts if required and relearn the context transition model as we have demonstrated in Appendix E5 (Learning new contexts).
>
> We will try to further address this comment in the coming weeks by adding more experiments.
>
> **There are more clear descriptions in the appendix, but it was not clear to me from reading section 3 what the step by step algorithm is. I could not reproduce these results from the main body of the paper alone.**
>
> Thank you for the comment and apologize for this oversight!  We added a high-level pseudo code of the main algorithm in the main text and a few comments discussing the missing details.
>
> **Can you discuss a little how RNN-PPO relates to RL2, which in essence just fuses the belief state estimate and PPO training steps?**
>
> Thank you for this comment! The RNN-PPO algorithm is not directly related to RL2 ([1]), it is rather a modification of the code of [3] through changing the RNN model. In [3] the hidden state of the RNN takes as inputs previous states and actions, while it  is also the only input to the policy. That is the policy depends only on the hidden state of the RNN. In our case, the RNN predicts only the belief, while the state is received from the environment.
>
> In our understanding, RL2 ([1]) has an RNN linking different episodes of the training trials. In RL2 the trial can be seen a collection (an episode) of episodes. Hence our setting reduces to RL2, if we restrict context switching times to the end of episodes. We suppose RL2 is more related to [2] rather than our method. One could also claim that RL2 and [2] are less general as they restrict the switching times of the context process (switches occur after every episode in a trial, i.e., every T time steps), we do not make such an assumption.
>
> References:
>
> [1] RL2 : FAST REINFORCEMENT LEARNING VIA SLOW REINFORCEMENT LEARNING Yan Duan, John Schulman, Xi Chen, Peter L. Bartlett, Ilya Sutskever, Pieter Abbeel
>
> [2]  Annie Xie, James Harrison, and Chelsea Finn. Deep reinforcement learning amidst lifelong non-stationarity.arXiv preprint arXiv:2006.10701, 20
>
> [3] Ilya Kostrikov. Pytorch implementations of reinforcement learning algorithms.https://github.com/ikostrikov/pytorch-a2c-ppo-acktr-gail, 2018
>
> [4] Mengdi Xu, Wenhao Ding, Jiacheng Zhu, Zuxin Liu, Baiming Chen, and Ding Zhao. Task-agnostic online reinforcement learning with an infinite mixture of Gaussian processes. Advances in Neural Information Processing Systems, 33, 2020
>
> [5] Yee Whye Teh, Michael I Jordan, Matthew J Beal, and David M Blei. Hierarchical Dirichlet processes. Journal of the American Statistical Association, 101(476):1566–1581, 2006.
>
> [6] Generalization in Reinforcement Learning by Soft Data Augmentation
> Nicklas Hansen, Xiaolong Wang
>
> [7] Image Augmentation Is All You Need: Regularizing Deep Reinforcement Learning from Pixels. Ilya Kostrikov, Denis Yarats, Rob Fergus

---

> ### Author Response · Authors · 2021-11-15
> **Response 3 (continuation of Response 2 below)**
>
> **The choices of Drone and Intesection are a little strange. Are there no Deep Mind Control Suit or similar environments that could work? I haven't seen these environments before in RL, so it is hard for me to evaluate how impressive these results are.**
>
> Thank you for this comment! The results are here to highlight that more challenging environments in terms of the number of states can also be solved using our method. From our point of view the main baseline was FI-SAC (i.e., the situation when have access to the full information and use soft-actor-critic).
>
> We chose the highway-intersection as it was used by Xu et al in [4]. In fact, we used their modifications of the environment and then adapted it to our setting. However, their environment had a very small time horizon (40 time steps) and increasing the time horizon made the MPC approach much less effective even with full information requiring redefining the reward function. We chose not to add a comparison with GPMM for the highway intersection as we did not want to perform unfair comparisons. The cart-pole swing up task was also taken from [4], where the comparison is fairer from our perspective.
>
> **A lot of the results are done on cartpole, which is fine for trying to pull apart the method. Although it is a bit concerning because cartpole famously works even when the underlying RL algorithm is broken.**
>
> We thank the reviewer for this comment. From our point of view the Cart-Pole Swing-Up task is challenging enough while also simple enough to make evaluations of the approach. We are aware of some issues regarding the stabilization task in the literature, but not the swing up task.
>
> **I would like to see results on further hard RL environments.**
>
> Thank you for this comment! We will try to address this comment in the coming week by adding more experiments with environments with larger number of states.
>
> **In Table 3, the environment is called Highway. At the bottom of page 8 (bold text) it's called Intersection.**
>
> Corrected.
>
> References:
>
> [1] RL2 : FAST REINFORCEMENT LEARNING VIA SLOW REINFORCEMENT LEARNING Yan Duan, John Schulman, Xi Chen, Peter L. Bartlett, Ilya Sutskever, Pieter Abbeel
>
> [2]  Annie Xie, James Harrison, and Chelsea Finn. Deep reinforcement learning amidst lifelong non-stationarity.arXiv preprint arXiv:2006.10701, 20
>
> [3] Ilya Kostrikov. Pytorch implementations of reinforcement learning algorithms.https://github.com/ikostrikov/pytorch-a2c-ppo-acktr-gail, 2018
>
> [4] Mengdi Xu, Wenhao Ding, Jiacheng Zhu, Zuxin Liu, Baiming Chen, and Ding Zhao. Task-agnostic online reinforcement learning with an infinite mixture of Gaussian processes. Advances in Neural Information Processing Systems, 33, 2020
>
> [5] Yee Whye Teh, Michael I Jordan, Matthew J Beal, and David M Blei. Hierarchical Dirichlet processes. Journal of the American Statistical Association, 101(476):1566–1581, 2006.
>
> [6] Generalization in Reinforcement Learning by Soft Data Augmentation
> Nicklas Hansen, Xiaolong Wang
>
> [7] Image Augmentation Is All You Need: Regularizing Deep Reinforcement Learning from Pixels. Ilya Kostrikov, Denis Yarats, Rob Fergus

---

> > ### Comment · Reviewer_w49w · 2021-11-19
> > **Highway Intersection**
> >
> > Hi,
> >
> > Thank you for taking the time to write such a detailed response.
> >
> > Overall, I remain unconvinced that highway-intersection is a challenging or interesting environment. While I do appreciate that it was used in Xu, I think it was also a mistake for them to use it. Cartpole swingup is also quite easy.
> >
> > I found your comments about distillation and RL2 to be insightful, and cleared up some confusion I previously had. Further, your comments about the difference between this setting and domain shift/meta learning were insightful, although ultimately I remain unconvinced that this is a truly distinct setting. Meta and continual RL already have on the fly context switching. I suppose I just find the paper oddly positioned. But I'm fully willing to admit that's me.
> >
> > Overall, the changes are positive and I will raise my score. I think it's a good piece of research.

---

> > > ### Author Response · Authors · 2021-11-19
> > > **We thank the reviewer for kind words, our discussion and raising the score!**
> > >
> > > We hope that our discussion will help future readers to understand finer points of our approach!
> > >
> > > On the final note, we revised our qualitative assessment of Highway and Drone environments as "high-dimensional" and simply state "twelve dimensional" to make things less vague.

---

> ### Author Response · Authors · 2021-11-29
> **Please let us know if any further clarifications are required!**
>
> We are happy to respond to any further requests for clarification!

---

### Official Review · Reviewer_w8qw · 2021-11-02

**Correctness:** 3
**Technical Novelty And Significance:** 3
**Empirical Novelty And Significance:** 2
**Recommendation:** 6
**Confidence:** 4

**Details Of Ethics Concerns:**

No ethics concerns.

**Main Review:**

Pros:
- This paper is well-written and the proposed methods are discussed in detail. The literature review is pretty comprehensive and the authors did a good job at classifying the related works.
- As far as I can tell, the idea is novel although it is built on a line of previous works.
- The theoretical results are principled and clearly defined. I didn’t check all the details but it seems the proofs are good.
- The experiments are relatively well done, although I would suggest including more baselines.

Cons:
- More details need to be added to the algorithm part. For example, you could point out the update rule or algorithm rather than just say “Update generative model/policy parameters”. Also, I suggest putting one overall algorithm in the main text. In addition, is the distillation step supposed to be included in algorithm A1?
- In general, the model is relatively complex. Ideally, a principled Bayesian inference procedure estimated the posterior distribution for all the parameters. In this work, the parameter (e.g. the transition functions) are obtained by point estimation. What will be the error caused by this approximation?
- Experiments are not sufficient. In terms of the learning curve, only comparing with CEM could not be sufficient. It seems that CEM is even outperforming the proposed method in this case. Adding more natural baselines and the corresponding training curve will be a strong plus.

Minor concerns:
- It would be better to introduce the full name of the methods (e.g. Soft Actor-Critic) before using their abbreviations.
- Figure 5 is wrongly cropped.

Question:
In the related work [Xu et al., 2020], their method can model an infinite number of environments based on the Dirichlet process. In this work, the number of contexts is always truncated by a fixed number, is there any possibility that this work can deal with the potential infinite contexts with HDP?


**Summary Of The Paper:**

This paper proposed a contextual Markov Decision Process with a Hierarchical Dirichlet Process transition prior (HDP-C-MDP) that aims at solving the problem of non-stationary changing environments. Specifically, the context variable is obtained via a truncated variational inference and a context distillation method is applied to remove the spurious ones. The authors also provide theoretical studies concerning the distillation method and the intuition that performance improvement could be brought by knowing the underlying contexts. The authors compared their proposed method with a few reasonable baselines and demonstrated the effectiveness of their method.

**Summary Of The Review:**

This paper addresses a very interesting problem where there exists unknown nonstationarity in RL problems. The proposed method takes the Hierarchical Dirichlet Process as the transition prior. Overall the idea of using HDP is original. However, this paper still lacks sufficient experimental results that make the proposed method substantially stand out. I will give a 6 but and I am still on the boardline.

---

> ### Author Response · Authors · 2021-11-15
> **Thank you for the careful consideration of the manuscript and for a very constructive criticism!**
>
> Please see the revised version of the manuscript in the pdf file. All significant changes are in magenta.  We apologize for possible new typos, but we wanted to respond as fast as we could. Note for improved readibility of the table of contexts (requested by the other reviewer) we have changed the appendix section labels. See our answers below.
>
> **More details need to be added to the algorithm part. For example, you could point out the update rule or algorithm rather than just say “Update generative model/policy parameters”. Also, I suggest putting one overall algorithm in the main text. In addition, is the distillation step supposed to be included in algorithm A1?**
>
> Thank you for the comment!  We added a high-level pseudo code of the main algorithm in the main text and a few comments discussing the missing details. We added a distillation step in the algorithm as well to correct the oversight pointed out by the reviewer.
>
> **In general, the model is relatively complex. Ideally, a principled Bayesian inference procedure estimated the posterior distribution for all the parameters. In this work, the parameter (e.g. the transition functions) are obtained by point estimation. What will be the error caused by this approximation?**
>
> Thank you for the comment! We used the parameters point estimates for the parameters $\nu$ (parameters of the base DP distribution in the HDP) in order to promote uni-modality in the Markov chain parameters, which is the desirable outcome. Further, the full Beta variational distribution for $\nu$ is supposed to be uni-modal, hence a point estimation can be a reasonable simplification. Additionally, optimizing over Beta distribution can potentially lead to vanishing gradients. These claims are somewhat long to present and therefore we kindly refer the reviewer to Appendix C.3, where we provide some empirical evidence to support this claim using toy examples. We apologize for not presenting the claims here.
>
>  **Experiments are not sufficient. In terms of the learning curve, only comparing with CEM could not be sufficient. It seems that CEM is even outperforming the proposed method in this case. Adding more natural baselines and the corresponding training curve will be a strong plus.**
>
> Thank you for the comment! We apologize for this confusion the label in the figure should have read C-CEM, i.e., CEM is used with our HDP-C-MDP model as well, hence it is not a competing baseline, but rather another version of our algorithm. Our main message from these experiments with CEM and SAC is that that our model is amenable for the use with MPC and policy gradient algorithms alike. While CEM works better on a low-dimensional cart-pole swing-up it struggles on the intersection environment. Furthermore, the policy learning curves suggest that the belief model estimation/belief model learning does not restrict the policy learning process.
>
> Finding and choosing natural baselines for our algorithm was a challenge as most of the methods have either too restrictive assumptions (meta-RL or continual-RL) or too general assumptions (POMPD). Nevertheless, we aim to add further baselines as the reviewer suggested.
>
> We added some experiments (e.g., Appendix E4 - Breaking assumptions) already and we hope to add more by the deadline
>
> **Minor concerns:**
>
> Corrected
>
> **Question: In the related work [Xu et al., 2020], their method can model an infinite number of environments based on the Dirichlet process. In this work, the number of contexts is always truncated by a fixed number, is there any possibility that this work can deal with the potential infinite contexts with HDP?**
>
> Thank you for the question! Firstly, our model differs significantly from Xu et al., 2020 as they do not model context transitions making their setup easier to represent in the infinite dimensional case. However, this limits the generality of their model as we demonstrated in our experiments. Further, the number of contexts can always be upper bounded by the number of steps in the observed trajectories. Therefore, from our point of view, an HDP model with infinite discrete contexts is rarely necessary. The reviewer is correct that the number of contexts must be finite as we perform the gradient ascent over ELBO requiring the finite number of parameters. However, using a different learning method or a different HDP representation (e.g., Chinese restaurant franchise) can potentially result in a different solution (see for example [1]) more amenable for the infinite context set cardinality assumption.
>
> Finally, even using our current algorithm, we can always add further contexts if required and relearn the context transition model as we have demonstrated in Appendix E.5 (Learning new contexts).
>
> [1] Yee Whye Teh, Michael I Jordan, Matthew J Beal, and David M Blei.  Hierarchical Dirichlet processes. Journal of the American Statistical Association, 101(476):1566–1581, 2006.

---

> ### Author Response · Authors · 2021-11-19
> **New baselines and experiments**
>
> We added further baselines in Appendix E.6. We used an algorithm similar to GPMM [1], but used Deep Neural Network and Attentive Neural Process mixture models ([2], [3]) as was used in [1]. The new experimental results are consistent with the existing ones.
>
> These experiments are in addition to the ones we added in our first series of responses to the reviewers:
>
> In Appendix C.3. we added a few toy examples explaining our design choices in model learning algorithm.
>
> We added further experiments in Appendix E2 (Distillation acts as a regularizer), where we increased $K$ to $100$ and obtained similar results.
>
> In Appendix E.4 (Breaking assumptions in Hidden Markov models), we modeled the ground truth context transitions by non-Markov processes: in the first experiment the next context depends on the previous context, instead of the current one, in the second experiment, the context transition is state-dependent. In both cases, our approach was successful in estimating current contexts.
>
> We hope that this additional baselines and experiments showcase the performance of our approach in a better way.
>
> **References:**
>
> [1] Mengdi Xu, Wenhao Ding, Jiacheng Zhu, Zuxin Liu, Baiming Chen, and Ding Zhao. Task-agnosticonline reinforcement learning with an infinite mixture of Gaussian processes. Advances in NeuralInformation Processing Systems, 33, 2020
>
> [2] Hyunjik Kim, Andriy Mnih, Jonathan Schwarz, Marta Garnelo, Ali Eslami, Dan Rosenbaum, OriolVinyals, and Yee Whye Teh. Attentive neural processes.arXiv preprint arXiv:1901.05761, 2019
>
> [3] Shenghao Qin, Jiacheng Zhu, Jimmy Qin, Wenshuo Wang, and Ding Zhao. Recurrent attentive neuralprocess for sequential data.arXiv preprint arXiv:1910.09323, 2019

---

> ### Author Response · Authors · 2021-11-29
> **Please let us know if any further clarifications are required!**
>
> We are happy to respond to any further requests for clarification!

---

### Official Review · Reviewer_q8Tn · 2021-11-06

**Correctness:** 3
**Technical Novelty And Significance:** 2
**Empirical Novelty And Significance:** 2
**Recommendation:** 6
**Confidence:** 3

**Main Review:**

Strengths:
- This paper studies a pretty difficult problem, and proposes to leverage the Hierarchical Dirichlet Process, which can accurately model changes in the agent’s environment and adapt its behavior accordingly.

- It is overall well written and very thorough. Similarly, the experimental section studies several alternative priors and RL algorithms in non-stationary settings, and probes their learned model in various ways.

- The experiment in Appendix A5.4 shows (in a simple experiment) that the model is flexible enough to accommodate new contexts, which is an important capability for continual/lifelong settings.

Weaknesses:
- At times, the paper becomes convoluted and overloaded with details. For example, the presentation of the HDP could be condensed. I would also suggest a table of contents for the appendix.

- In the experiments, there seems to be a tension between the selected K (cardinality of context set) and distillation threshold. Specifically, if one overestimates K significantly, then the distillation threshold needs to be larger to compensate for an increased number of spurious contexts. However, the threshold can’t be so large that it prunes out relevant contexts. So, careful selection of these hyperparameters is important.

- The experimental settings are relatively simple as they operate in low-dimensional state-action spaces (the largest being 12-dim states and 4-dim actions). The settings also have a relatively small number of contexts that they switch between. I’m curious how well the HDP prior can model large numbers of modes (for example, 100) which meta-RL algorithms often operate in.

- The contexts in all three experimental domains correspond to multiplying the actions by some constant. To see how scalable the model is, it’d be useful to study contexts with multiple variables that parameterize the transition dynamics in more complex ways.

- Assumptions about the environment have to be made for their model to work well, which could be unrealistic in some setups. How effective is the HDP when the assumptions do not hold? For example, what if the mode transitions are not Markovian and depend on earlier modes, or depend on the states and actions? What if we underestimated the cardinality of the context set?

Finally, there are some details that are unclear to me:
- In the comparison to RNN-PPO, is there a reconstruction loss on prediction of the next state s_t+1? This supervision should be useful for the RNN model, but it’s not clear if it’s added based on the description in Appendix A4.1.

- How do the model learning and policy learning components interact, e.g., are they trained simultaneously? This detail seems to be missing in Section 3.

Other comments:
- Some of the titles and axes of Figures 5(b-d) are cut off.

- In Tables 2 and 3, it’d be clearer if the highest number of each column were bolded.


**Summary Of The Paper:**

This paper proposes to model mode transitions (where each mode corresponds to a different MDP) in reinforcement learning with the Hierarchical Dirichlet Process (HDP) prior in a contextual MDP. It makes some assumptions about the evolving environment, such as how it can only take on a finite number of modes and the mode transitions are Markovian. During model learning, it also prunes out modes that have low transition probabilities into them to avoid modeling spurious contexts.

**Summary Of The Review:**

The paper studies a challenging problem setting, and demonstrates the effectiveness of the HDP to model this particular setting. The experimental section presents a lot of varied analysis of their model already, but is missing results that demonstrate its scalability to more complex environments. Since the main argument of this work is that the HDP is best suited here, I think there are aspects (described in the main review) that still need to be evaluated.

---

> ### Author Response · Authors · 2021-11-15
> **Thank you for carefully evaluating our work and for a very constructive criticism! (response 1)**
>
> We want to thank the reviewer for carefully evaluating our work and for a very constructive criticism, which improved our paper!  Please see the revised version of the manuscript in the pdf file. All significant changes are marked by magenta color. We apologize for possible new typos, but we wanted to respond as fast as we could.
>
> **At times, the paper becomes convoluted and overloaded with details. For example, the presentation of the HDP could be condensed. I would also suggest a table of contents for the appendix.**
>
> Thank you for this comment, we have shortened the HDP section and added a table of contents, which made appendix more readable!
>
> **In the experiments, there seems to be a tension between the selected K (cardinality of context set) and distillation threshold. Specifically, if one overestimates K significantly, then the distillation threshold needs to be larger to compensate for an increased number of spurious contexts. However, the threshold can’t be so large that it prunes out relevant contexts. So, careful selection of these hyperparameters is important.**
>
> Thank you for the comment!
>
> We would like to stress that choosing a good threshold can only improve the model learning procedure by reducing uncertainty. However, setting the threshold equal to zero during training already produces better results than Dirichlet priors or MLE. In this case, one could use distillation after training, which can be done by carefully choosing the threshold
>
> Furthermore, we highlighted that the ambiguity in the choice of the distillation threshold can work at our advantage by merging the contexts automatically! We suggest treating the distillation threshold as a design choice rather than a hyper-parameter.
>
> Having said that, it is certainly true that there is trade-off between presumed number of contexts and the optimal distillation threshold, which we felt was important to highlight from a practical point of view. We could not think of a reason why there would be a general rule for solving this trade-off (similarly to L-1 sparse regression or LASSO), but we felt it is important to highlight that there is such a trade-off that could potentially improve the results during training.
>
> Overall, based on our experiments we recommend choosing a large K and treating the distillation threshold as a design choice rather a than hyper-parameter.
>
> **The contexts in all three experimental domains correspond to multiplying the actions by some constant. To see how scalable the model is, it’d be useful to study contexts with multiple variables that parameterize the transition dynamics in more complex ways.**
>
> Thank you for this comment! We chose the simple cases so as to demonstrate interpretability of our model. We aim to address this comment by performing some additional experiments with increased context set cardinality in further responses.
>
> **In the comparison to RNN-PPO, is there a reconstruction loss on prediction of the next state $s_t+1$? This supervision should be useful for the RNN model, but it’s not clear if it’s added based on the description in Appendix A4.1.**
>
> Thank you for this comment! The RNN model is used only for the belief estimation and prediction thus we would not be able to use the loss on prediction of $s_{t+1}$ in our training. We reasoned that for a model-free algorithm we only need a belief estimation process and we can remove the state predictions. We note that the RNN in this PPO implementation is learned while having the access to full trajectories.
>
> EDIT: Appendix A4.1 is now Appendix D.1
> References:
>
> [1] Zhe Dong, Bryan Seybold, Kevin Murphy, and Hung Bui. Collapsed amortized variational inference for switching nonlinear dynamical systems. In International Conference on Machine Learning, pp. 2638–2647. PMLR, 2020
>
> [2] Philip Becker-Ehmck, Jan Peters, and Patrick Van Der Smagt. Switching linear dynamics for
> variational Bayes filtering. arXiv preprint arXiv:1905.12434, 2019.
>
> [3]  Annie Xie, James Harrison, and Chelsea Finn. Deep reinforcement learning amidst lifelong non-stationarity. arXiv preprint arXiv:2006.10701, 20
>
> [4] Anusha Nagabandi, Chelsea Finn, and Sergey Levine. Deep online learning via meta-learning: Continual adaptation for model-based RL. In International Conference on Learning Representations,2018
>
> [5] Mengdi Xu, Wenhao Ding, Jiacheng Zhu, Zuxin Liu, Baiming Chen, and Ding Zhao. Task-agnostic online reinforcement learning with an infinite mixture of Gaussian processes. Advances in Neural Information Processing Systems, 33, 2020
>
> [6] Yash Chandak, Georgios Theocharous, James Kostas, Scott M. Jordan, and Philip S. Thomas.
> Learning action representations for reinforcement learning. In Kamalika Chaudhuri and Ruslan
> Salakhutdinov (eds.), Proceedings of Machine Learning Research, pp. 941–950. PMLR, 2019.

---

> ### Author Response · Authors · 2021-11-15
> **response 2 (continuation of 1)**
>
> **The experimental settings are relatively simple as they operate in low-dimensional state-action spaces (the largest being 12-dim states and 4-dim actions). The settings also have a relatively small number of contexts that they switch between. I’m curious how well the HDP prior can model large numbers of modes (for example, 100) which meta-RL algorithms often operate in.**
>
> Thank you for the comment! We would like to draw reviewer's attention that the environments in [3-6] (which are comparable studies from our perspective) are of comparable complexity in the number of states. Furthermore, even though the intersection environment has only 12 states, solving the task is not easy as we need to reach the goal on the road, while avoiding crashing into incoming vehicle.
> Regarding a comparison to other meta-RL algorithms, there is a major difference between our frameworks. Our contexts change during the episode and hence we need to adapt very fast to a new context, which is feasible if we assume that there is a finite number of contexts. In our case, as long as we estimate the context we can optimally compute the policy. Some meta-RL methods can assume infinite cardinality of the context set, however, at the expense of fast adaptation. Hence comparison to most of the meta-RL algorithms is not entirely fair. That is most of the meta-RL algorithms would work better in some settings (many contexts that switch only before the episode starts), but worse in other settings (fewer contexts that switch during the episode). We chose [5] as a representative baseline to show that meta-RL methods can fail in our setting.
>
> We added further experiments in Appendix~E2 to demonstrate the situations with a large number of contexts for cart-pole swing-up experiments. In particular, we increased the number of states in HDP to $100$.
>
> We continue working on adding new experiments and new environments, which we hope to finish by the deadline.
>
> **How do the model learning and policy learning components interact, e.g., are they trained simultaneously? This detail seems to be missing in Section 3.**
>
> Thank you for this comment! We apologize for this oversight. We added the details at the expense of some experiments. But the short answer we follow the standard model-based scheme:
>
> *  update the model parameters using gradient ascent on ELBO
> * rollout trajectories new trajectories:
> * Update policy if needed:
>    - Relabel beliefs in the observed data for policy update
>    - Update the policy parameters
>
>
> **Minor Comments**
>
> * Some of the titles and axes of Figures 5(b-d) are cut off.
> * In Tables 2 and 3, it’d be clearer if the highest number of each column were bolded.
>
> Corrected
>
> References:
>
> [1] Zhe Dong, Bryan Seybold, Kevin Murphy, and Hung Bui. Collapsed amortized variational inference for switching nonlinear dynamical systems. In International Conference on Machine Learning, pp. 2638–2647. PMLR, 2020
>
> [2] Philip Becker-Ehmck, Jan Peters, and Patrick Van Der Smagt. Switching linear dynamics for
> variational Bayes filtering. arXiv preprint arXiv:1905.12434, 2019.
>
> [3]  Annie Xie, James Harrison, and Chelsea Finn. Deep reinforcement learning amidst lifelong non-stationarity. arXiv preprint arXiv:2006.10701, 20
>
> [4] Anusha Nagabandi, Chelsea Finn, and Sergey Levine. Deep online learning via meta-learning: Continual adaptation for model-based RL. In International Conference on Learning Representations,2018
>
> [5] Mengdi Xu, Wenhao Ding, Jiacheng Zhu, Zuxin Liu, Baiming Chen, and Ding Zhao. Task-agnostic online reinforcement learning with an infinite mixture of Gaussian processes. Advances in Neural Information Processing Systems, 33, 2020
>
> [6] Yash Chandak, Georgios Theocharous, James Kostas, Scott M. Jordan, and Philip S. Thomas.
> Learning action representations for reinforcement learning. In Kamalika Chaudhuri and Ruslan
> Salakhutdinov (eds.), Proceedings of Machine Learning Research, pp. 941–950. PMLR, 2019.

---

> ### Author Response · Authors · 2021-11-15
> **response 3 (continuation of 2)**
>
> **Assumptions about the environment have to be made for their model to work well, which could be unrealistic in some setups. How effective is the HDP when the assumptions do not hold? For example, what if the mode transitions are not Markovian and depend on earlier modes, or depend on the states and actions? What if we underestimated the cardinality of the context set?**
>
> Thank you for this comment! We added several simple examples to showcase the situations where the HDP assumptions do not hold, i.e., the transitions are non-Markovian (transition probabilities depend on the previous modes), the transitions are state-dependent (see Appendix~E4). In both cases, our model was able to estimate the correct current contexts. We believe this happens because the model takes states into account for current context estimation. Naturally, we cannot claim that the HDP model would always work for non-Markovian or state-dependent cases, but our model is generalizable to some of these cases.
>
> Overall, we argue that our setting offers another design choice to the ones existing in the literature. While we agree that our assumptions can be unfulfilled in some cases, but we would argue that so are any other assumptions. One could claim there is always a trade-off in choosing how general/restrictive assumptions are. Too general assumptions could result in an over-fit, poor performance in RL training (as we demonstrate with RNN-PPO experiments), while too restrictive assumptions can result in an under-fit and poor predictions (as we demonstrate with GPMM experiments).
>
> Underestimating the true context set cardinality can lead to failure, but we have demonstrated that the model can try merging different context (see Appendix~E3). In general, we still recommend overestimating K and choosing the distillation threshold appropriately. In a cart-pole swing-up experiment we have 2 meaningful contexts and set K=100, while still obtaining a sufficiently good model.
>
> We hope to add additional examples showcasing the underestimation case.
>
> References:
>
> [1] Zhe Dong, Bryan Seybold, Kevin Murphy, and Hung Bui. Collapsed amortized variational inference for switching nonlinear dynamical systems. In International Conference on Machine Learning, pp. 2638–2647. PMLR, 2020
>
> [2] Philip Becker-Ehmck, Jan Peters, and Patrick Van Der Smagt. Switching linear dynamics for
> variational Bayes filtering. arXiv preprint arXiv:1905.12434, 2019.
>
> [3]  Annie Xie, James Harrison, and Chelsea Finn. Deep reinforcement learning amidst lifelong non-stationarity. arXiv preprint arXiv:2006.10701, 20
>
> [4] Anusha Nagabandi, Chelsea Finn, and Sergey Levine. Deep online learning via meta-learning: Continual adaptation for model-based RL. In International Conference on Learning Representations,2018
>
> [5] Mengdi Xu, Wenhao Ding, Jiacheng Zhu, Zuxin Liu, Baiming Chen, and Ding Zhao. Task-agnostic online reinforcement learning with an infinite mixture of Gaussian processes. Advances in Neural Information Processing Systems, 33, 2020
>
> [6] Yash Chandak, Georgios Theocharous, James Kostas, Scott M. Jordan, and Philip S. Thomas.
> Learning action representations for reinforcement learning. In Kamalika Chaudhuri and Ruslan
> Salakhutdinov (eds.), Proceedings of Machine Learning Research, pp. 941–950. PMLR, 2019.

---

> > ### Comment · Reviewer_q8Tn · 2021-11-20
> > **Thanks for your response**
> >
> > Hi, thank you for the detailed response and additional experiments. They’ve addressed/clarified majority of my concerns.
> >
> > One remaining concern I have is that the experiments study contexts that are trivially separable, i.e., they correspond to different multiplicative constants that scale the actions. For me, this is different from the context set cardinality. I’m curious about the performance on, for example, the contextual CartPole env from [1], which introduces several context features that affect the dynamics.
> >
> > Overall, I'm impressed by the thoroughness of the new results and have raised my score.
> >
> > [1] Benjamins et al. CARL: A Benchmark for Contextual and Adaptive Reinforcement Learning.

---

> > > ### Author Response · Authors · 2021-11-23
> > > **final experiment**
> > >
> > > Hi, we have conducted further experiments with a larger number of ground truth contexts. This experiment illustrated a possible interplay between various parameters and its influence on context estimation. Our environment mimics some context settings from [1], which we cited in the paper as possible future work. The experimental results are in Appendix E.7 (Experiments with larger number of contexts). Note that we used $16$ ground truth contexts to obtain easily interpretable and visualizable results.
> > >
> > > [1] Benjamins et al. CARL: A Benchmark for Contextual and Adaptive Reinforcement Learning.

---

> > > > ### Comment · Reviewer_q8Tn · 2021-11-29
> > > > **Thanks for the additional experiment**
> > > >
> > > > Thank you for the addition of this experiment. It's definitely helpful to see how the model handles multiple hidden context variables. It's interesting to see that the model clusters some of the contexts, which might generate similar MDPs, into a smaller group.

---

> > > > > ### Author Response · Authors · 2021-11-29
> > > > > **a further clarification**
> > > > >
> > > > > Thank you for the response and suggesting this experiment in the first place!
> > > > >
> > > > > We wanted to add an interpretation of our numerical results using the model of the cart pole, which in continuous-time can be written as follows (cf. [lecture notes](https://ocw.mit.edu/courses/electrical-engineering-and-computer-science/6-832-underactuated-robotics-spring-2009/readings/MIT6_832s09_read_ch03.pdf))
> > > > >
> > > > > $\ddot x = f_1(\theta, \dot \theta, \dot x)= \dfrac{F_{\rm max} a - \mu \dot x + m_p\sin(\theta) (l \dot \theta^2 + g \cos(\theta)) }{m_c+m_p \sin^2(\theta)}$,
> > > > >
> > > > > $\ddot \theta = f_2(\theta, \dot \theta, \dot x) =  \dfrac{- (F_{\rm max} a - \mu \dot x) \cos(\theta) - m_p l \dot \theta^2\cos(\theta)\sin(\theta) - (m_c + m_p) g \sin(\theta) }{l (m_c+m_p \sin^2(\theta))},$
> > > > >
> > > > > where $m_p$, $m_c$ are masses of the pole and the cart respectively, $F_{\rm max}$ - the maximum force, $\mu$ - the friction coefficient, $g$ - the gravity constant, $l$ - the length of the pole, $x$ - the position of the cart, $\theta$ - the pole angle with respect to the vertical axis.
> > > > >
> > > > > We can directly make a few observations:
> > > > >
> > > > > * for relatively small cart velocity values $\dot x$ the effect of changing the friction coefficient $\mu$ from $0$ to $0.1$ is negligible for $F_{\rm max} = 20$ or $F_{\rm max} = 40$
> > > > >
> > > > > * the effect of changing $m_c$ from $0.5$ to $5$ is substantial in $f_1$, i.e., the acceleration $\ddot x$
> > > > >
> > > > > Now let us study the effect of changes in gravity and the maximum force. Taking the derivative of $f_1$ and $f_2$ with respect to $g$ and $F_{\rm max}$ gives us:
> > > > >
> > > > > $\dfrac{\partial f_1}{\partial g} =  \dfrac{m_p\sin(\theta)  \cos(\theta) }{m_c+m_p \sin^2(\theta)}, ~~~~~~~~~~~~~~\dfrac{\partial f_1}{\partial F_{\rm max}} =  \dfrac{-a }{m_c+m_p \sin^2(\theta)},$
> > > > >
> > > > > $\dfrac{\partial f_2}{\partial g} =  \dfrac{-(m_c + m_p) \sin(\theta) }{l(m_c+m_p \sin^2(\theta)},~~~~~~~~~~\dfrac{\partial f_2}{\partial F_{\rm max}} =  \dfrac{-a\cos(\theta)}{l(m_c+m_p \sin^2(\theta))} $
> > > > >
> > > > > We can observe that if $m_c =m_p= 0.5$ , then the values of $\dfrac{\partial f_i}{\partial g}$ and $\dfrac{\partial f_i}{\partial F_{\rm max}}$ are comparable in magnitudes. However, for larger values of $m_c$ the magnitudes of the derivatives with respect to $g$ are larger than the ones with respect to $F_{\rm max}$.
> > > > >
> > > > > These observations could perhaps explain why our algorithm still struggles to fully separate the learned contexts $Z_{10}=${$C_4, C_5, C_{12}, C_{13}$} and $Z_{18} =$ {$C_0, C_8$}. Recall that in all these cases $m_c = 0.5$, in $C_0,$ $C_8$, the gravity is set to $9.82$ , while in $C_4$, $C_5$, $C_{12}$, $C_{13}$ it is set to $50$. In  $C_0,$ $C_4$, $C_8$, $C_{12}$ the maximum force is $F_{\rm max} = 40$, while $C_5$, $C_{13}$ it is set to $20$. Therefore, the gravity parameter is the main difference between $Z_{10}$ and $Z_{18}$. However, it appears that the increased force makes it harder to separate $C_4$ and $C_0$, $C_8$. This is consistent with the analysis above, since the derivatives of $f_i$ with respect to $F_{\rm max}$ and $g$ are of similar magnitude for $m_c=0.5$.
> > > > >
> > > > > Please let us know if any further clarifications are required!

---

### Official Review · Reviewer_HPaX · 2021-11-07

**Correctness:** 3
**Technical Novelty And Significance:** 4
**Empirical Novelty And Significance:** 3
**Recommendation:** 8
**Confidence:** 3

**Main Review:**

Weakness:
- It would be informative to see some experiments with problems that have already been tackled in the literature  (in the context-dependent RL or PO-MDPs framework), to see how the developed method compares to existing ones on problems they were optimized for.
- Results for the running time are not presented. It would be interesting to see though; it is not clear how efficiently the proposed algorithm can be implemented.
- (4) seems to be quite a simplification. Can you provide some intuition on how good/bad it is in some realistic examples?
- The paper is lacking a concise presentation of the algorithm

Strength:
- The method is based on an elegant approach with Dirichlet Processes
- Solves a previously untackled but realistic problem
- The algorithm outperforms the baselines
- Clear intuition is provided for most steps


Further questions:
-----------------
Could the proposed method work in the more general transition learning setup?

**Summary Of The Paper:**

The paper proposes a novel approach for policy learning in a context-dependent reinforcement learning setting (a subset of POMDPs). In particular, the approach is a Bayesian method using a generative model based on a hierarchical Dirichlet process.

The experiments demonstrate that the proposed method performs well even on tasks where the benchmark algorithms chosen by the authors struggle or even fail.


**Summary Of The Review:**

The proposed method is based on an elegant approach with Dirichlet Processes, and performs well on tasks where the benchmark algorithms chosen by the authors struggle or even fail. Additional experiments, metrics and explanations would be informative though.

---

> ### Author Response · Authors · 2021-11-15
> **Thank you for the comments and kind words!**
>
> We want to thank the reviewer for carefully evaluating our work and for kind words about it!  Please see the revised version of the manuscript in the pdf file. All significant changes are in magenta.  We apologize for possible new typos, but we wanted to respond as fast as we could.
>
> **It would be informative to see some experiments with problems that have already been tackled in the literature (in the context-deendent RL or PO-MDPs framework), to see how the developed method compares to existing ones on problems they were optimized for.**
>
> Thank you for this comment! We agree that this is a preferable option and therefore we chose the cart-pole swing-up environment to highlight why other approaches can fail (they don't necessarily fail in general). Choosing other problems tailored to existing methods was slightly problematic as our case is more general than many meta-RL / continual RL algorithms can handle, and less general than many POMDPs methods should be applied to!
>
> We chose highway-intersection as it was used by Xu 2020 et al, which we use as a baseline. In fact, we used their modifications of the environment and then adapted it to our setting. However, their environment had a very small time horizon (40 time steps) and increasing the time horizon made the MPC approach much less effective even with full information requiring redefining the reward function. We chose not to add a comparison with GPMM for the highway intersection as we did not want to perform unfair comparisons. The cart-pole swing up task was also taken from Xu et.al 2020, where the comparison is fairer from our perspective as Xu et al designed the environment and we have modified only the context transitions.
>
> We continue working on adding new experiments and new environments, which we hope to finish by the deadline.
>
> **Results for the running time are not presented. It would be interesting to see though; it is not clear how efficiently the proposed algorithm can be implemented.**
>
> Thank you for this question! We timed the solution for the cart-pole and drone environments. For the cart-pole environment, C-SAC without distillation during training was 2.0 times slower than FI-SAC (full information, i.e. no model was used),  C-SAC with distillation during training was 2.4 times slower than FI-SAC. For the drone environment, C-SAC without distillation during training was 4.0 times slower than FI-SAC,  C-SAC with distillation during training was 4.4 times slower than FI-SAC. All the hyperparameters and settings were similar for the runs of C-SAC and FI-SAC. These numbers give an indication that learning the model does not add a catastrophic overhead in policy learning.
>
> **(4) seems to be quite a simplification. Can you provide some intuition on how good/bad it is in some realistic examples?**
>
> Thank you for this question! The mean field assumption is a commonly used technique in variational inference in order to increase tractability of the algorithm. We added the following reference for discussion on the mean filed assumption
>
> Blei, David M., Alp Kucukelbir, and Jon D. McAuliffe. "Variational inference: A review for statisticians." Journal of the American statistical Association 112.518 (2017): 859-877
>
> We provided additional details in Appendix C.3, where we also discuss the choice of point estimates of some parameters.
>
> EDIT: equation (4) is now equation (5)
>
> **The paper is lacking a concise presentation of the algorithm**
>
> Thank you for the comment and we apologize for this oversight!  We added a high-level pseudo code of the main algorithm in the main text and a few comments discussing the missing details.
>
> **Could the proposed method work in the more general transition learning setup?**
>
> Thank you for the comment! We do not anticipate major theoretical issues, if a Gaussian one-step transition model is replaced with other probabilistic models. However, we have not tested experimentally such variations.
>
> Furthermore, a combination of a DP mixture with image classification tasks was reported in:
>
> Lee, Soochan, et al. "A neural Dirichlet process mixture model for task-free continual learning." arXiv preprint arXiv:2001.00689 (2020).
>
> This approach can be directly extended to the HDP case by using the proposed method as well.

---

> ### Author Response · Authors · 2021-11-29
> **Please let us know if any further clarifications are required!**
>
> We are happy to respond to any further requests for clarification!

---

### Official Review · Reviewer_D986 · 2021-11-07

**Correctness:** 4
**Technical Novelty And Significance:** 3
**Empirical Novelty And Significance:** Not applicable
**Recommendation:** 8
**Confidence:** 2

**Main Review:**

CMDP is a powerful framework capable of modeling a broad range of applications. There is a growing literature on RL in CMPs, but most such works, to my knowledge, assume adversarially chosen contexts or contexts sampled in an i.i.d. fashion. The paper is amongst very few works considering the so-called Markovian contexts, an evolution model which, I believe, is relevant in many applications. Learning in CMDPs is a challenging task and so is the problem studied in this paper.

The studied problem, while interesting and important, is also highly relevant for the ICLR community as learning in this setting involves some representation learning.

The paper is well-written and mostly well-organized, and presents an adequate literature review. I have limited knowledge about learning in the CMDP setting here and the Bayesian approach employed. It appears to me that the authors rely on some elements from existing works while some elements sound novel. Nonetheless, due to challenging nature of the learning problem considered, I consider a fruitful orchestration of all these as a contribution. It is also evident that they solidly compare their model and chosen algorithmic elements to existing ones.

It is so pity that almost all pseudo-codes are presented in the appendix, and a large part of the main text is occupied by experimental results. Perhaps bringing a high-level pseudo-code of the main algorithm to the main text could improve the current organization.

I have one technical comment: The authors use a context distillation procedure to remove spurious contexts. Doesn’t removal of such contexts affect the Markovian property of the contexts?

Minor:

- p. 2: this is case => this is the case

- p. 3: independent and identically distributed => independently …



**Summary Of The Paper:**

This paper studies reinforcement learning where the environment is mathematically modeled as a contextual Markov decision process (CMDP) with finite action-space and context-space, and continuous state-space. The contexts, which are assumed unobservable, may abruptly evolve according to a Markov chain -- hence the name, discrete Markovian context evolution.

The main contribution of the paper is to present an algorithm for this setting, whose design follows a Bayesian approach. It also relies on Hierarchical Dirichlet Processes (HDP) as priors for context transitions, and uses a context distillation procedure to remove spurious contexts. The performance of the proposed method is demonstrated through numerical experiments.


**Summary Of The Review:**

This well-written paper presents a Bayesian learning algorithm for RL in CMDPs, which is an interesting and challenging task, and of high relevance to ICLR.

---

> ### Author Response · Authors · 2021-11-15
> **Thank you for the comments and kind words!**
>
> We want to thank the reviewer for carefully evaluating our work and for kind words about it!  Please see the revised version of the manuscript in the pdf file. All significant changes are in magenta.  We apologize for possible new typos, but we wanted to respond as fast as we could. See our response below
>
> **It is so pity that almost all pseudo-codes are presented in the appendix, and a large part of the main text is occupied by experimental results. Perhaps bringing a high-level pseudo-code of the main algorithm to the main text could improve the current organization.**
>
> Thank you for the comment! We added a high-level pseudo code of the main algorithm in the main text and a few comments discussing the missing details.
>
> **I have one technical comment: The authors use a context distillation procedure to remove spurious contexts. Doesn’t removal of such contexts affect the Markovian property of the contexts?**
>
> Our distillation procedure ensures that the distilled process is a Markov chain as shown in Theorem 1.
>
> Perhaps, it could be useful to see this approach from another perspective: we compute a new Markov chain with fewer number of states (contexts) than the original one. The new states (contexts) have the same interpretation as some of the states in the original Markov chain. Furthermore, the Markov chains are similar in the sense that the stationary distributions of the distilled states (contexts) and the subset of the original states (contexts) match. We hope this clarifies our approach.
>
> **Minor:**
>  * p. 2: this is case => this is the case
>  * p. 3: independent and identically distributed => independently …
>
> Corrected!

---

> ### Author Response · Authors · 2021-11-29
> **Please let us know if any further clarifications are required!**
>
> We are happy to respond to any further requests for clarification!

---

### Author Response · Authors · 2021-11-23
**List of added experiments**

We want to thank all the reviewers for their comments and we want to reiterate our gratitude for carefully reading our draft and for providing comments improving our work. To summarize, we have conducted the following experiments in the past two weeks and added them to our submission:

* In Appendix C.3 we added a few toy examples explaining our design choices in model learning algorithm.
* We added further experiments in Appendix E.2 (Distillation acts as a regularizer), where we increased $K$ to $100$ and obtained similar results.
* In Appendix E.4 (Breaking assumptions in Hidden Markov models), we modeled the ground truth context transitions by non-Markov processes: in the first experiment the next context depends on the previous context, instead of the current one, in the second experiment, the context transition is state-dependent. In both cases, our approach was successful in estimating current contexts.
* We added further baselines for the cart-pole experiments in Appendix E.6 We used an algorithm similar to GPMM [1], but used Deep Neural Network and Attentive Neural Process mixture models ([2], [3]) as was used in [1]. The new experimental results are consistent with the existing ones.
* In Appendix E.7 (Experiments with larger number of contexts), we conducted further experiments with a larger number of ground truth contexts and illustrated a possible interplay between various parameters and its influence on context estimation. Our environment mimics some context settings from [4].

We hope that this additional baselines and experiments showcase the performance of our approach in a better way.

References:

[1] Mengdi Xu, Wenhao Ding, Jiacheng Zhu, Zuxin Liu, Baiming Chen, and Ding Zhao. Task-agnostic online reinforcement learning with an infinite mixture of Gaussian processes. Advances in Neural Information Processing Systems, 33, 2020

[2] Hyunjik Kim, Andriy Mnih, Jonathan Schwarz, Marta Garnelo, Ali Eslami, Dan Rosenbaum, Oriol Vinyals, and Yee Whye Teh. Attentive neural processes.arXiv preprint arXiv:1901.05761, 2019

[3] Shenghao Qin, Jiacheng Zhu, Jimmy Qin, Wenshuo Wang, and Ding Zhao. Recurrent attentive neural process for sequential data.arXiv preprint arXiv:1910.09323, 2019

[4] Benjamins et al. CARL: A Benchmark for Contextual and Adaptive Reinforcement Learning

---

### Decision · Program_Chairs · 2022-01-20

**Decision:**

Accept (Poster)

**Comment:**

The paper proposes a Bayesian approach to learning in contextual MDPs where the contexts can dynamically vary during the episode.
The authors did well in their rebuttal and alleviated most of the reviewers' concerns. During the discussion there was an agreement that the paper should be accepted.
Please take all reviewer comments into account when preparing the final version.